# Diffusion Models in Simulation-Based Inference: A Tutorial Review

## Abstract

Diffusion models have recently emerged as powerful learners for simulation-based inference (SBI), enabling fast and accurate estimation of latent parameters from simulated and real data. Their score-based formulation offers a flexible way to learn conditional or joint distributions over parameters and observations, thereby providing a versatile solution to various modeling problems. In this tutorial review, we synthesize recent developments on diffusion models for SBI, covering design choices for training, inference, and evaluation. We highlight opportunities created by various concepts such as guidance, score composition, flow matching, consistency models, and joint modeling. Furthermore, we discuss how efficiency and statistical accuracy are affected by noise schedules, parameterizations, backbones, and samplers. Finally, we illustrate these concepts with case studies across parameter dimensionalities, simulation budgets, and model types and outline open questions for future research.

## 1 Introduction

Simulation-based inference (SBI; Cranmer et al., 2020) is concerned with a fundamental question of computational science: given a simulation model of a target system, what can we learn about the unknown parameters $\boldsymbol{\theta}$ of the system given manifest observations $\mathbf{y}$? The same question has been asked from different perspectives, including (Bayesian) statistics (Diggle & Gratton, 1984), inverse problems (Tarantola & Valette, 1982), uncertainty quantification (Klir, 2006), and deep learning (Kingma et al., 2014). SBI can be viewed as the intersection of these fields and involves the following three essential components (Figure 1):

1. A *simulator* that produces synthetic observations $\mathbf{y}$ given latent parameters $\boldsymbol{\theta}$;

2. A *prior* over the latent parameters, $p(\boldsymbol{\theta})$, which, implicitly or explicitly, specifies the domain of plausible parameters;

3. An *approximator* (e.g., a generative network) that can recover the distribution of latent parameters $\boldsymbol{\theta}$ from synthetic or real observations $\mathbf{y}$.

The recent progress in neural network architectures has directly translated into progress in SBI, with more powerful architectures leading to more powerful approximators for inverting simulators. The flurry of SBI methods and applications can largely be attributed to the realization that simulations serve as training data for (generative) neural networks.

Most recently, diffusion models, a generative model family, initially risen to fame for their ability to generate realistic natural images (Sohl-Dickstein et al., 2015; Song & Ermon, 2019; Ho et al., 2020), have permeated the SBI landscape (Table 3). At a surface level, diffusion models can serve as plug-in replacements for any generative model trained on pairs $(\boldsymbol{\theta}, \mathbf{y})$. Starting from pure noise, they learn to remove the noise gradually until a realistic target instance (e.g., parameters or observations) remains. Their appeal, however, goes beyond matching existing approaches: they employ a score-based formulation and unconstrained architectures (i.e., any kind of neural network can be used) to provide a level of flexibility that previous model families (e.g., Jacobian-constrained normalizing flows) cannot easily match (Chen et al., 2025b). This flexibility makes

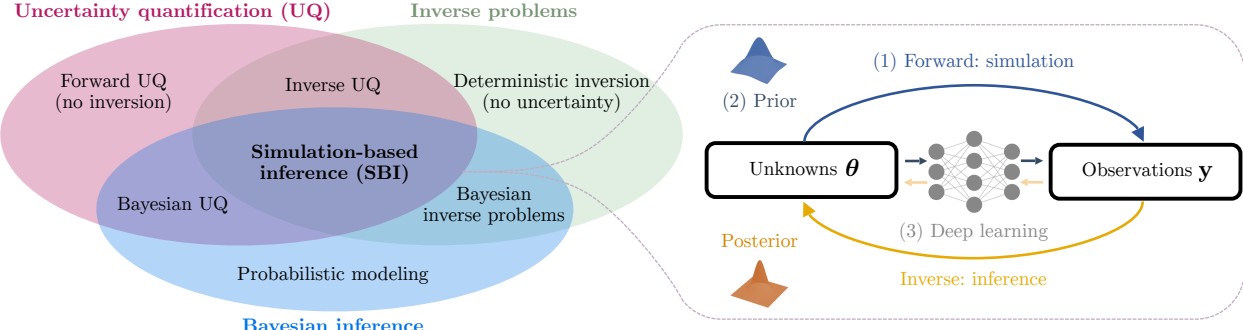

Figure 1: *The three overarching fields whose intersection gives rise to simulation-based inference (SBI).* Uncertainty quantification, inverse problems, and Bayesian inference. The ingredients of SBI are (1) a simulator that can generate synthetic observations **y** given latent parameters $\boldsymbol{\theta}$; (2) a prior over the latent parameters; and (3) an approximator (e.g., a diffusion model) that plays a role in estimating the posterior distribution of parameters from the observations.

them excel at the often idiosyncratic modeling setups encountered in SBI, where problem formulations may differ in parameterization, conditioning, or training objective, yet ultimately target the same inference goal (Geffner et al., 2023; Sharrock et al., 2024; Gloeckler et al., 2024).

As diffusion models have been rapidly adopted in SBI and now underpin numerous recent developments (Table 3), this paper addresses three needs (Figure 2): First, it introduces the foundations of SBI (Section 2) and diffusion models for SBI (Section 3) in a tutorial style, establishing a common conceptual and methodological baseline. Second, it provides a scoping review of diffusion model applications and adaptations in SBI (Section 4). Third, it elucidates, via conceptual exposition (Section 5) and tutorial-style empirical demonstration (Section 6), the specific design considerations needed to turn diffusion models into general-purpose SBI engines. Thus, our tutorial review offers both a synthesis of recent advances and practical insights to inform future work, and an overview of applications in this rapidly evolving area.

## 2 Problem Formulation

### 2.1 Simulation-Based Inference

The general simulation-based inference setup is as follows. We are given an *observation* $\mathbf{y}_{\text{obs}} \in \mathcal{Y}$, which is a realization of a potentially high-dimensional random structure $\mathcal{Y}$ (e.g., vectors, sets, graphs) of an arbitrary type (e.g., continuous, discrete, mixed). We assume that we have a *joint model* $p(\boldsymbol{\theta}, \mathbf{y})$ of the system that can simulate pairs $(\boldsymbol{\theta}, \mathbf{y})$ of *synthetic observations* $\mathbf{y}$ and the corresponding data-generating parameters, $\boldsymbol{\theta} \in \mathbb{R}^D$ with dimension $D$. The joint model itself can have both analytic and/or non-analytic components. It generally consists of a *prior* $p(\boldsymbol{\theta})$ that specifies the distribution of plausible parameters and a *data model* $p(\mathbf{y} \mid \boldsymbol{\theta})$ that specifies how observations can be generated from (latent) parameters. Accordingly, the joint probability model is defined as $p(\boldsymbol{\theta}, \mathbf{y}) = p(\mathbf{y} \mid \boldsymbol{\theta}) \, p(\boldsymbol{\theta})$.

In *likelihood-based* models, the data model $p(\mathbf{y} \mid \boldsymbol{\theta})$ is explicitly defined as a probability distribution (the likelihood) which can be analytically or numerically evaluated for any pair $(\boldsymbol{\theta}, \mathbf{y})$. In contrast, in *simulation-based* models, the data model is realized through a stochastic simulator $\texttt{Sim}(\boldsymbol{\theta}, \texttt{rng})$ that can generate synthetic observations given parameters and a random number generator $\texttt{rng}$ but cannot evaluate their likelihood (Table 1). Assuming that the model parameters $\boldsymbol{\theta}$ provide a useful summary of the real observations $\mathbf{y}_{\text{obs}}$, the goal of SBI is the same as that of statistical inference as a whole: extract the best estimate of the unknown parameters from the data along with a calibrated quantification of uncertainty for *any* model $p(\boldsymbol{\theta}, \mathbf{y})$.

The way in which SBI differs from established (i.e., "classical") statistical inference methods is therefore not in its goal, but in its approach of leveraging model simulations to learn which parameters are compatible with the real data. To motivate this approach, consider again the distinction between *likelihood-based* and *simulation-*

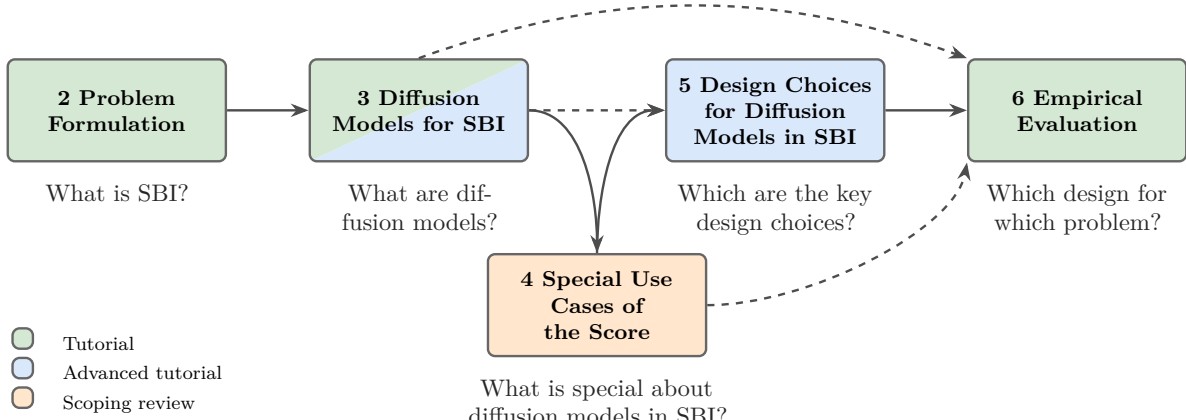

Figure 2: *Organization and reading paths through the paper.* Tutorial sections (green) establish foundational concepts and benchmarks, advanced sections (blue) develop technical methodology and notation, and the review section (orange) addresses SBI-specific adaptations of diffusion models. Solid arrows mark the main progression through the paper; dashed arrows indicate optional shortcuts and alternative entry points.

*based* models (Table 1). In the classical case, both the prior and the data model have closed forms (e.g., Gaussian), and gold-standard Monte Carlo methods are able to approximate the often intractable *posterior* implied by the famous Bayes' proportionality, $p(\boldsymbol{\theta} \mid \mathbf{y}_{\text{obs}}) \propto p(\mathbf{y}_{\text{obs}} \mid \boldsymbol{\theta})p(\boldsymbol{\theta})$. However, if we cannot evaluate either the prior or the likelihood (e.g., because only a sampling algorithm is specified, or a multidimensional integral involved), estimating the posterior becomes *doubly intractable*, hence a computational nightmare.

SBI is typically motivated as a solution to problems in settings, where all we can do is to simulate the joint model and use the simulations in a smart way to estimate $p(\boldsymbol{\theta} \mid \mathbf{y}_{\text{obs}})$ or $p(\mathbf{y}_{\text{obs}} \mid \boldsymbol{\theta})$. The approach was first introduced by (Diggle & Gratton, 1984) who used kernel density estimation (KDE) for approximating intractable likelihoods. However, SBI is applicable in settings where the likelihood is tractable and where it is not, and hence can be viewed as a general solution to Bayesian computation. In the simplest case, we collect model simulations into a labeled dataset, $\mathcal{D}_{\text{sim}} = \{\mathbf{y}^{(s)}, \boldsymbol{\theta}^{(s)}\}_{s=1}^{S}$, along with the unlabeled dataset of real observations, $\mathcal{D}_{\text{obs}} = \{\mathbf{y}_{\text{obs}}^{(r)}\}_{r=1}^{R}$. A key realization in modern SBI was that one can treat the labeled simulations as *training data* for generative networks, such as diffusion models—the workhorse of this tutorial review. Here, SBI can produce flexible parameter or data generators that can estimate or mimic practically any parametric model. When inferring parameters from real observations or emulating the data model, these generators need to solve a "Sim2Real" problem (Elsemüller et al., 2025), which can create extrapolation biases (Frazier et al., 2024) and calls for a principled Bayesian workflow (LI et al., 2026).

## 2.2 Inference Scenarios and Considerations

This section introduces a non-exhaustive taxonomy of common scenarios in SBI. We summarize recurring choices and trade-offs, such as inference targets, amortization, simulation budgets, and dimensionality, that are useful for situating existing methods and applications.

**Inference targets** Typical inference targets in SBI are the prior (Daras et al., 2024), likelihood (Sharrock et al., 2024), posterior (Wildberger et al., 2023), or joint distributions (Gloeckler et al., 2024), which can be approximated by diffusion models (Table 3). Each inference target comes with its own trade-offs, depending on the structure of the problem and the dimensionality of $\mathbf{y}$ and $\boldsymbol{\theta}$. Learning the posterior directly offers the most efficient inference, but may require re-training when model components change (e.g., the prior), unless some form of inference-time adaptation is employed (Yang et al., 2026). In contrast, learning the likelihood decouples model training from prior specification. This enables flexible reuse, for instance, in partial pooling or regression models where parameters $\boldsymbol{\theta}$ can be functions of other variables. However, posterior inference with learned likelihoods requires an additional sampling stage and may not be feasible for estimating many

Table 1: Comparison of likelihood-based and simulation-based models from a Bayesian perspective, which can both be estimated with diffusion models.

| | Likelihood-Based | Simulation-Based |
|---|---|---|
| **Joint model** $p(\boldsymbol{\theta}, \mathbf{y})$ | $\boldsymbol{\theta} \sim p(\boldsymbol{\theta}), \quad \mathbf{y} \sim p(\mathbf{y} \mid \boldsymbol{\theta})$ | $\boldsymbol{\theta} \sim p(\boldsymbol{\theta}), \quad \mathbf{y} = \texttt{Sim}(\boldsymbol{\theta}, \texttt{rng})$ |
| **Prior** $p(\boldsymbol{\theta})$ | can be sampled and evaluated | can be sampled and *optionally* evaluated |
| **Data model** $p(\mathbf{y} \mid \boldsymbol{\theta})$ | can be sampled and evaluated | can be sampled but *not* evaluated |
| **Posterior** $p(\boldsymbol{\theta} \mid \mathbf{y})$ | (usually) intractable | doubly intractable |

posteriors. We discuss learning the joint in Section 4.3.3; for learning the prior in image spaces, we refer to the literature on inverse problems (Daras et al., 2024).

**Amortized vs. direct inference** In *amortized inference*, the diffusion model learns a functional $q(\boldsymbol{\theta} \mid \mathbf{y})$ which—barring any simulation gaps (Schmitt et al., 2023; Frazier et al., 2024)—remains valid for any observation $\mathbf{y}_{\text{obs}}$. This approach offloads computation to an upfront training phase and rewards users with instant target estimation over the entire set of observations $\mathcal{D}_{\text{obs}}$. In contrast, *direct inference* repeats certain computational steps for each observation from scratch. There are two main direct approaches. First, one may learn a likelihood estimator $q(\mathbf{y} \mid \boldsymbol{\theta})$ that can be used in tandem with established Markov chain Monte Carlo (MCMC) (Homan & Gelman, 2014) samplers for downstream posterior inference. However, even though this likelihood estimator may be trained to remain valid for all pairs $(\boldsymbol{\theta}, \mathbf{y})$, posterior inference is *not* amortized due to the reliance on MCMC for eventually approximating the posterior. Alternatively, one may employ a sequential training scheme that specializes the posterior estimator for a particular observation $\mathbf{y}_{\text{obs}}$ (Sharrock et al., 2024). In this case, the approximator learns a tailored function $q(\boldsymbol{\theta} \mid \mathbf{Y} = \mathbf{y}_{\text{obs}})$ that can generate arbitrarily many posterior draws for $\mathbf{y}_{\text{obs}}$ but needs to be re-trained for each new observation.

**Simulation and real data budgets** The simulation and real data budgets are another key consideration in SBI (Table 2). The simulation budget depends on the "computational price" it costs to create a pair $(\boldsymbol{\theta}, \mathbf{y})$. In some fields, such as cognitive science, simulations are relatively inexpensive, allowing for a potentially infinite stream of simulations for online training (Radev et al., 2020; Fengler et al., 2021). These fields are well-positioned to benefit from high-throughput SBI (von Krause et al., 2022). In other fields, such as fluid dynamics, producing a high-fidelity pair $(\boldsymbol{\theta}, \mathbf{y})$ can take multiple days (Tumuklu & Hanquist, 2023), rendering simulation-based training hardly feasible. These domains represent fruitful avenues for future high-impact research, with recent work exploring diffusion models for multi-fidelity parameter estimation (Tatsuoka et al., 2025).

The availability of real data has different constraints than that of simulated data and cannot be increased in general without significant costs. However, the amount of simulated and real data can help decide whether to use an amortized or a non-amortized estimator when many posteriors need to be estimated. Even for a small number of real observations, amortized inference may still be the preferable option, since it serendipitously unlocks powerful diagnostics that require estimating hundreds or thousands of posteriors (e.g., simulation-based calibration, Talts et al., 2018; Yao & Domke, 2023; Modrák et al., 2025). Recent SBI research has also considered *semi-supervised* approaches that learn from both simulated and real data (Huang et al., 2023; Swierc et al., 2024; Elsemüller et al., 2025; Mishra et al., 2026), factoring in the real data budget as a source of training signal.

**Dimensionality of parameters and observations** The dimensionality of the model parameters $\boldsymbol{\theta}$ and the observations $\mathbf{y}$ are further key aspects that, together with the simulation budget, form eight problem archetypes (Table 2). Low-dimensional models typically serve as toy examples for evaluating SBI algorithms (e.g., Gaussian mixtures, two moons). Indeed, the SBI benchmarking suite by Lueckmann et al. (2021) features ten low-dimensional synthetic toy models (e.g., the maximum dimension of the observation $\mathbf{y}$ is 100). In contrast to synthetic benchmarks, most SBI applications in science come with high-dimensional observations and low-dimensional parameters. This is reflective of the nature of most scientific inquiries:

Table 2: *Example simulation models.* Featuring different simulation budgets available, dimensionalities of parameters ($\boldsymbol{\theta}$) and observations ($\mathbf{y}$). Setting arbitrary cutoffs, we consider parameter and data spaces high-dimensional if $\dim(\boldsymbol{\theta}) > 30$ and $\dim(\mathbf{y}) > 10,000$, respectively.

| Simulation budget | $\dim(\boldsymbol{\theta})$ | $\dim(\mathbf{y})$ | Example model family |
|---|---|---|---|
| High
Low | Low | Low | Current SBI benchmarks (Lueckmann et al., 2021)
*Quijote N*-body simulations (Delaunoy et al., 2024) |
| High
Low | Low | High | Single-molecule experiments (Dingeldein et al., 2025)
Galaxy simulations (Zhou et al., 2025) |
| High
Low | High | Low | 2D Darcy flow (Parno et al., 2016)
Groundwater (Hunt et al., 2020) |
| High
Low | High | High | Bayesian neural networks (BNNs; Izmailov et al., 2021)
Ultrasound imaging (Orozco et al., 2025) |

our hypotheses live and die by the sword of some interpretable summaries (e.g., the location and size of an exoplanet) of less interpretable observations (e.g., high-resolution images).

A typical high-dimensional target space arises when image pixels are treated as hidden parameters that need to be estimated from a corrupted image (e.g., Bayesian denoising) or another modality (Orozco et al., 2025). These target spaces are the natural habitat of diffusion models, which first rose to prominence as powerful generative models for images (Sohl-Dickstein et al., 2015; Song & Ermon, 2019; Ho et al., 2020). Within SBI, however, the focus is not on generating visually pleasing samples but on maximizing the *statistical accuracy of the learned distribution* (Frazier et al., 2024). This emphasis has driven adaptations to model architectures, training objectives, and inference schedules to harmonize with the demands of SBI tasks (Geffner et al., 2023; Wildberger et al., 2023; Holzschuh & Thuerey, 2024; Dirmeier & Mira, 2025), yielding measurable gains beyond those achievable with image quality-oriented objectives alone.

**Evaluation criteria**   A final consideration in our non-exhaustive list is the choice of metric for evaluating diffusion models in SBI. Fortunately, since most SBI methods are framed in a Bayesian setting, we can tap into the large pool of general, estimator-agnostic metrics put forward in the Bayesian literature (Bürkner et al., 2023) and recommended as part of principled Bayesian workflows (Gelman et al., 2020). In general, the Pareto front in SBI moves along two dimensions: *efficiency* and *accuracy*. Efficiency can be operationalized via wall-clock time $T$ (or computational complexity) and/or algorithmic complexity necessary for obtaining $L$ samples from one posterior given a budget of $B$ simulations:

$$T_{\text{total}}(B, L) = T_{\text{sim}}(B) + T_{\text{train}}(B) + T_{\text{inference}}(L). \tag{1}$$

For non-amortized (sequential) methods, inference costs $R \cdot T_{\text{inference}}(L)$ grow quickly with increasing number of posteriors $R$ due to the need to re-train the diffusion estimator for each observation, $R \cdot T_{\text{train}}(B)$, and potentially generate new simulations $R \cdot T_{\text{sim}}(B)$ (Greenberg et al., 2019; Griesemer et al., 2024).

Accuracy can be operationalized via a (strictly proper) distance $D$ between the true target distribution $p$ and the approximate distribution $q$ elicited by the diffusion model (as described shortly):

$$\text{Accuracy}(q) = \mathbb{E}_{\mathbf{y}_{\text{obs}} \sim p_{\text{obs}}(\mathbf{y})} \left[ D\big(p(\cdot \mid \mathbf{Y} = \mathbf{y}_{\text{obs}}) \parallel q(\cdot \mid \mathbf{Y} = \mathbf{y}_{\text{obs}})\big) \right], \tag{2}$$

where the expectation runs over the true data-generating distribution. This ideal metric is intractable, since the analytic posterior $p(\cdot \mid \mathbf{Y} = \mathbf{y}_{\text{obs}})$ and the full true data-generating distribution $p_{\text{obs}}(\mathbf{y})$ cannot be explicitly evaluated. Thus, proxy metrics that bypass evaluation, such as simulation-based calibration (SBC; Talts et al., 2018; Yao & Domke, 2023; Lemos et al., 2023; Modrák et al., 2025), are typically used and actively researched (e.g., Säilynoja et al., 2026; Bansal et al., 2025).

Finally, one may argue that posterior predictive checks (PPCs) and metrics such as the Expected Log Predictive Density (ELPD; Piironen & Vehtari, 2017) may also be used to evaluate diffusion models for SBI. This is

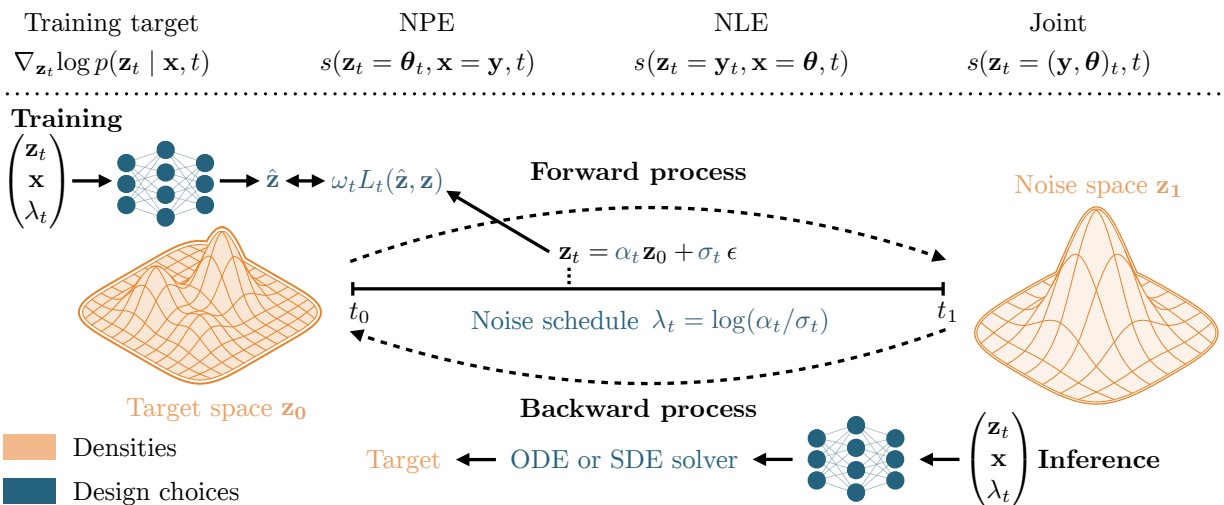

Figure 3: *Conceptual overview of diffusion models for simulation-based inference.* Diffusion models can solve canonical tasks in SBI, such as neural posterior estimation (NPE), neural likelihood estimation (NLE), or even joint estimation. They do so by recasting sampling from a complex target distribution into a denoising process that starts with a sample $\mathbf{z}_1$ from a simple noise distribution and progressively removes the noise to arrive at a target sample $\mathbf{z}_0$.

certainly valid *in silico*, where the Bayesian posterior guarantees optimality in posterior predictions (Aitchison, 1975). However, the relationship between metrics in $\boldsymbol{\theta}$-space and $\mathbf{y}$-space is not always straightforward (Scholz & Bürkner, 2025). Further, data-space checks with respect to $\mathbf{y}_{\text{obs}}$ inevitably conflate two sources of error: (1) errors due to an inaccurate posterior approximation (e.g., an underexpressive diffusion estimator), and (2) errors due to simulation gaps. Accordingly, good predictive performance in the *open world* can be achieved by *deviating from the target posterior under the potentially misspecified model* (Lai & Yao, 2024). Thus, even though PPCs are vital in practical Bayesian workflows (Gelman et al., 2020), they are ill-suited as a proxy for measuring posterior accuracy unless they are performed *in silico* or recast as generalized Bayesian inference by using proper scoring rules or divergence measures in place of the likelihood (Gao et al., 2023; Pacchiardi et al., 2024).

## 3 Diffusion Models for Simulation-Based Inference

**A note on notation** In this section, we let $\mathbf{z}$ denote the target variables to be inferred, and $\mathbf{x}$ the conditioning variables used for inference. In *posterior estimation*, we want to infer model parameters $\boldsymbol{\theta} \equiv \mathbf{z}$ conditioned on observations $\mathbf{y} \equiv \mathbf{x}$. In *likelihood estimation*, the roles are reversed: the goal is to estimate the likelihood of observations $\mathbf{y} \equiv \mathbf{z}$ given model parameters $\boldsymbol{\theta} \equiv \mathbf{x}$. Finally, in *joint estimation*, we target $p(\mathbf{z} = (\boldsymbol{\theta}, \mathbf{y}))$ directly, and $\mathbf{x} = \varnothing$. This notation allows us to discuss different inference targets using the same symbols, without redefining them each time (Figure 3).

**Probabilistic modeling with diffusion models** Diffusion models treat target generation as a denoising process (Sohl-Dickstein et al., 2015; Song & Ermon, 2019; Ho et al., 2020): Starting from pure noise (usually from a Gaussian), they gradually remove the noise until a realistic target instance remains. Diffusion—the gradual addition of noise until the targets are no longer recognizable—is the opposite process and plays a crucial role in the training of the denoising algorithm. Intuitively, diffusion models record a movie of the target dissolving into noise and then replay it in reverse to recreate the target. Starting with a real target instance $\mathbf{z}_0$, the "noising" process at time point $t$ (with initial time $t = 0$ and final time $t = 1$) performs iterations of the form

$$\mathbf{z}_{t+\Delta t} = \mathbf{z}_t + f_t \mathbf{z}_t \Delta t + g_t \sqrt{\Delta t}\, \boldsymbol{\epsilon}_t \quad \text{with} \quad \boldsymbol{\epsilon}_t \sim \mathcal{N}(\mathbf{0}, \mathbf{I}). \tag{3}$$

Here, the scalar $f_t \leq 0$ is a shrinkage term that optionally scales the target towards 0, and $g_t$ is the scalar noise standard deviation at time $t$. By making time steps infinitely small, $\Delta t \to 0$, (3) can be reformulated as a *stochastic differential equation* (SDE) (Song et al., 2021c)

$$d\mathbf{z}_t = f(t)\,\mathbf{z}_t\,dt + g(t)\,d\mathbf{W}_t, \tag{4}$$

which unlocks a large body of established theory. The first term is the *drift* coefficient, which realizes a deterministic change of the current state $\mathbf{z}_t$. The second term is the non-deterministic *diffusion* coefficient, with $d\mathbf{W}_t$ defined as a (multivariate) *Wiener process*, the SDE equivalent of standard normal noise. The specific form of (4) implies a simple analytic expression for the conditional density of states $p(\mathbf{z}_t \mid \mathbf{z}_0)$, that is, the distribution of outcomes when the SDE is started at $\mathbf{z}_0$ and executed to time $t$. Importantly, this closed-form expression eliminates the need to integrate the SDE until we reach $\mathbf{z}_t$, and allows us to evaluate and sample from $p(\mathbf{z}_t \mid \mathbf{z}_0)$ directly:

$$p(\mathbf{z}_t \mid \mathbf{z}_0) = \mathcal{N}(\alpha_t\,\mathbf{z}_0,\ \sigma_t^2\,\mathbf{I}) \quad\Longleftrightarrow\quad \mathbf{z}_t = \alpha_t\,\mathbf{z}_0 + \sigma_t\,\boldsymbol{\epsilon}_t \quad\text{with}\quad \boldsymbol{\epsilon}_t \sim \mathcal{N}(\mathbf{0},\mathbf{I}). \tag{5}$$

The starting point $\mathbf{z}_0 \sim p(\mathbf{z}_0)$ is a sample from the target-generating distribution (i.e., a member of the training set). This means that training tuples $(\mathbf{z}_0, \mathbf{z}_t, t)$ can be created efficiently, without explicit integration of the SDE (4). The functions $\alpha_t$ and $\sigma_t$ are known as the SDE's *noise schedule* and relate to the corresponding SDE coefficients via $f(t) = \alpha'_t/\alpha_t$ and $g(t)^2 = 2\sigma_t(\sigma'_t - \alpha'_t/\alpha_t\,\sigma_t)$. The choice of noise schedule (e.g., $\alpha_t := \sqrt{1-t}$ and $\sigma_t := \sqrt{t}$) is an important distinction between different diffusion models, as we discuss in Section 5. To generate a sample from the target distribution from noise, the SDE must be run in reverse. The reverse SDE can be solved backwards in time, essentially using negative time steps $d\tilde{t} < 0$, and has the same form as (4) with a different drift $\tilde{f}$:

$$d\mathbf{z}_{\tilde{t}} = \tilde{f}(\tilde{t}, \mathbf{z}_{\tilde{t}})\,d\tilde{t} + g(\tilde{t})\,d\mathbf{W}_{\tilde{t}}. \tag{6}$$

For ease of notation, we will also use $t$ for the reverse SDE. The SDE is usually solved numerically in a "one step towards $\mathbf{z}_0$, small step away" manner: The first term on the RHS reduces the noise, but the second term adds some fresh noise in every iteration. The new drift term can also be expressed analytically (Anderson, 1982):

$$\tilde{f}(t, \mathbf{z}_t) = f(t)\,\mathbf{z}_t - g(t)^2\,\nabla_{\mathbf{z}_t}\log p(\mathbf{z}_t). \tag{7}$$

The quantity $\nabla_{\mathbf{z}_t}\log p(\mathbf{z}_t)$ is called the *score* of the *marginal distribution* $p(\mathbf{z}_t)$. Its presence in (7) ensures that the denoising SDE indeed returns to the distribution of the starting point $p(\mathbf{z}_0)$ of the corresponding "noising" SDE.

Alternatively to a reverse SDE, we can also define a *deterministic* denoising process that follows the same marginal distribution $p(\mathbf{z}_t)$. This process is called the *probability flow*, which is an ordinary differential equation (ODE):

$$d\mathbf{z}_t = v(\mathbf{z}_t, t)\,dt \qquad\text{with}\qquad v(\mathbf{z}_t, t) = f(t)\,\mathbf{z}_t - \frac{1}{2}g(t)^2\,\nabla_{\mathbf{z}_t}\log p(\mathbf{z}_t). \tag{8}$$

This ODE is derived from the Fokker-Planck equation, where the factor $1/2$ arises from converting the variance of the Wiener process into a deterministic drift (for a detailed derivation, see Holderrieth & Erives, 2025). Since the ODE also depends on the score, the learning problems for (6) and (8) are closely related.

**Learning diffusion models** As (7) and (8) show, simulating the reverse process requires the score $\nabla_{\mathbf{z}_t}\log p(\mathbf{z}_t)$ of the marginal distribution. This poses a problem for training: $p(\mathbf{z}_t)$ depends on the unknown data distribution and is usually not available in closed form. The *conditional* score $\nabla_{\mathbf{z}_t}\log p(\mathbf{z}_t \mid \mathbf{z}_0)$, by contrast, is known analytically: for the forward process in (5), the conditional score is given by $\nabla_{\mathbf{z}_t}\log p(\mathbf{z}_t \mid \mathbf{z}_0) = -\boldsymbol{\epsilon}_t/\sigma_t$.

*Denoising score matching* (Vincent, 2011; Song & Ermon, 2019) exploits the property that regressing onto the conditional score $\nabla_{\mathbf{z}_t}\log p(\mathbf{z}_t \mid \mathbf{z}_0)$ actually recovers the marginal score $\nabla_{\mathbf{z}_t}\log p(\mathbf{z}_t)$. Hence, we can train an approximator (usually a deep neural network) $\hat{s}(\mathbf{z}_t, t)$ via regression onto conditional scores:

$$\hat{s} = \operatorname*{argmin}_{s}\ \mathbb{E}_{\mathbf{z}_0 \sim p(\mathbf{z}_0),\ t \sim \mathcal{U}(0,1),\ \mathbf{z}_t \sim p(\mathbf{z}_t \mid \mathbf{z}_0)}\left[\omega_t\,\big\|s(\mathbf{z}_t, t) - \nabla_{\mathbf{z}_t}\log p(\mathbf{z}_t)\big\|_2^2\right] \tag{9}$$

$$= \operatorname*{argmin}_{s}\ \mathbb{E}_{\mathbf{z}_0 \sim p(\mathbf{z}_0),\ t \sim \mathcal{U}(0,1),\ \boldsymbol{\epsilon}_t \sim \mathcal{N}(\mathbf{0},\mathbf{I})}\left[\omega_t\,\big\|s(\alpha_t\,\mathbf{z}_0 + \sigma_t\,\boldsymbol{\epsilon}_t,\ t) + \boldsymbol{\epsilon}_t/\sigma_t\big\|_2^2\right]. \tag{10}$$

The *weighting function* $\omega_t$ adjusts the importance of different time steps to maximize model accuracy and is therefore one of many design choices. Since the denoising score reduces to the simple expression $-\boldsymbol{\epsilon}_t/\sigma_t$, the neural network learns to estimate the noise $\hat{\boldsymbol{\epsilon}}_t \approx \boldsymbol{\epsilon}_t$ contained in the given instance $\mathbf{z}_t$ up to the scaling $-1/\sigma_t$. Equivalently, one can rearrange the loss such that $\hat{s}(\mathbf{z}_t, t)$ predicts the noise-free instance $\hat{\mathbf{z}}_0 = \mathbf{z}_t - \hat{\boldsymbol{\epsilon}}_t \approx \mathbf{z}_0$ instead. This *parameterization* of the network is a crucial design choice, which we discuss further in Section 5.4.

Beyond denoising score matching, some approaches attempt to estimate the score at the target $\mathbf{z}_0$ directly, without dependence on time. This typically requires computing the Jacobian trace of the score, but it enables direct regularization of the score itself to ease training (Osada et al., 2024), particularly in likelihood score estimation, where structural properties such as additivity, curvature, and mean-zero behavior are known (Jiang et al., 2025). In regions of low target density, however, score matching may lack sufficient signal to estimate score functions reliably (Song & Ermon, 2019).

A popular alternative is *flow matching* (Lipman et al., 2023; Liu et al., 2022). It trains a neural network $\hat{u}(\mathbf{z}_t, t) \approx v(\mathbf{z}_t, t)$ to approximate the vector field in the probability flow (8). In combination with the noise schedule $\alpha_t = 1 - t$ and $\sigma_t = t$ (with end time $T = 1$), this leads to the *flow-matching loss*:

$$\hat{u} = \operatorname*{argmin}_{u} \ \mathbb{E}_{\mathbf{z}_0, \mathbf{z}_t, t \sim p(\mathbf{z}_0, \mathbf{z}_t, t)} \left[ \omega_t \left\| u(\mathbf{z}_t, t) - v(\mathbf{z}_t, t) \right\|_2^2 \right] \tag{11}$$

$$= \operatorname*{argmin}_{u} \ \mathbb{E}_{\mathbf{z}_0 \sim p(\mathbf{z}_0), \, t \sim \mathcal{U}(0,1), \, \boldsymbol{\epsilon}_t \sim \mathcal{N}(\mathbf{0}, \mathbf{I})} \left[ \omega_t \left\| u\big((1-t)\,\mathbf{z}_0 + t\,\boldsymbol{\epsilon}_t, t\big) - (\boldsymbol{\epsilon}_t - \mathbf{z}_0) \right\|_2^2 \right], \tag{12}$$

where $\omega_t$ is a weighting function and the analytic expressions of $p(\mathbf{z}_t \mid \mathbf{z}_0)$, $v(\mathbf{z}_t, t)$, and $\mathbf{z}_t$ under the given noise schedule were inserted to arrive at the second form of the loss objective. The prediction $\hat{u}(\mathbf{z}_t, t)$ is therefore the average difference vector between pure noise $\boldsymbol{\epsilon}_t$ and noise-free data $\mathbf{z}_0$ of the (infinitely many) SDE trajectories that cross the location $\mathbf{z}_t$ at time $t$. Section 5.5 discusses the connections between score matching and flow matching in greater detail.

**Conditional diffusion models** In SBI, we typically want to control the generative process by conditioning on auxiliary information $\mathbf{x}$, such as observations in posterior estimation or parameters in likelihood estimation (i.e., surrogate modeling). The conditional objective can also be related back to the unconditional score (Batzolis et al., 2021; Li et al., 2024), which is an extension of (Vincent, 2011), and hence the corresponding objective is a conditional extension of the denoising score matching loss (9):

$$\hat{s} = \operatorname*{argmin}_{s} \ \mathbb{E}_{\mathbf{z}_0, \mathbf{x} \sim p(\mathbf{z}_0, \mathbf{x}), \, t \sim \mathcal{U}(0,1), \, \boldsymbol{\epsilon}_t \sim \mathcal{N}(\mathbf{0}, \mathbf{I})} \left[ \omega_t \left\| s(\mathbf{z}_t, \mathbf{x}, t) - \nabla_{\mathbf{z}_t} \log p(\mathbf{z}_t \mid \mathbf{x}) \right\|_2^2 \right] \tag{13}$$

$$= \operatorname*{argmin}_{s} \ \mathbb{E}_{\mathbf{z}_0, \mathbf{x} \sim p(\mathbf{z}_0, \mathbf{x}), \, t \sim \mathcal{U}(0,1), \, \boldsymbol{\epsilon}_t \sim \mathcal{N}(\mathbf{0}, \mathbf{I})} \left[ \omega_t \left\| s(\mathbf{z}_t, \mathbf{x}, t) - \nabla_{\mathbf{z}_t} \log p(\mathbf{z}_t \mid \mathbf{z}_0) \right\|_2^2 \right] \tag{14}$$

Intuitively, the score at time $t$ is determined by the target $\mathbf{z}_0$ alone, but the conditioning on $\mathbf{x}$ during the reverse process helps steer denoising toward the *subsets of the distribution* that match the observations or parameters of interest. Thus, conditioning restricts the space of plausible $\mathbf{z}_0$: without $\mathbf{x}$, the network must cover all possible clean signals, but with $\mathbf{x}$ it only needs to explain those consistent with the condition.

**Sampling and density estimation** Given a trained score model $\hat{s}(\mathbf{z}_t, \mathbf{x}, t)$, we can sample from the target distribution $p(\mathbf{z} \mid \mathbf{x})$ by integrating either the reverse SDE (6) or the reverse ODE (8) using the score model instead of the true score. This continuous formulation allows us to use state-of-the-art SDE and ODE solvers (Song et al., 2021c). A typical procedure starts with a draw from the base distribution $\mathbf{z}_1 \sim p_1(\mathbf{z}_1)$, which is usually a diagonal Gaussian distribution, and then solves the flow from $t = 1$ down to $t = 0$, producing a draw from the target distribution $\mathbf{z}_0 \sim p(\mathbf{z} \mid \mathbf{x})$:

$$\mathbf{z}_0 = \Phi_{1 \to 0}\big(\mathbf{z}_1, \mathbf{x}; \hat{s}\big), \qquad \mathbf{z}_1 \sim p_1(\mathbf{z}), \tag{15}$$

where $\Phi$ denotes the flow induced by either the reverse SDE (6) or the probability-flow ODE (8). The need to repeatedly evaluate the neural network within a numerical solver *for each random draw* is what makes inference with diffusion models slower than single-step methods (e.g., normalizing flows, Papamakarios et al., 2021). This motivates the use of distillation techniques (Section 5.6.2), which train a separate model to mimic

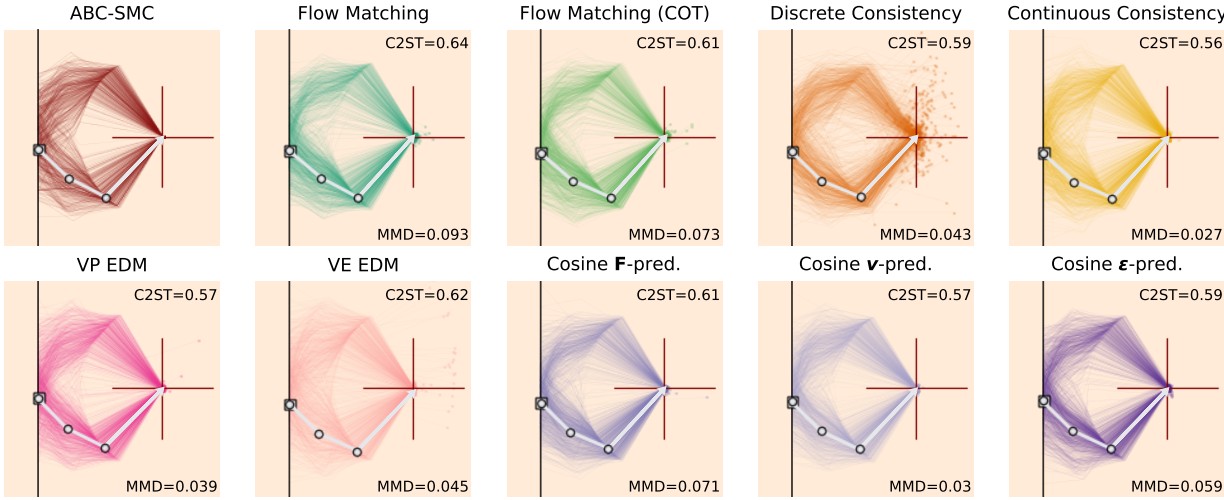

Figure 4: *Inverse kinematics toy example.* Each panel shows 1,000 approximate posterior samples representing possible arm configurations $\boldsymbol{\theta}$ given an end-effector position $\mathbf{y}_{\mathrm{obs}}$ (indicated by the red crosshair). The first panel shows the reference distribution estimated with ABC-SMC in `pyABC` (Schälte et al., 2022), while the remaining panels display samples from nine different diffusion models (see Section 5 for more details). The white segments indicate maximum a posteriori estimates. The quality of the posterior samples differs widely between the models, as also captured by differences in the maximum mean discrepancy (MMD) metric computed against the ABC posterior.

the multi-step diffusion process in a single forward pass, and have so far remained markedly underutilized in SBI.

In addition to sampling, we can evaluate (log) densities $p(\mathbf{z} \mid \mathbf{x})$ by integrating the probability flow ODE (8) from the target back to the base distribution and applying the *instantaneous change of variables formula* (Chen et al., 2018):

$$\log p(\mathbf{z}_0 \mid \mathbf{x}) = \log p(\mathbf{z}_1) + \int_0^1 \operatorname{div} v(\mathbf{z}_t, \mathbf{x}, t) \, \mathrm{d}t, \tag{16}$$

where the divergence of the velocity field, $\operatorname{div} v(\mathbf{z}_t, \mathbf{x}, t) = \operatorname{Tr}(\nabla_{\mathbf{z}_t} v(\mathbf{z}_t, \mathbf{x}, t))$, quantifies how much the flow expands or contracts at each time step. In order to compute the divergence efficiently, we can use an unbiased estimator known as the Skilling-Hutchinson trace estimator (Skilling, 1989; Hutchinson, 1989)

$$\operatorname{div} v(\mathbf{z}_t, \mathbf{x}, t) = \mathbb{E}_{\boldsymbol{\epsilon}_t \sim \mathcal{N}(\mathbf{0}, \mathbf{I})}[\boldsymbol{\epsilon}_t^T \nabla_{\mathbf{z}_t} v(\mathbf{z}_t, \mathbf{x}, t) \boldsymbol{\epsilon}_t], \tag{17}$$

which approximates the trace of the Jacobian of the velocity field. The ODE formulation is particularly useful for SBI applications that require likelihood or evidence estimation, such as Bayesian model comparison (Gunes et al., 2025).

**Example: Inverse Kinematics**   In the previous sections, we introduced diffusion models for conditional density estimation in SBI. We now turn to a concrete application and demonstrate how these concepts are used in practice. We set the stage with the *inverse kinematics* problem from Kruse et al. (2021), which provides an intuitive and visually accessible test case for SBI (Figure 4). The goal is to reconstruct the configuration of a planar robot arm from the observed position of its end effector. The unknown configuration is described by a four-dimensional parameter vector $\boldsymbol{\theta} \in \mathbb{R}^4$, where $\theta_1$ denotes the vertical offset of the arm's base and $\theta_2$, $\theta_3$, $\theta_4$ are the angles at the three joints. Given a configuration $\boldsymbol{\theta}$, a deterministic simulator returns the two-dimensional end-effector position $\mathbf{y}_{\mathrm{obs}} \in \mathbb{R}^2$.

In the notation introduced above, we identify the inference target with the parameters, $\mathbf{z} \equiv \boldsymbol{\theta}$, and the conditioning variable with the observation, $\mathbf{x} \equiv \mathbf{y}_{\mathrm{obs}}$. The prior $p(\boldsymbol{\theta})$ is a Gaussian distribution over plausible

arm configurations, and the data model is defined implicitly by the simulator as the distribution over end-effector positions induced by $\boldsymbol{\theta}$. The task is to estimate the posterior distribution $p(\boldsymbol{\theta} \mid \mathbf{y}_{\mathrm{obs}})$, which captures all arm configurations that are consistent with a given observed end-effector position $\mathbf{y}_{\mathrm{obs}}$. Owing to the design of the forward kinematics, this posterior is highly non-Gaussian and multi-modal, as distinct angle combinations of the joints can lead to the same endpoint.

We train diffusion models with different design choices, discussed in detail in Section 5, such as the noise schedule and the learning target (e.g., the score $s$ or the vector field $v$), using simulated pairs $(\boldsymbol{\theta}, \mathbf{y})$. Training proceeds by gradually diffusing samples of $\boldsymbol{\theta}$ and learning to reverse this process conditioned on the observation $\mathbf{y}_{\mathrm{obs}}$. At inference time, we obtain posterior draws by running the reverse SDE or probability-flow ODE from an initial noise draw toward $\boldsymbol{\theta}$, guided by the position $\mathbf{y}_{\mathrm{obs}} = (0, 1.5)$. For this illustration, we train nine diffusion-based posterior estimators under a limited simulation budget of $B = 10{,}000$, keeping network architecture and training hyperparameters fixed across models (Section A.1). For reference, we additionally approximate a high-fidelity posterior using ABC-SMC implemented in `pyABC` (Schälte et al., 2022) with a minimum acceptance threshold of $\epsilon_{\mathrm{ABC}} = 0.002$.

Under the same restricted simulation budget, the diffusion models exhibit noticeably different posterior quality (Figure 4), both visually and in terms of maximum mean discrepancy (MMD; Gretton et al., 2012) and classifier-two-sample-test (C2ST; Lopez-Paz & Oquab, 2017). All models recover the characteristic multi-modality of the inverse kinematics problem by producing distinct, plausible arm configurations that reach the same end-effector position. Yet, differences between the posteriors are clearly visible when visualizing posterior samples as arm configurations and are reflected quantitatively in higher MMD values (Figure 4). The observed variability highlights the need for a systematic analysis of diffusion model design decisions for SBI, which we introduce in Section 5 and evaluate in Section 6. Before turning to these design considerations, however, we first examine how learned score functions can be exploited beyond standard posterior or likelihood estimation, since it is precisely this flexibility at training and inference time that makes diffusion models particularly powerful for SBI.

## 4 Special Use Cases of the Score: Compositional and Guided Inference

Thus far, we have focused our attention on approximating a conditional score $\nabla_{\mathbf{z}_t} \log p(\mathbf{z}_t \mid \mathbf{x})$, along with methods to utilize the score for evaluating or sampling from an approximate distribution $q(\mathbf{z} \mid \mathbf{x}) \approx p(\mathbf{z} \mid \mathbf{x})$. In SBI, this covers common targets by choosing $(\mathbf{z}, \mathbf{x})$ appropriately, for instance, neural posterior estimation (NPE; $\mathbf{z} \equiv \boldsymbol{\theta}$, $\mathbf{x} \equiv \mathbf{y}$) and neural likelihood estimation (NLE; $\mathbf{z} \equiv \mathbf{y}$, $\mathbf{x} \equiv \boldsymbol{\theta}$) as introduced in Section 3. The following sections turn to *model specializations and inference-time use.* Three properties make score-based SBI unusually flexible compared to other approximators:

1. Scores can be *modified or guided during inference* to incorporate new information, constraints, or preferences without retraining.

2. Scores can be *additively combined* to represent arbitrary factorizations of joint or conditional distributions from simpler building blocks.

3. Scores can *encode arbitrary inductive biases*, realizing inference over complex random structures such as images, graphs, or time series.

Together, these properties provide unmatched flexibility in reusing trained approximators: we can *combine* separate components in a divide-and-conquer fashion, *adjust* them to incorporate new priors or constraints, and *adapt* to changing data (Figure 5)—all without retraining the score model.

In neural posterior estimation, we typically train a conditional score model $\hat{s}(\boldsymbol{\theta}_t, \mathbf{y}, t)$ to sample from an approximate posterior $q(\boldsymbol{\theta} \mid \mathbf{y})$. When priors, constraints, or objectives change, *guidance* offers a direct alternative to retraining: the learned score can be steered during inference by adding or reweighting prior and likelihood terms implied by the identity

$$\nabla_{\boldsymbol{\theta}_t} \log p(\boldsymbol{\theta}_t \mid \mathbf{y}) = \nabla_{\boldsymbol{\theta}_t} \log p(\boldsymbol{\theta}_t) + \nabla_{\boldsymbol{\theta}_t} \log p(\mathbf{y} \mid \boldsymbol{\theta}_t). \tag{18}$$

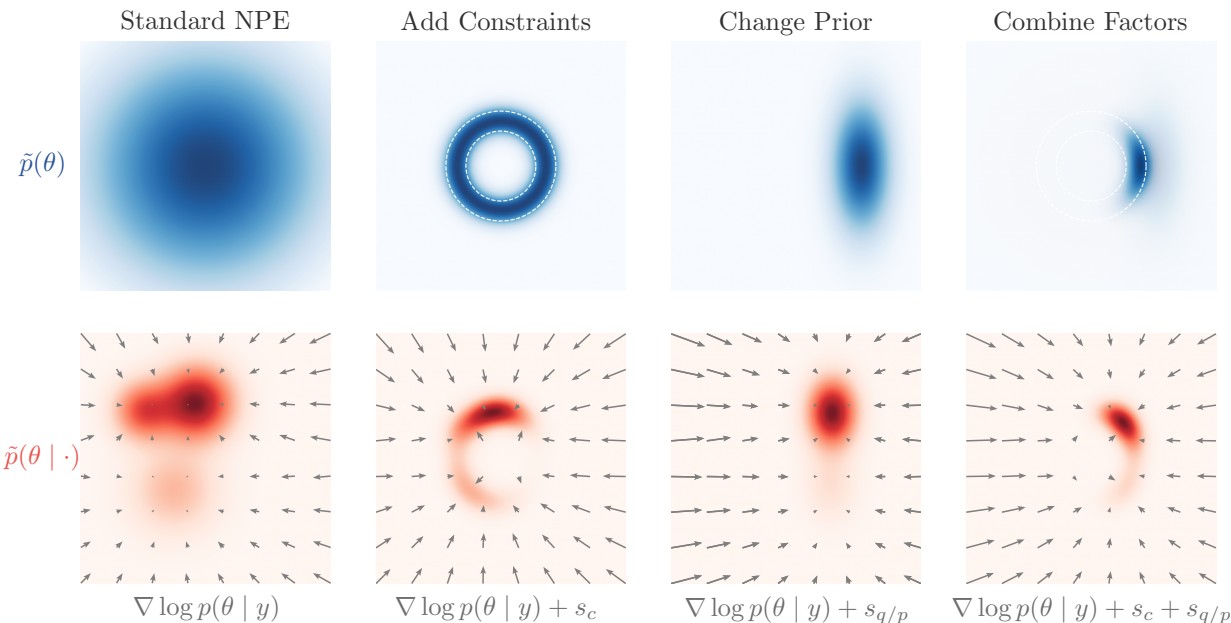

Figure 5: *Schematic guided score fields.* Panels show the resulting densities for (surrogate) priors $\tilde{p}(\theta)$ (*top*) and target posteriors $\tilde{p}(\theta \mid \cdot)$ (*bottom*). Arrows show the score used for inference. Columns differ only by the extra guidance term added to $\nabla \log p(\theta)$ (*left to right*): classifier-free (baseline), constraints $+ s_c$ (annulus), prior-adaptive $+ s_{q/p}$, and compositional $+ s_c + s_{q/p}$.

In the following subsections, we review recent developments that leverage this flexibility in SBI, organized into three themes: (i) adaptive inference, (ii) compositional estimation in structured observations, and (iii) estimation of special structured targets.

## 4.1 Adaptive Inference

### 4.1.1 Guiding the Reverse Process

Assuming a trained unconditional score estimator approximating $\nabla_{\boldsymbol{\theta}_t} \log p(\boldsymbol{\theta}_t)$ in (18), there are two main approaches to approximate the likelihood score $\nabla_{\boldsymbol{\theta}_t} \log p(\mathbf{y} \mid \boldsymbol{\theta}_t)$.

In conditional image generation, a widely used method known as *classifier guidance* (Dhariwal & Nichol, 2021) approximates the likelihood score via an auxiliary classifier that predicts class labels from noisy samples. Its gradient with respect to the input is then added to the reverse process, and the resulting samples are steered toward the target class-conditional distribution. Classifier guidance is rarely useful for SBI, where the conditions are typically continuous quantities rather than discrete labels.

A more flexible method is *classifier-free guidance* (Ho & Salimans, 2021). Instead of relying on an external classifier, the diffusion model is trained jointly in two randomly switching modes: conditional (with access to the conditioning variables, i.e., the posterior) and unconditional (with the conditioning masked out, i.e., the prior). The likelihood score can be estimated from the difference between the conditional and unconditional scores as

$$\nabla_{\boldsymbol{\theta}_t} \log p(\mathbf{y} \mid \boldsymbol{\theta}_t) \approx \hat{s}(\boldsymbol{\theta}_t, \mathbf{y}, t) - \hat{s}(\boldsymbol{\theta}_t, \varnothing, t), \tag{19}$$

where $\varnothing$ is realized by conditioning on a tensor of zeros or a learnable tensor with the same dimension as $\mathbf{y}$. This decomposition allows the prior and likelihood contributions to be modified separately. For example, one can define an adapted posterior score by combining the two outputs *post hoc* through interpolation during sampling

$$\nabla_{\boldsymbol{\theta}_t} \log \tilde{p}(\boldsymbol{\theta}_t \mid \mathbf{y}) \approx \alpha_1(\hat{s}(\boldsymbol{\theta}_t, \varnothing, t) + \beta_1) + \alpha_2 \left( \hat{s}(\boldsymbol{\theta}_t, \mathbf{y}, t) - \hat{s}(\boldsymbol{\theta}_t, \varnothing, t) + \beta_2 \right), \tag{20}$$

Table 3: Summary of papers employing or adapting diffusion models in simulation-based inference (SBI). Design choices are described in Section 5.

| | Paper | Use Case | Design Choice |
|---|---|---|---|
| **NPE & NLE** | Simons et al. (2023) | high-quality samples | VE linear, SDE, score |
| | Sharrock et al. (2024) | sequential correction in score space, high quality for single observation | VP/VE linear, ODE, score |
| | Yu & Liu (2025) | multiple prediction tasks with one model | VP/VE linear, ODE, score |
| | Liang & Wang (2025) | high quality samples & high dimensional | flow matching / consistency model |
| **NPE** | Geffner et al. (2023) | compositional score for complete pooling | VP linear, Langevin, score |
| | Wildberger et al. (2023) | high-dimensional & complex posteriors | flow matching (power-law) |
| | Andry (2023) | decomposing score for time varying prior/posterior & state space models | VP, SDE, $\epsilon$-prediction |
| | Pidstrigach et al. (2024) | infinite-dimensional score for non-parametric inference | VP linear, SDE, score |
| | Holzschuh & Thuerey (2024) | decomposing flow for high-quality samples with simulator feedback | flow matching |
| | Orozco et al. (2024) | high-quality samples | VE-EDM, SDE, **F**-prediction |
| | Fluri & Hofmann (2024) | high-quality samples | OT flow matching |
| | Schmitt et al. (2025) | single-step sampling | VE EDM, consistency model |
| | Dasgupta et al. (2025) | high-dimensional | VP linear, Langevin, score |
| | Gebhard et al. (2025) | high-quality samples | flow matching ($\sqrt[4]{t} \sim \mathcal{U}[0,1]$) |
| | Gloeckler et al. (2025) | compositional score for time series pooling | VP linear, SDE, score |
| | Chen et al. (2025b) | high-quality samples | VE EDM, ODE, **F**-prediction |
| | Zeng et al. (2025b) | high-dimensional | EDM, SDE, **F**-prediction |
| | Zeng et al. (2025a) | high-quality samples | sub-VP linear, ODE, score |
| | Dirmeier & Mira (2025) | causal network architecture | flow matching |
| | Cirakman et al. (2025) | multi-scale wavelet score composition for high-dimensional problems | VE EDM, SDE, **F**-prediction |
| | Raymond et al. (2025) | high-quality samples | flow matching |
| | Orsini et al. (2025a) | high-quality samples | flow matching (power-law) |
| | Zhao et al. (2025) | fast & high-quality samples | VE EDM, consistency model |
| | Orsini et al. (2025b) | high-quality samples | flow matching (power-law) |
| | Touron et al. (2025) | compositional score for complete pooling | VP linear, $\epsilon$-prediction |
| | Ruhlmann et al. (2025) | robust inference under model misspecification | flow matching |
| | Ko & Geffner (2025) | simulation efficiency using tractable score | VP linear, SDE, score |
| | Linhart et al. (2026) | compositional score for complete pooling | VP linear, $\epsilon$-prediction |
| | Yang et al. (2026) | score guidance for post hoc prior change | VE EDM, SDE, score |
| | Nautiyal et al. (2026) | high-quality samples | VP cosine/quadratic/linear, $\epsilon$-prediction |
| | Arruda et al. (2026) | compositional score for hierarchical models | VP cosine, SDE, $\boldsymbol{v}$-prediction |
| | Viterbo & Buck (2026) | high-quality samples | flow matching |
| | Villarreal et al. (2026) | high-quality samples | flow matching & VE linear, SDE, $\epsilon$-prediction |
| **NLE** | Haitsiukevich et al. (2024) | multiple prediction tasks with PDEs | VP linear/VE EDM, ODE, $\epsilon$- & **F**-prediction |
| | Shysheya et al. (2025) | forecasting PDE | VP linear, SDE, score |
| | Mudur et al. (2025) | high-dimensional | VP linear, SDE, $\epsilon$-prediction |
| | Legin et al. (2025) | learn complex noise model | VE linear, score |
| **Joint** | Gloeckler et al. (2024) | score decomposition to sample arbitrary conditionals & guidance for post hoc adaptation to new priors/likelihoods | VP/VE linear, SDE, score |
| | Tesso et al. (2025) | non-parametric spatial inference | VE linear, SDE, score |
| | Gunes et al. (2025) | model comparison using likelihood and inference with posterior | VE linear, SDE/ODE, score |
| | Jeong et al. (2025) | theoretical guarantee on consistency | flow matching |
| | Charles et al. (2025) | hierarchical models | flow matching |

*Includes papers whose first public version appeared before 2026.*

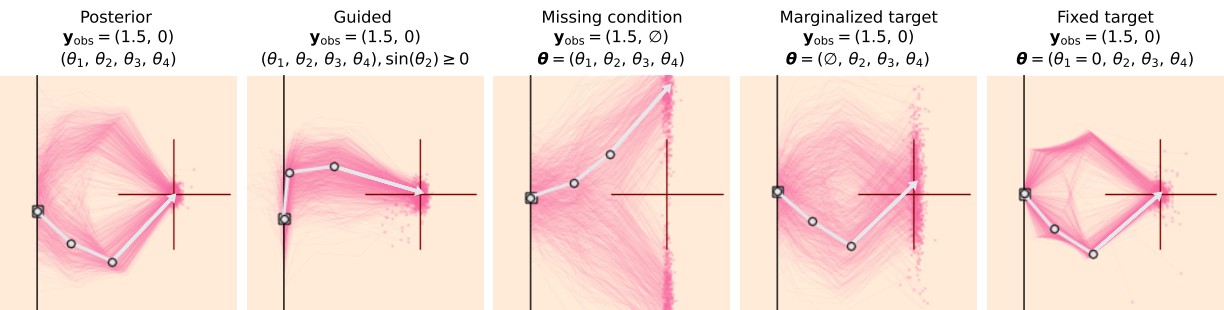

Figure 6: *Inverse kinematics toy example extended.* Each panel shows 1,000 approximate posterior samples from the same diffusion model representing possible arm configurations $\boldsymbol{\theta}$ given an end-effector position $\mathbf{y}_{\text{obs}}$ (indicated by the red crosshair). The first panel shows the usual posterior, while the other panels show inference-time adaptations: guidance to one of the two modes, masking conditions, marginalizing out targets, or fixing targets to a specific value.

where the scales $(\alpha_1, \alpha_2)$ and optional shifts $(\beta_1, \beta_2)$ act as guidance dials, allowing tempering and shifting of the prior and likelihood contributions (Gloeckler et al., 2024). In theory, these dials enable tempered-like posterior sampling and incorporation of side information, since the user can control how strongly the model adheres to the conditions. Such modifications can be used to perform sensitivity analysis with respect to prior and likelihood choices (Elsemüller et al., 2024), or to reweigh parts of the likelihood in order to emphasize specific subsets of the data, such as recent or high-quality observations.

Notably, guidance is not limited to labels or direct conditioning variables (Figure 5). Recent works have shown that diffusion models can be guided by almost arbitrary functions of the generated sample (Bansal et al., 2023) or even surrogates of their scores in cases of non-differentiability (Shen et al., 2025). This makes it possible to design and impose custom constraints during the reverse process without retraining the model. For instance, one can require that generated samples satisfy inequality constraints, lie within prescribed intervals, or fix the target value (Figure 6). Consider imposing $K$ constraints of the form $c_k(\mathbf{z}_t) \leq 0$. One can incorporate them directly into the score estimate by adding a correction term derived from the gradient of a soft penalty. A possible formulation is

$$\nabla_{\mathbf{z}_t} \log p(\mathbf{z}_t \mid c_1, \ldots, c_K) \approx \hat{s}(\mathbf{z}_t, t) + \nabla_{\mathbf{z}_t} \sum_{k=1}^{K} \log \text{sigmoid}\left(-h(t)\, c_k(\mathbf{z}_t)\right), \tag{21}$$

where sigmoid denotes the logistic sigmoid and $h(t)$ is a scaling function that diverges as $t \to 0$, typically linked to the noise schedule, for instance, $h(t) = \alpha_t^2/\sigma_t^2$. This ensures that as the process approaches the target sample, the constraints are enforced with increasing strength. Similarly, the posterior can be conditioned on observation intervals or other context variables (Gloeckler et al., 2024).

However, the flexibility that makes guidance attractive also carries a cost: any guidance biases the reverse diffusion process (Chidambaram et al., 2024). Changing the score means that one implicitly assumes that the marginal density $p(\mathbf{z}_t \mid \mathbf{z}_0)$ of the reverse diffusion process changes accordingly. Depending on the type of guidance, this can make the new score unstable and require Langevin, weighted or MCMC-based samplers to correct for the error (Geffner et al., 2023; Skreta et al., 2025; Sjöberg et al., 2026). However, the need for correction depends on the application and the guidance strength. Moreover, Vuong et al. (2025) argue that under a Wasserstein-gradient-flow interpretation, diffusion sampling can remain effective even when the neural vector field is not an exact score, since correct marginal transport does not require exact reverse-time path equivalence. This, however, does not remove guidance bias; a guidance term still changes the effective vector field, and the resulting marginal flow is reliable only insofar as this altered field remains compatible with the intended density evolution. Hence, applying guidance in the context of SBI necessitates careful checking of posterior calibration (Modrák et al., 2025) to quantify potential bias induced by guidance.

Guidance provides a general mechanism for steering diffusion models during sampling, ranging from simple conditional adjustments to constraint-based steering, enabling the enforcement of domain-specific requirements without retraining. The design space is broad, and additional guidance schemes can be tailored to specialized applications.

### 4.1.2 Incorporating Simulator Feedback

So far, guidance signals have only been injected at inference time: the total score used in the reverse process is steered toward desirable properties without retraining the model. A complementary strategy is to integrate feedback *during training.* In the flow-matching setting of Holzschuh & Thuerey (2024), this leads to a two-phase procedure.

In the first phase, a primary flow is trained to estimate a vector field $\hat{\mathbf{u}}$ that transports noise to parameters via the reverse probability flow (8). In addition to conditioning on $\mathbf{y}$, a form of self-conditioning (Chen et al., 2023a) is introduced to improve the posterior approximation. At each time $t < 0.5$, where we are closer to the clean target in the diffusion process, the model makes a *one-step prediction* of the clean state $\tilde{\boldsymbol{\theta}}_0$ from the current noisy state $\boldsymbol{\theta}_t$ and then self-conditions the velocity field on that prediction:

$$\hat{\mathbf{u}} = u\big([\boldsymbol{\theta}_t, \tilde{\boldsymbol{\theta}}_0], \mathbf{y}, t\big), \quad \tilde{\boldsymbol{\theta}}_0 = \boldsymbol{\theta}_t + t\, u\big([\boldsymbol{\theta}_t, \mathbf{0}], \mathbf{y}, t\big). \tag{22}$$

In the second phase, a lightweight auxiliary *control flow* $u^C(\cdot)$ (about 10% of the primary flow's parameters) is trained to correct the velocity prediction of the primary flow based on some control signal $c(\tilde{\boldsymbol{\theta}}_0, \mathbf{y})$, for instance, the residual between observation and posterior-predicted observations. The resulting adapted vector field $\tilde{\mathbf{u}}$ is obtained by correcting the pretrained flow with a control-aware vector field

$$\tilde{\mathbf{u}} = \hat{\mathbf{u}} + \hat{\mathbf{u}}^C, \quad \hat{\mathbf{u}}^C = u^C\big(\hat{\mathbf{u}}, \boldsymbol{c}, t\big). \tag{23}$$

The control flow is trained only for small diffusion times (empirically $t < 0.2$) to focus corrections where denoising is most sensitive and to keep the additional computational overhead limited.

For diffusion models, in cases where the score $\nabla_{\boldsymbol{\theta}_t} \log p(\boldsymbol{\theta}_t \mid \mathbf{y})$ is tractable, incorporating the score in the objective to provide an unbiased estimator of the marginal score leads to higher simulation efficiency (Ko & Geffner, 2025). Beyond SBI, related work has also explored incorporating feedback into flow-based models by differentiating through the flow and directly optimizing a loss to find the optimal source noise (Ben-Hamu et al., 2024), or by guiding diffusion models to satisfy a terminal cost (Pandey et al., 2025). In such approaches, the loss could be likelihood-based, providing an alternative form of simulator feedback that modifies the generative process via optimization rather than through additional observation-specific networks.

### 4.1.3 Inference-Time Prior Adaptation

In many applications, priors can change after the model has been trained. For instance, one can shift from population-level to subject-specific priors, or perform Bayesian updating, which promotes yesterday's posterior to today's prior. Building on the guidance view above, prior adaptation handles the case where only the prior changes while the observation model remains the same. Instead of retraining, the learned posterior score approximator can be reused by adding an inference-time correction that propagates the prior ratio through diffusion time (Yang et al., 2026).

Concretely, given a model trained with prior $p(\boldsymbol{\theta})$, we want samples from $\tilde{p}(\boldsymbol{\theta} \mid \mathbf{y}_{\mathrm{obs}}) \propto p(\mathbf{y}_{\mathrm{obs}} \mid \boldsymbol{\theta})\tilde{p}(\boldsymbol{\theta})$ given the new prior $\tilde{p}(\boldsymbol{\theta})$ by adjusting the score during sampling. The adjusted score can be written as

$$\nabla_{\boldsymbol{\theta}_t} \tilde{p}(\boldsymbol{\theta}_t \mid \mathbf{y}_{\mathrm{obs}}) = \nabla_{\boldsymbol{\theta}_t} \log p(\boldsymbol{\theta}_t \mid \mathbf{y}_{\mathrm{obs}}) + \nabla_{\boldsymbol{\theta}_t} \log \int \frac{\tilde{p}(\boldsymbol{\theta}_0)}{p(\boldsymbol{\theta}_0)} p(\boldsymbol{\theta}_0 \mid \boldsymbol{\theta}_t, \mathbf{y}_{\mathrm{obs}})\, \mathrm{d}\boldsymbol{\theta}_0 \tag{24}$$

$$= \nabla_{\boldsymbol{\theta}_t} \log p(\boldsymbol{\theta}_t \mid \mathbf{y}_{\mathrm{obs}}) + \nabla_{\boldsymbol{\theta}_t} \log \mathbb{E}_{p(\boldsymbol{\theta}_0 \mid \boldsymbol{\theta}_t, \mathbf{y}_{\mathrm{obs}})} \left[ \frac{\tilde{p}(\boldsymbol{\theta}_0)}{p(\boldsymbol{\theta}_0)} \right]. \tag{25}$$

The first term is the original posterior score under the training prior, while the second term serves as a guidance correction that propagates the effect of the new prior at diffusion time $t$. To evaluate the additional

term in practice, Yang et al. (2026) propose to approximate the reverse transition kernel $p(\boldsymbol{\theta}_0 \mid \boldsymbol{\theta}_t, \mathbf{y}_{\mathrm{obs}})$ and the prior ratio using a generalized mixture of Gaussians, which yields a tractable closed-form correction.

Furthermore, prior adaptation could unlock continual Bayesian updating by using the posterior from one step as the new prior in the next, thereby chaining inference tasks without retraining the diffusion estimator. It also connects to validation methods for neural posterior estimators. For instance, simulation-based calibration (SBC) (Cook et al., 2006; Talts et al., 2018; Säilynoja et al., 2022; Lemos et al., 2023; Modrák et al., 2025) provides a default framework to assess calibration properties (e.g., over- or underdispersion) of the estimator *in silico*, and posterior SBC extends this idea by conditioning on a fixed observation $\mathbf{y}_{\mathrm{obs}}$ (Säilynoja et al., 2026). This would usually require training a model to handle an augmented posterior $p(\boldsymbol{\theta} \mid \mathbf{y}, \mathbf{y}_{\mathrm{obs}})$. Under the prior adaptation view, however, this is either equivalent to changing the prior to $p(\boldsymbol{\theta} \mid \mathbf{y}_{\mathrm{obs}})$ or to composing two observations, $\mathbf{y}$ and $\mathbf{y}_{\mathrm{obs}}$. Both variants can be implemented with diffusion models without retraining on both observations simultaneously. allowing posterior SBC to be efficiently checked.

## 4.2 Aggregating Scores for Structured Observations

A key property of training a diffusion model to estimate a score is that product factorizations in Bayesian models become sums in score space. For instance, as we have seen earlier, the posterior score decomposes into the sum of the likelihood and prior scores. This additive structure makes score-based models well-suited for estimating *structured* Bayesian models in which observations exhibit exchangeability, temporal ordering, spatial dependence, or hierarchies.

Such structures arise widely across scientific domains. Examples include: repeated measurements in cognitive psychology (Lee, 2011); single-cell or multi-modal biological data with large i.i.d. cohorts (Kilian et al., 2024); dynamical systems and ODE/SDE models in epidemiology or biology (Radev et al., 2021); multi-level or grouped observations in pharmacokinetics (Wakefield, 1996); or sequential updating in climate modeling (Houtekamer & Zhang, 2016), to name just a few.

Score decomposition converts these probabilistic symmetries into additive components that can be estimated independently and composed at inference time, allowing training on partial simulations rather than full-model simulation. As a result, the choice of a Bayesian model determines whether parameters are inferred for each observation independently or shared across many observations. Thus, common pooling assumptions in Bayesian analysis (Gelman et al., 2013) shape how diffusion models can exploit factorizations to reduce simulation and training costs dramatically.

### 4.2.1 No-Pooling Models

A *no-pooling* model assigns independent parameter sets $\boldsymbol{\theta}$ to each observation $\mathbf{y}_{\mathrm{obs}}$, resulting in an observation-specific posterior distribution

$$p(\boldsymbol{\theta} \mid \mathbf{y}_{\mathrm{obs}}) \propto p(\boldsymbol{\theta})\, p(\mathbf{y}_{\mathrm{obs}} \mid \boldsymbol{\theta}). \tag{26}$$

Consequently, no information is shared between observations, and each posterior estimate relies solely on its corresponding individual observation. Most diffusion model adaptations in SBI focus on the no-pooling setting (Table 3), which has emerged as the predominant target of SBI applications (Zammit-Mangion et al., 2024) and benchmarks (Lueckmann et al., 2021). When the number of observations $R$ is large, a single diffusion model can be trained to estimate all posteriors for $\{\mathbf{y}_{\mathrm{obs}}^{(r)}\}_{r=1}^{R}$ without further re-training (i.e., *amortized inference*, see Section 2.1).

**Unordered observations** So far, we have not paid much attention to the structure of each $\mathbf{y}$. Many Bayesian models assume exchangeable observations $\mathbf{y}_{1:N} := \{\mathbf{y}_1, \mathbf{y}_2, \ldots, \mathbf{y}_N\} \equiv \mathbf{y}_{\mathrm{obs}}$, for instance, due to repeated independent experiments, leading to a factorizable model (Gelman et al., 1995):

$$p(\boldsymbol{\theta}, \mathbf{y}_{1:N}) = p(\boldsymbol{\theta}) \prod_{n=1}^{N} p(\mathbf{y}_n \mid \boldsymbol{\theta}). \tag{27}$$

In NPE, one typically uses a permutation-invariant network, such as `DeepSet` (Zaheer et al., 2017) or `SetTransformer` (Lee et al., 2019), to encode this exchangeability (Radev et al., 2020). However, this

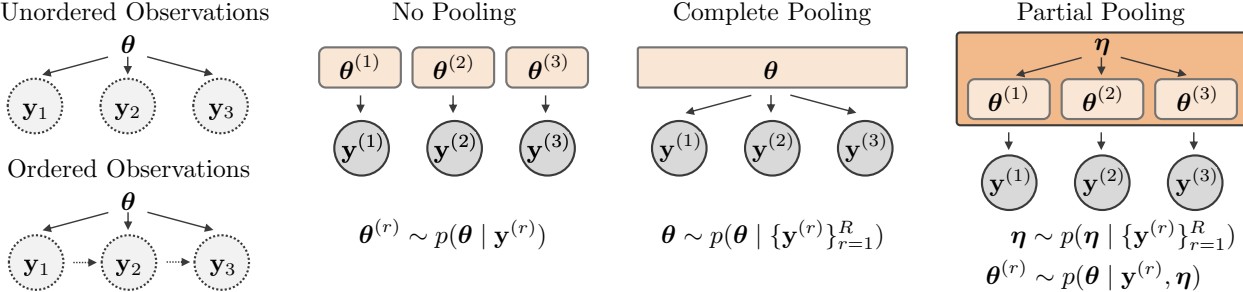

Figure 7: *Different probabilistic symmetries amenable to compositional score aggregation.* Score aggregation allows training on partial rather than full model simulations, leading to substantial gains in simulation efficiency. Scores can be combined across multiple unordered or ordered observations in no-pooling settings, or across multiple sets of observations in complete or partial pooling settings.

approach requires $N$ simulator calls to generate a single training instance. With score-based models, simulation and training efficiency can be increased by targeting the *compositional posterior*

$$p(\boldsymbol{\theta} \mid \mathbf{y}_{1:N}) \propto p(\boldsymbol{\theta})^{1-N} \prod_{n=1}^{N} p(\boldsymbol{\theta} \mid \mathbf{y}_n), \tag{28}$$

which follows directly from Bayes' rule. Transforming this expression to score-space, Geffner et al. (2023) introduced compositional score matching (CSM) and showed that the time-dependent score of the total posterior can be expressed as a linear combination of the prior and $N$ posterior factors:

$$\nabla_{\boldsymbol{\theta}_t} \log p(\boldsymbol{\theta}_t \mid \mathbf{y}_{1:N}) \approx (1-N)(1-t) \nabla_{\boldsymbol{\theta}} \log p(\boldsymbol{\theta})\big|_{\boldsymbol{\theta}=\boldsymbol{\theta}_t} + \sum_{n=1}^{N} \nabla_{\boldsymbol{\theta}_t} \log p(\boldsymbol{\theta}_t \mid \mathbf{y}_n), \tag{29}$$

where $\nabla_{\boldsymbol{\theta}} \log p(\boldsymbol{\theta})\big|_{\boldsymbol{\theta}=\boldsymbol{\theta}_t}$ is the score of the prior evaluated at $\boldsymbol{\theta}_t$. CSM's primary goal is to improve the *simulation efficiency* of SBI by requiring simulations only for the individual components during training, which can then be aggregated at inference to recover the score of the full model. This allows us to train a single score network on samples $(\boldsymbol{\theta}, \mathbf{y}_n) \sim p(\boldsymbol{\theta})p(\mathbf{y}_n \mid \boldsymbol{\theta})$ and saves $N-1$ simulations of $p(\mathbf{y}_n \mid \boldsymbol{\theta})$ per training instance. After training, posterior draws are obtained by sampling from the new base distribution $p(\boldsymbol{\theta}_1) = \mathcal{N}(\mathbf{0}, \frac{1}{N}\mathbf{I})$ and computing the compositional score (29) for each time point $t$ of the reverse diffusion process.

However, compositional estimation does not necessarily follow the true score of the marginal distribution $p(\boldsymbol{\theta}_t \mid \mathbf{y}_{1:N})$ induced by composition and is exact only for $t=0$. Thus, naive composition is prone to error accumulation, making (29) unstable and necessitating careful scheduling or error damping by using the base distribution $p(\boldsymbol{\theta}_1) = \mathcal{N}(\mathbf{0}, I)$ and adaptive SDE solvers (Arruda et al., 2026). Moreover, instead of composing over all $N$ elements, one can also use partial factorization over multiple elements (Geffner et al., 2023), which can reduce error accumulation but increases the computational costs during training as more simulations are needed.

**Ordered observations** Ordered observations (i.e., time series) $\mathbf{y}_{1:N} := (\mathbf{y}_1, \mathbf{y}_2, \dots, \mathbf{y}_N) \equiv \mathbf{y}_{\text{obs}}$ are ubiquitous in Bayesian modeling. Practically relevant examples include ODE/SDE type models such as susceptible-infected-recovered (SIR) models (Radev et al., 2021), or Kolmogorov flows (Rozet & Louppe, 2023). A common case is a first-order *Markov model*, where dependencies are limited to adjacent observations in a sequence:

$$p(\boldsymbol{\theta} \mid \mathbf{y}_{1:N}) \propto p(\boldsymbol{\theta}) \, p(\mathbf{y}_1 \mid \boldsymbol{\theta}) \prod_{n=2}^{N} p(\mathbf{y}_n \mid \mathbf{y}_{n-1}, \boldsymbol{\theta}). \tag{30}$$

Compositional estimation can be extended to Markov models by learning a local score estimator for $\nabla_{\boldsymbol{\theta}_t} \log p(\boldsymbol{\theta}_t \mid \mathbf{y}_n, \mathbf{y}_{n-1})$ which requires only single-step transitions from $\mathbf{y}_{n-1}$ to $\mathbf{y}_n$ during training (Gloeckler

et al., 2025). This avoids the need to evolve a first-order Markov process over $N$ steps for each training instance. At inference, the individual scores can be aggregated similarly to (29) across the time series:

$$\nabla_{\boldsymbol{\theta}_t} \log p(\boldsymbol{\theta}_t \mid \mathbf{y}_{1:N}) \approx \Lambda(\boldsymbol{\theta}_t)^{-1} \Big( (1-N) \Sigma_t^{-1} \nabla_{\boldsymbol{\theta}_t} \log p(\boldsymbol{\theta}_t)$$

$$+ \Sigma_{t,1}^{-1} \nabla_{\boldsymbol{\theta}_t} \log p(\boldsymbol{\theta}_t \mid \mathbf{y}_1) + \sum_{n=2}^{N} \Sigma_{t,n}^{-1} \nabla_{\boldsymbol{\theta}_t} \log p(\boldsymbol{\theta}_t \mid \mathbf{y}_n, \mathbf{y}_{n-1}) \Big),$$

where $\Lambda(\boldsymbol{\theta}_t) = \sum_{n=1}^{N} \Sigma_{t,n}^{-1} + (1-N)\Sigma_t^{-1}$, and $\Sigma_t^{-1}$ and $\Sigma_{t,n}^{-1}$ are the denoising prior and posterior precision matrices respectively. The marginal prior score $\nabla_{\boldsymbol{\theta}_t} \log p(\boldsymbol{\theta}_t)$ is typically known analytically, while Linhart et al. (2026) propose two approximation methods for $\Sigma_{t,n}^{-1}$: `GAUSS`, which assumes that the posterior is Gaussian, and analytically computes the denoising posterior covariances; and `JAC`, which iteratively estimates them via Tweedie's moment projection (Boys et al., 2024) using the Jacobian of the score model. Ensuring $\Lambda(\boldsymbol{\theta}_t)$ to be positive definite makes the approximation more stable (Gloeckler et al., 2025), but Touron et al. (2025) show that error accumulation under the assumptions of `GAUSS` is directly related to the sampling quality of each individual posterior and provide a bound on the mean squared error between the compositional score and its estimate as a function of the individual score errors and the error in the estimation of the precision matrix $\Sigma_{t,n}^{-1}$.

A key ingredient here is a proposal distribution $\mathbf{y}_n \sim q_n(\mathbf{y})$ for generating single-step transitions at arbitrary time points $n$ without simulating the full-time horizon. Crucially, the support of the proposal should overlap with the support of the likelihood. If the proposal is too narrow, important regions of the transition space may never be explored; if it is too wide, the estimator may waste capacity on implausible transitions and learn slowly.

**Time-varying parameters** In recursive Bayesian estimation, the aim is to estimate the current latent state $\boldsymbol{\theta}_n$ given a sequence of past observations $\mathbf{y}_{1:n}$. This is a standard setting in filtering problems, where sequential Bayesian updates are carried out in high-dimensional dynamical systems using ensemble-based approximations such as the Ensemble Kalman Filter (Evensen, 2009; Houtekamer & Zhang, 2016). The posterior distribution of interest is therefore

$$p(\boldsymbol{\theta}_n \mid \mathbf{y}_{1:n}) \propto p(\mathbf{y}_n \mid \boldsymbol{\theta}_n, \mathbf{y}_{1:n-1}) \, p(\boldsymbol{\theta}_n \mid \mathbf{y}_{1:n-1}). \tag{31}$$

For every new observation, the posterior needs to be updated, hence both the prior $p(\boldsymbol{\theta}_n \mid \mathbf{y}_{1:n-1})$ and the posterior are time-varying. Similar to (18), applying the score operator to Bayes' rule decomposes this modeling problem into the following (Andry, 2023):

$$\nabla_{\boldsymbol{\theta}_{t,n}} \log p(\boldsymbol{\theta}_{t,n} \mid \mathbf{y}_{1:n}) = \nabla_{\boldsymbol{\theta}_{t,n}} \log p(\mathbf{y}_n \mid \boldsymbol{\theta}_{t,n}, \mathbf{y}_{1:n-1}) + \nabla_{\boldsymbol{\theta}_{t,n}} \log p(\boldsymbol{\theta}_{t,n} \mid \mathbf{y}_{1:n-1}). \tag{32}$$

This shows that the posterior score is the sum of the prior score, which describes the dynamics of the latent state, and the perturbed likelihood score, which incorporates the information from the observation process. One can either train a score network directly to approximate the posterior score on the left (Section 3), or learn the two terms on the right separately and combine them at inference time (Section 4.1). The second option can be advantageous if one has access to an explicit description of the observation process, as this allows the likelihood score to be approximated directly. This line of work is more closely related to inverse problems rather than SBI, and has been applied to posterior sampling in noisy inverse problems (Chung et al., 2023). In this way, decomposing the posterior clarifies that posterior inference in state-space models with diffusion can be addressed either by directly learning the posterior score or by combining learned dynamics with known likelihood terms (Andry, 2023).

### 4.2.2 Complete Pooling Models

A *complete pooling* model assumes that the entire set of observations $\{\mathbf{y}_{\text{obs}}^{(r)}\}_{r=1}^{R}$ is generated by a single, global set of parameters $\boldsymbol{\theta}$. This leads to a single posterior distribution

$$p(\boldsymbol{\theta} \mid \{\mathbf{y}_{\text{obs}}^{(r)}\}_{r=1}^{R}) \propto p(\boldsymbol{\theta}) \, p(\{\mathbf{y}_{\text{obs}}^{(r)}\}_{r=1}^{R} \mid \boldsymbol{\theta}) \propto p(\boldsymbol{\theta})^{1-R} \prod_{r=1}^{R} p(\boldsymbol{\theta} \mid \mathbf{y}_{\text{obs}}^{(r)}). \tag{33}$$

that aggregates all observations without considering potential group or individual differences, thus enforcing complete homogeneity across data points (Bardenet et al., 2017). Complete pooling is a rare target in SBI (and Bayesian inference in general) since its homogeneity assumption is ordinarily too restrictive in practice. However, it provides a straightforward probabilistic aggregation recipe that does not rely on *ad hoc* postprocessing. As shown by Linhart et al. (2026), diffusion models can pool data from multiple experiments by using compositional score aggregation.

The internal probabilistic symmetry of each $\mathbf{y}_{\text{obs}}^{(r)}$ is problem-dependent (e.g., exchangeable, Markovian, *etc.*) and thus, each $\mathbf{y}_{\text{obs}}^{(r)}$ can be a sequence or a set of varying lengths $N_r$. Training a standard diffusion model to sample directly from the posterior over all observations (33) requires $\sum N_r$ simulator runs per training instance if $\mathbf{y}_{\text{obs}}^{(r)}$ itself consists of multiple elements, which is computationally prohibitive. Thus, Linhart et al. (2026) proposed to estimate the individual posterior scores for each $\mathbf{y}_{\text{obs}}^{(r)}$ and combine them using the approximate compositional score expression

$$
\begin{aligned}
\nabla_{\boldsymbol{\theta}_t} \log p(\boldsymbol{\theta}_t \mid \{\mathbf{y}_{\text{obs}}^{(r)}\}_{r=1}^R) \approx \Lambda(\boldsymbol{\theta}_t)^{-1} \Big( & (1-R)\Sigma_t^{-1} \nabla_{\boldsymbol{\theta}_t} \log p(\boldsymbol{\theta}_t) \\
& + \sum_{r=1}^R \Sigma_{t,r}^{-1} \nabla_{\boldsymbol{\theta}_t} \log p(\boldsymbol{\theta}_t \mid \mathbf{y}_{\text{obs}}^{(r)}) \Big),
\end{aligned}
\tag{34}
$$

$\Lambda(\boldsymbol{\theta}_t) = \sum_{r=1}^R \Sigma_{t,r}^{-1} + (1-R)\Sigma_t^{-1}$, and $\Sigma_t^{-1}$ and $\Sigma_{t,r}^{-1}$ are the denoising prior and posterior precision matrices respectively as introduced in the previous section.

### 4.2.3 Partial Pooling Models

A *partial pooling* or hierarchical model represents an intermediate between complete pooling and no-pooling (Gelman et al., 2012). In these models, not only do we have structured observations, but we also introduce additional structure to the parameters: observation-specific parameters $\boldsymbol{\theta}^{(r)}$, which are assumed to be drawn from a shared higher-level distribution governed by hyperparameters $\boldsymbol{\eta}$. This structure naturally arises in many domains with inherent hierarchies, such as patients in pharmacokinetic models (Wakefield, 1996), longitudinal patient omics (Kilian et al., 2024), single-cells in gene expression models (Llamosi et al., 2016), or participants in trails in cognitive modeling (Lee, 2011).

Formally, the posterior has the hierarchical structure:

$$
p(\{\boldsymbol{\theta}^{(r)}\}_{r=1}^R, \boldsymbol{\eta} \mid \{\mathbf{y}_{\text{obs}}^{(r)}\}_{r=1}^R) \propto p(\boldsymbol{\eta}) \left[ \prod_{r=1}^R p(\boldsymbol{\theta}^{(r)} \mid \boldsymbol{\eta}) \, p(\mathbf{y}_{\text{obs}}^{(r)} \mid \boldsymbol{\theta}^{(r)}) \right],
\tag{35}
$$

for i.i.d. observations. This gives rise to one global posterior and $R$ local posteriors, respectively:

$$
p(\boldsymbol{\eta} \mid \{\mathbf{y}_{\text{obs}}^{(r)}\}_{r=1}^R) \propto p(\boldsymbol{\eta})^{1-R} \prod_{r=1}^R p(\boldsymbol{\eta} \mid \mathbf{y}_{\text{obs}}^{(r)})
\tag{36}
$$

$$
p(\boldsymbol{\theta}^{(r)} \mid \mathbf{y}_{\text{obs}}^{(r)}, \boldsymbol{\eta}) \propto p(\boldsymbol{\theta}^{(r)} \mid \boldsymbol{\eta}) \, p(\boldsymbol{\theta}^{(r)} \mid \mathbf{y}_{\text{obs}}^{(r)}, \boldsymbol{\eta}) \quad \text{for} \quad r = 1, \ldots, R.
\tag{37}
$$

Such models allow individual parameters to be informed by global estimates (37), stabilizing inference at low $N_r$ by borrowing information across observations. Even though partial pooling has been espoused as a default choice in Bayesian analysis (Gelman et al., 2012; McElreath, 2018), it typically incurs high computational cost and has only recently received attention in SBI (Heinrich et al., 2024; Habermann et al., 2025; Arruda et al., 2024; Charles et al., 2025).

To amortize two-level models, for example, we can train two diffusion models, one for the global and one for the local posterior. During training, the local model is conditioned on the true parameter $\boldsymbol{\eta}$ and during inference, an ancestral sampling scheme can be used by sampling first from the global and then from the local model. Both models can share an embedding for individual observations. However, targeting the joint posterior (35) or even the global posterior (36) without composition would require $R$ simulations per training instance. Thus, Arruda et al. (2026) ported compositional score estimation to the hierarchical setting,

enabling amortized hierarchical estimation *without ever simulating the full hierarchical model.* To address stability issues, the authors sampled from the unscaled base distribution and applied an error-damping schedule $d(t)$ to the reverse diffusion with a mini-batch sampler using $M \ll S$ randomly sampled observations at each time step $t$. The resulting compositional estimator,

$$\nabla_{\boldsymbol{\eta}_t} \log p(\boldsymbol{\eta}_t \mid \{\mathbf{y}_{\text{obs}}^{(r)}\}_{r=1}^R) \approx d(t)\left((1-R)(1-t)\nabla_{\boldsymbol{\eta}} \log p(\boldsymbol{\eta})\big|_{\boldsymbol{\eta}=\boldsymbol{\eta}_t} + \frac{R}{M}\sum_{m=1}^M \nabla_{\boldsymbol{\eta}_t} \log p(\boldsymbol{\eta}_t \mid \mathbf{y}_{\text{obs}}^{(m)})\right), \quad (38)$$

affords inference for extremely large hierarchical models with $R > 250,000$ observation-specific posteriors.

### 4.3 Inference for Structured Targets

Beyond adaptive inference and compositional modeling, one can change *what* is learned. In this section, we highlight four such targets: (i) structuring parameters across *multiple spatial resolutions*; (ii) embedding *causal structure* directly into the generative mechanism; (iii) learning the *joint* $p(\boldsymbol{\theta}, \mathbf{y})$ to enable arbitrary conditioning without retraining; and (iv) extending scores to *non-Euclidean or infinite-dimensional* spaces.

#### 4.3.1 Multiscale Estimation

Many simulation-based inference problems involve high-dimensional, *spatially structured parameters*, where the resolution can be modeled hierarchically. So instead of multiple observations, we have one high-dimensional observation $\mathbf{y}_{\text{obs}}$, which can be mapped to different resolutions using a downscaling function $h(\mathbf{y}_{\text{obs}}, r)$. We can then offload generation across multiple scales: we start at a coarse resolution and progressively upscale while refining detail. This idea was popularized in conditional image generation by cascaded diffusion pipelines that train separate (class-conditional) models at $(64 \rightarrow 128 \rightarrow 256)$ px resolution stages, conditioning each stage on the upscaled output of the previous one (Ho et al., 2022).

In the SBI setting, Cirakman et al. (2025) combine diffusion models with wavelet decompositions to infer high-resolution velocity fields from seismic data. Here a diffusion model is learned for the top-level coarse component $p(\boldsymbol{\eta}^{(R)} \mid h(\mathbf{y}_{\text{obs}}, R))$ and, for each resolution level $r$, a conditional detail model $p(\boldsymbol{\theta}^{(r)} \mid \boldsymbol{\eta}^{(r)}, h(\mathbf{y}_{\text{obs}}, r))$. This can be combined with an inverse wavelet transform to obtain the conditioning variable for the next finer stage $\text{WT}^{-1}(\boldsymbol{\eta}^{(r)}, \boldsymbol{\theta}^{(r)}) = \boldsymbol{\eta}^{(r-1)}$. The joint posterior over all resolution levels factorizes as

$$\begin{aligned} p(\{\boldsymbol{\eta}^{(r)}\}_{r=0}^R, \{\boldsymbol{\theta}^{(r)}\}_{r=1}^R \mid \mathbf{y}_{\text{obs}}) = p(\boldsymbol{\eta}^{(R)} \mid h(\mathbf{y}_{\text{obs}}, R)) \, p(\boldsymbol{\theta}^{(R)} \mid \boldsymbol{\eta}^{(R)}, h(\mathbf{y}_{\text{obs}}, R)) \\ \times \prod_{r=1}^{R-1} p(\boldsymbol{\theta}^{(R-r)} \mid \boldsymbol{\eta}^{(R-r)}, h(\mathbf{y}_{\text{obs}}, R-r)), \end{aligned} \quad (39)$$

so that training and ancestral sampling proceed naturally from coarse to fine scales. Each score estimator can be trained separately due to the exact factorization above and the analytic deconstruction path from high-resolution training data via $\text{WT}(\boldsymbol{\eta}^{(r)}) = (\boldsymbol{\eta}^{(r+1)}, \boldsymbol{\theta}^{(r+1)})$. While the high-dimensional posterior has long-range spatial correlations, the individual conditional posteriors are closer to Gaussian distributions even when the high-dimensional posterior is not, making estimation of the individual factors easier (Guth et al., 2022). This idea was first demonstrated with normalizing flows by Yu et al. (2020), and Guth et al. (2022) extended it to score-based generative models. Diffusion models are particularly well suited for this setting, as they scale to high dimensions and are very flexible in how the conditioning is used within the neural network architecture.

#### 4.3.2 Causal Posterior Estimation

Causal posterior estimation extends simulation-based inference to settings where the posterior distribution inherits the structure of a causal model (Dirmeier & Mira, 2025). A key idea is to integrate the causal directed acyclic graph (DAG) directly into the network architecture of the diffusion model and the probability flow.

When the posterior program of the model can be represented as a DAG, one can factorize the posterior into components corresponding to the graph structure (as in Section 4.2.3):

$$p(\boldsymbol{\theta} \mid \mathbf{y}_{\text{obs}}) = \prod_{i=1}^{d_{\boldsymbol{\theta}}} p(\boldsymbol{\theta}_{\omega_i} \mid \boldsymbol{\theta}_{<\omega_i}, \mathbf{y}_{\text{obs}}). \tag{40}$$

Each factor in (40) is then parameterized by a neural network $f_i$ such that $\boldsymbol{\theta}_{\omega_i} = f_i(\boldsymbol{\theta}_{\leq \omega_i})$, where $\omega_i$ indexes the node and $\boldsymbol{\theta}_{\leq \omega_i}$ denotes its parents in the graph. This construction ensures that conditional dependencies encoded by the causal model are preserved in the learned posterior. This makes it possible to disentangle effects, propagate interventions through the posterior, and answer causal queries.

The prior can be integrated into the generative flow by selecting the prior distribution as the base measure and defining the probability flow as

$$v(\boldsymbol{\theta}, \mathbf{y}_{\text{obs}}, t) = \gamma \boldsymbol{\theta} + (1 - \gamma)\tilde{v}(\boldsymbol{\theta}, \mathbf{y}_{\text{obs}}, t), \tag{41}$$

where $\tilde{v}$ represents the flow learned by the neural network, and $\gamma \in [0, 1]$ is a trainable parameter that balances the influence of the prior against the learned vector field. With $\gamma = 0$ we recover the classical flow matching target (11) only with a changed base distribution.

### 4.3.3 Joint Estimation

Instead of training only a conditional model for $p(\boldsymbol{\theta} \mid \mathbf{y})$, one can train a diffusion model on the joint $\mathbf{z} = (\boldsymbol{\theta}, \mathbf{y})$ (Gloeckler et al., 2024). Using a transformer as the neural network architecture for the diffusion model, one can process any subset of factors forming the joint distribution. Yet, this architectural choice is prohibitive for constrained architectures (e.g., normalizing flows), as they typically require explicit conditioning on fixed inputs rather than being able to marginalize variables implicitly. In practice, the network is trained with random masking over components of $\mathbf{z}_t^{M_C} = (1 - M_C) \cdot \mathbf{z}_t + M_C \cdot \mathbf{z}_0$, where $M_C \in \{0, 1\}^d$, leading to the modified objective

$$\hat{s} = \underset{s}{\arg\min} \ \mathbb{E}_{M_C \sim p(M_C), \mathbf{z}_0, \mathbf{z}_t, t \sim p(\mathbf{z}_0, \mathbf{z}_t, t)} \left[ \omega_t \left\| (1 - M_C) \cdot \left( s(\mathbf{z}_t^{M_C}, t) - \nabla_{\mathbf{z}_t} \log p(\mathbf{z}_t \mid \mathbf{z}_0) \right) \right\|_2^2 \right]. \tag{42}$$

Implicitly all conditionals and marginals of $p(\boldsymbol{\theta}, \mathbf{y})$ are captured, so it learns to denoise any subset of variables while treating the complementary subset as observed. At inference time, observed variables are fixed at their conditioning values, and the reverse diffusion process is applied to all unobserved variables. Even when the diffusion model is trained only for posterior estimation, the same idea of masking extends to any transformer backbone and allows marginalizing out parameters during inference or conditioning on partial observations (see Figure 6).

As a result, the same trained model learns many conditioning patterns without architectural changes or retraining, which can handle missing or unstructured observations. In contrast, previous approaches require the structure to be encoded during training (Wang et al., 2023; Wang et al., 2024). Similarly, one can learn the joint distribution and provide frequentist coverage guarantees (Jeong et al., 2025) or target the full hierarchical posterior using flow matching (Charles et al., 2025). However, learning all possible conditionals can be computationally expensive, since many simulations are required, and not all conditionals are scientifically meaningful or relevant for the intended inference tasks. Yet, some results indicate that restricting training to posterior masks does not improve performance compared to learning the entire joint distribution (Gloeckler et al., 2024). Nevertheless, experiments on high-dimensional problems are still lacking, leaving the generalizability and practical scalability of learning the joint uncertain.

Beyond posterior approximation, joint training also enables Bayesian model comparison using both the likelihood and the posterior (Gunes et al., 2025). Having access to all conditional distributions makes this approach particularly appealing for sequential neural score estimation. For example, Sharrock et al. (2024) introduced SNPSE-C, which incorporates a score-based correction into the denoising score matching objective to refine posterior estimates iteratively. SNPSE-C requires access to likelihood and prior scores, which introduces additional simplifications and approximations, but these can be avoided when the diffusion model

is trained directly on the joint. Extensions to spatial and non-parametric inference have also been proposed. For instance, Tesso et al. (2025) adapted the framework to spatially correlated data and demonstrated inference over multiple spatial conditional distributions simultaneously by providing temporal and spatial information as inputs to a transformer-based summary network.

### 4.3.4 Learning Scores Beyond Euclidean Spaces

The concept of the score function can be generalized beyond Euclidean parameter spaces to distributions defined on other structured domains. A detailed survey of diffusion models for data with special structures unrelated to SBI is provided by Yang et al. (2023), covering cases such as discrete variables, symmetry-invariant data, and data constrained to manifolds. For targets with intrinsic geometric constraints, the score can be formulated directly on a manifold equipped with its metric structure. Examples include E(3)-equivariant diffusion models (Hoogeboom et al., 2022), which respect rigid-body symmetries, and Riemannian score-based models (Bortoli et al., 2022), where both forward and reverse processes are defined with respect to the manifold geometry. Jagvaral et al. (2022) demonstrated this approach for the estimation of astrophysical orientation distributions on SO(3) extendable to neural likelihood estimation (NLE).

The score can also be defined for infinite-dimensional spaces when the target variable is a function rather than a finite-dimensional vector. Pidstrigach et al. (2024) introduced diffusion models for Hilbert spaces and applied them to generative modeling of functions for Bayesian inverse problems and SBI. This formulation enables posterior estimation for high-dimensional objects like spatial fields, time series, or solutions to partial differential equations. A notable subtlety is that in infinite dimensions, the score matching objective is no longer equivalent to the standard denoising score matching loss, which requires careful reformulation of the training objective.

To address the fragmentation of generative objectives across different structured domains, Generator Matching reformulates the learning problem as matching the infinitesimal generator of a Markov process (Holderrieth et al., 2025). By targeting the linear operator governing the stochastic evolution rather than specific probability paths or noise distributions, this framework provides a unified objective for neural likelihood estimation on arbitrary state spaces, in particular it allows the construction of multi-modal generative models.

## 5 Design Choices for Diffusion Models in Simulation-Based Inference

We have reviewed various applications and adaptations of diffusion models for SBI. Across this growing body of work, researchers adopt different architectural and algorithmic design choices, each of which can substantially influence both computational efficiency and statistical accuracy. This section outlines the most consequential decisions and discusses their implications.

An overarching decision concerns the *inference backbone*, namely the network responsible for sampling the target quantities $\mathbf{z} \sim p(\mathbf{z} \mid \mathbf{x})$. In principle, any architecture capable of handling the structure of the condition $\mathbf{x}$ is admissible, for instance, hybrid convolutional-recurrent models for time series (Zhang & Mikelsons, 2023b), or permutation-invariant networks for sets (Zaheer et al., 2017; Lee et al., 2019). A common strategy for posterior estimation is to separate the model into a summary network and an inference network. The summary network compresses the data into a fixed-dimensional representation (i.e., an encoder or an embedding network), and the inference network is typically implemented as a stack of (residual) MLPs acting on this summary (Radev et al., 2020; Chen et al., 2023b). This has the advantage that for inference, the summary network needs to be applied only once during the whole reverse process. However, this separation is optional; the flexibility of diffusion models allows these components to be folded into a single transformer architecture (Gloeckler et al., 2024). Several transformer-based architectures and simplified training recipes have been proposed for image diffusion models, primarily to scale to higher resolutions and larger compute budgets (Peebles & Xie, 2023; Hoogeboom et al., 2023; 2025). These works show that comparatively plain designs can scale more reliably than the elaborate, multi-stage pipelines based on U-Net (Ronneberger et al., 2015) used in earlier diffusion models (Ho et al., 2020).

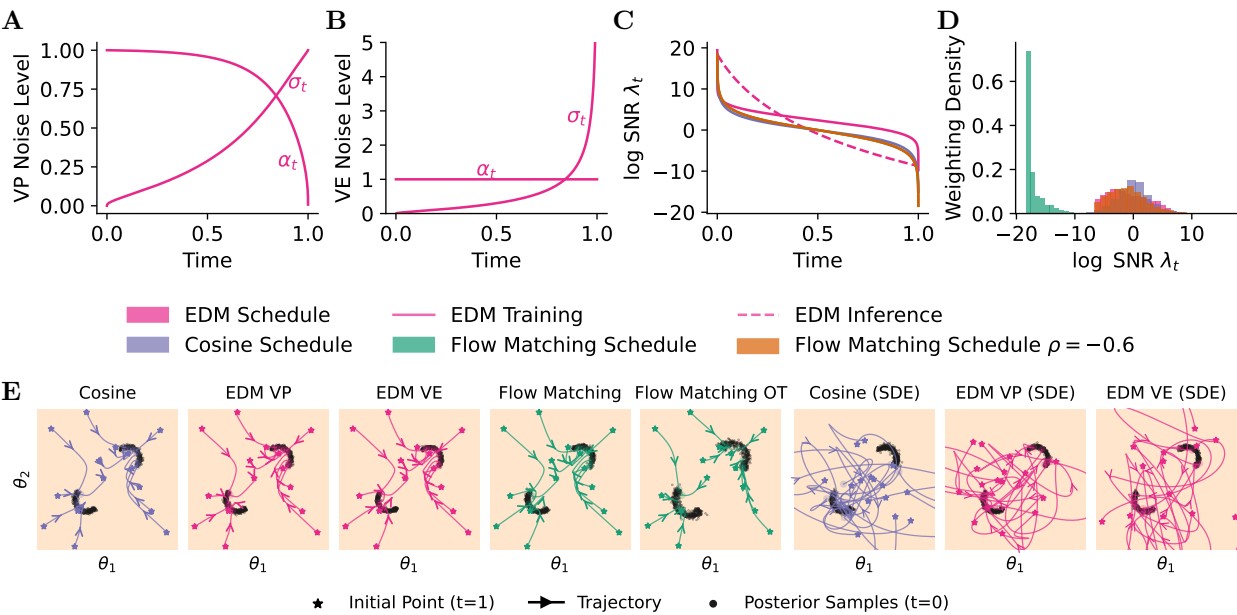

Figure 8: *Designing diffusion models for simulation-based inference (SBI).* **A** Variance-preserving (VP) EDM schedule. **B** Variance-exploding (VE) EDM schedule (until $\sigma_1 = 131.9$). **C** Noise schedules of the log signal-to-noise ratio (SNR). The cosine and flow matching schedules exhibit similar relationships between time and log SNR, yet differ markedly in how the log SNR is weighted in the loss. **D** Flow matching emphasizes regions of low SNR more strongly than the other schedules. This can be adapted by sampling time $t$ from a power-law distribution with $\rho = -0.6$ rather than uniformly, which aligns the effective weighting with that of the EDM schedule. **E** Deterministic and stochastic latent trajectories from noise ($t = 1$) to the posterior ($t = 0$) for diffusion models trained with different noise schedules. The learned trajectories also reflect these design differences. Flow matching with optimal transport (OT) produces relatively straight paths of the probability flow from noise to the posterior, while EDM VP tends to give straighter paths than EDM VE. Stochastic sampling displays highly irregular motion. The example problem shown here is the conditional two moons (Lueckmann et al., 2021) and the stochastic trajectories are smoothed with a simple moving average.

Beyond architectural choices, several training- and inference-level components play a critical role. In what follows, we examine the main elements: the *noise schedule*, the *weighting function*, the *parameterization* of the neural network output, and the *solver* used during inference for the reversed diffusion process.

## 5.1 Variance of the Noise Schedule

A core design choice in diffusion models concerns the denoising process itself. The forward process (5) is characterized by the functions $\alpha_t$ and $\sigma_t$, which together specify the *noise schedule*. These functions determine how signal and noise are blended over time, and thereby shape both the difficulty of the denoising task and the model's overall behavior (Song et al., 2021c). In the *variance-preserving (VP)* parameterization,

$$\alpha_t = \sqrt{1 - \sigma_t^2}, \tag{43}$$

the total variance $\alpha_t^2 + \sigma_t^2$ remains constant across $t$, assuming $\mathrm{Var}(\mathbf{z}_0) = 1$ and $0 \leq \sigma_t \leq 1$ (Figure 8A). This ensures a balance between signal and noise at every step and underlies the original diffusion model formulation (Ho et al., 2020), which is still widely used in practice.

In the *variance-exploding (VE)* case, the signal is held constant,

$$\alpha_t = 1, \qquad \sigma_t \uparrow \text{ as } t \to 1, \tag{44}$$

so that the total variance grows throughout the forward process (Figure 8B). In some low-dimensional SBI inference tasks, VE has been reported to outperform VP (Sharrock et al., 2024; Gloeckler et al., 2024), while

for high-dimensional imaging data, VP schedules generally perform better than VE schedules (Song et al., 2021c). In our experiments Section 6.1, we find the VP schedule to be superior in most SBI tasks.

A third option is the *sub-variance-preserving (sub-VP)* parameterization,

$$\alpha_t = \sqrt{1 - \sigma_t}, \tag{45}$$

which yields a lower total variance than the VP case for all $t$. For density estimation tasks, Song et al. (2021c) observed that a sub-VP noise schedule can improve accuracy. However, most current SBI papers adopt either a VP schedule combined with a linear noise schedule or a VE schedule, and only a single paper employed a sub-VP schedule (Zeng et al., 2025a, see also Table 3).

In practice, the choice of variance type interacts with both the noise schedule and the weighting function, jointly shaping the training difficulty over $t$, as we will discuss in the following sections. Lu & Song (2025) show that the variance type does not fundamentally affect sample quality, since any formulation can be transformed into a VP schedule by rescaling $\alpha_t$ and $\sigma_t$ with $\sqrt{\alpha_t^2 + \sigma_t^2}$ and adjusting the weighting function accordingly. They introduce a trigonometric reparameterization, TrigFlow, which maps arbitrary schedules into VP form. Moreover, Tang & Zhao (2024) proposed contractive variants of VP and sub-VP schedules that suppress the propagation of score and discretization errors, albeit at the cost of introducing possible bias.

## 5.2   Noise Schedule and Signal-to-Noise Ratio

The noise schedule determines how the log signal-to-noise ratio (SNR), $\lambda_t = \log(\alpha_t^2/\sigma_t^2)$, evolves over time $t$, and thus controls the relative difficulty of denoising at different steps. Over the past few years, several schedules have been proposed, each with different motivations and practical tradeoffs (Kingma & Gao, 2023). One can understand the noise schedule as defining a proposal distribution over the SNR (Figure 9), which determines the weight of a given noise level during training (Figure 8C–D). Hence, by applying an appropriate reweighting, one can transform samples from one schedule into those of another, since the weighting function converts the effective distribution over SNR levels (Kingma & Gao, 2023). The most common schedule in SBI is the *linear schedule* from the original diffusion model formulation (Ho et al., 2020), which is defined as

$$\lambda(t) = -\log\big(e^{t^2} - 1\big). \tag{46}$$

It is called linear because it leads to a linear probability flow ODE. However, it tends to concentrate weight on intermediate SNR levels, which can lead to inefficient training compared to newer schedules.

The *cosine schedule* (Nichol & Dhariwal, 2021) instead uses

$$\lambda(t) = -2\log\left(\tan\frac{\pi t}{2}\right) + 2s, \tag{47}$$

where $s$ is a small offset (typically $s = 0$). This schedule smooths the allocation of noise across time steps, avoiding the imbalance of the linear schedule, and has become a widely adopted default for imaging tasks (with $s <$ pixel bin size) due to its empirical robustness in generative modeling.

Finally, the *Elucidated Diffusion Model (EDM) schedule* (Karras et al., 2022), explicitly decouples training and sampling:

$$\lambda_{\text{train}}(t) = \mathcal{F}_{\mathcal{N}}^{-1}(1 - t; 2.4, 2.4^2) \tag{48}$$

$$\lambda_{\text{inference}}(t) = -2\rho \log\left(\sigma_{\text{max}}^{1/\rho} + (1 - t)\left(\sigma_{\text{min}}^{1/\rho} - \sigma_{\text{max}}^{1/\rho}\right)\right), \tag{49}$$

with common hyperparameters $\rho = 7$, $\sigma_{\text{min}} = 0.002$, and $\sigma_{\text{max}} = 80$. This formulation tends to improve generalization and sampling efficiency and is our recommended choice (Section 6).

A good schedule with high numerical accuracy requires the Jacobian of the velocity field to have a small Lipschitz norm (Tsimpos et al., 2025). Empirically, high Lipschitz norms tend to occur near the boundaries $t \approx 0$ and $t \approx 1$. In practice, schedules are often truncated so that the SNR remains well-defined at the boundaries, using the rescaled time $\tilde{t} = t_0 + (t_1 - t_0)t$. It is also common to condition on a normalized

version of $\lambda_t$ instead of the raw time value $t$ for numerical stability, for example, by rescaling $\lambda_t$ to lie within $[-1, 1]$ (Karras et al., 2022). Recent work shows that, to reduce the variance as $t \to 0$, incorporating the tractable joint score $\nabla_{\boldsymbol{\theta}} \log p(\boldsymbol{\theta})$ in the objective function yields an unbiased estimator of the marginal score and improves simulation efficiency (De Bortoli et al., 2024; Ko & Geffner, 2025).

While most SBI papers still employ a linear schedule (Table 3), Nautiyal et al. (2026) showed that the cosine schedule outperforms the linear variant on the low-dimensional benchmark problems of Lueckmann et al. (2021). We additionally show that the EDM schedule outperforms the cosine schedule on the same problems (Section 6.1). Moreover, for neural likelihood estimation (NLE) of a PDE model, Haitsiukevich et al. (2024) demonstrated that the EDM schedule combined with an ODE sampler achieves superior performance compared to the linear schedule.

## 5.3 Weighting Function

The weighting function $\omega(t)$ in score-matching (10) controls how much emphasis different noise levels receive during training. By scaling the contribution of each time step $t$ and thereby the corresponding $\lambda_t$, it biases optimization toward regions of the signal-to-noise ratio where improvements are expected to matter most (Figure 9). While some weighting rules have theoretical motivation (for example, ensuring consistency with variational bounds (Song et al., 2021b)), most are empirical design choices tuned for better convergence, stability, or perceptual quality. Here, we present the weighting functions for the noise-prediction loss (58). A widely used choice is *likelihood weighting* (Song et al., 2021b), which sets

$$\omega(t) = \frac{g(t)^2}{\sigma_t^2}, \tag{50}$$

where $g(t)$ is the diffusion coefficient of the forward SDE. This choice corresponds to minimizing an evidence lower bound (ELBO) of the target distribution $p(\mathbf{z}_0 \mid x)$, and is equivalent to applying uniform weighting over the log-SNR (Figure 9). It is therefore attractive when the goal is to match the forward process likelihood as closely as possible.

In contrast, *EDM weighting* (Karras et al., 2022) uses

$$\omega(t) = 1 + \frac{e^{\lambda_t}}{\sigma_{\text{target}}^2}, \tag{51}$$

which is designed to equalize the expected training loss across the entire $\lambda_t$ range. This weighting works particularly well in combination with the EDM noise schedule and leads to more uniform gradient contributions over time steps, often improving convergence speed and final performance.

Another option is *sigmoid weighting* (Kingma & Gao, 2023),

$$\omega(t) = \text{sigmoid}(-\lambda_t + 2), \tag{52}$$

which monotonically favors lower-SNR regions. This biases learning toward the most difficult denoising steps, where the model benefits most from additional capacity.

In practice, most SBI papers either omit explicit weighting, effectively setting $\omega(t) = 1$, or apply EDM weighting in combination with the EDM schedule (Table 3). In our experiments (Section 6), we find the EDM weighting to be superior for low-dimensional problems with regard to the statistical accuracy of the diffusion estimators.

## 5.4 Model Parameterization

Diffusion models can be parameterized in terms of what they predict, and each choice can be advantageous at different noise levels. As we will see next, we can easily convert one parameterization into another. In *score prediction* (Song & Ermon, 2019), the network predicts the score $\hat{\boldsymbol{s}}$ directly. This parameterization is statistically elegant because the score function uniquely characterizes the target distribution. It has also been the most common choice in SBI, often used without explicit reweighting, which implicitly puts more weight

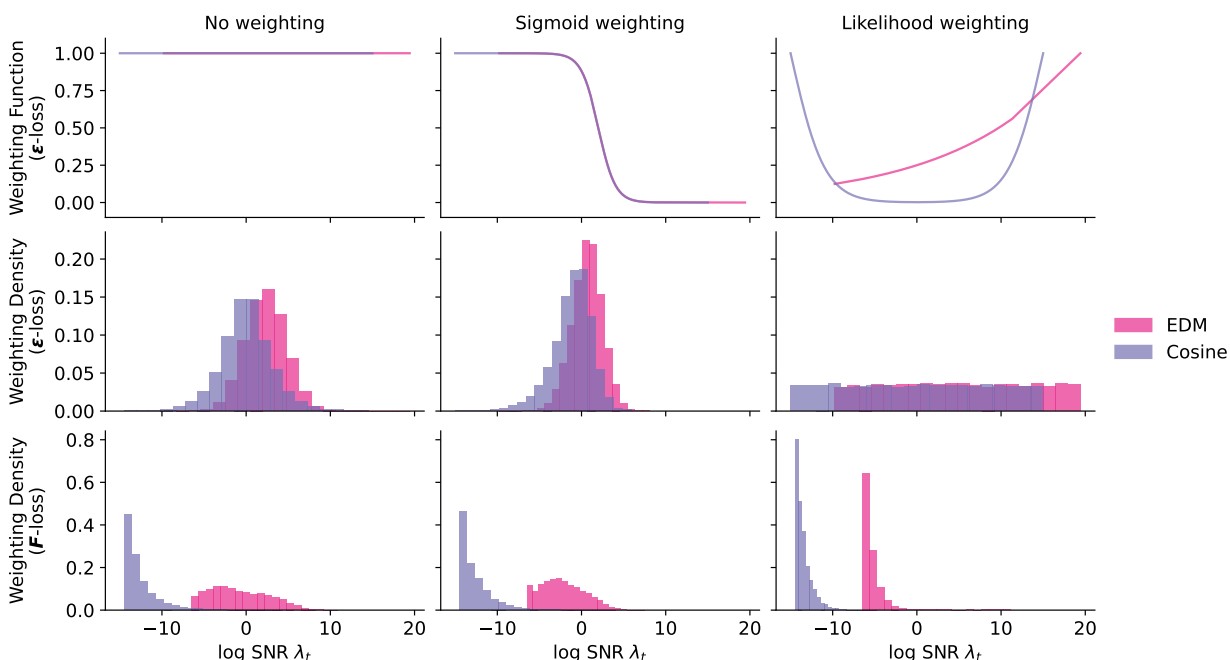

Figure 9: *Comparison of weighting functions and their effect on the effective weighting of the signal-to-noise ratio.* We use the EDM and cosine noise schedules with the loss defined on $\boldsymbol{\epsilon}$. Changing the weighting function can change the effective weighting of the log signal-to-noise ratio (SNR) $\lambda_t$ drastically. Given a fixed weighting, switching to $\mathbf{F}$-prediction (or equivalently applying the EDM weighting to the $\boldsymbol{\epsilon}$-loss) alters the implied weighting profile again.

on time steps in the low-noise regime. However, the network has to learn scores whose magnitudes vary by several orders across noise levels, which can make training difficult. To mitigate this, Song & Ermon (2020) proposed using a rescaled version of the score, $\boldsymbol{s}/\sigma_t$, as the training target, which normalizes the scale of the target across time steps and leads to more stable training across the full range of noise levels.

In *noise prediction* (Ho et al., 2020), the network instead predicts

$$\hat{\boldsymbol{\epsilon}}_t = \frac{\mathbf{z}_t - \alpha_t \mathbf{z}_0}{\sigma_t}. \tag{53}$$

Noise prediction is widely used in image-generating diffusion models because it yields a simpler loss (10) by using the fact that $\boldsymbol{s} = -\boldsymbol{\epsilon}_t/\sigma_t$. However, it tends to underweight very clean samples ($t \approx 0$), which can be detrimental in SBI when accurate recovery of $\mathbf{z}_0$ is important, and can fail drastically (Section 6.2).

An alternative is *target prediction* (Song et al., 2021a), where the diffusion model predicts the clean target

$$\hat{\mathbf{z}}_0 = \frac{\mathbf{z}_t - \sigma_t \boldsymbol{\epsilon}_t}{\alpha_t}. \tag{54}$$

This choice emphasizes the reconstruction quality of $\mathbf{z}_0$ and often performs better at very low noise levels. However, it can become numerically unstable at high noise ($t \approx 1$), where the network must extrapolate $\mathbf{z}_0$ from nearly pure noise.

In *velocity prediction* (Salimans & Ho, 2022), the model predicts a linear combination of $\boldsymbol{\epsilon}_t$ and $\mathbf{z}_0$:

$$\hat{\mathbf{v}}_t = \alpha_t \boldsymbol{\epsilon}_t - \sigma_t \mathbf{z}_0, \qquad \hat{\mathbf{z}}_0 = \alpha_t \mathbf{z}_t - \sigma_t \hat{\mathbf{v}}_t \quad \text{(VP case)}. \tag{55}$$

This reparameterization aims to make the prediction target have constant variance across $t$, leading to more balanced training and often faster convergence compared to pure noise or data prediction.

Finally, *EDM prediction* (Karras et al., 2022) generalizes this idea by reconditioning the target with

$$\hat{\mathbf{F}} = -\frac{\sqrt{e^{-\lambda_t} + \sigma_{\text{target}}^2}}{\sigma_{\text{target}}}\boldsymbol{\epsilon}_t + \frac{e^{\lambda/2}\left(e^{-\lambda_t} + \sigma_{\text{target}}^2 - \sigma_{\text{target}}^2\alpha_t^2\right)}{\sqrt{e^{-\lambda_t} + \sigma_{\text{target}}^2}\sigma_{\text{target}}\alpha_t}\mathbf{z}_t, \tag{56}$$

where typically $\sigma_{\text{target}} = 1$. The original EDM formulation used $\sigma_{\text{target}} = 0.5$ specifically for modeling images that are normalized to the range $[-1, 1]$, whereas in SBI contexts, it is more natural to standardize the target distributions. This prediction choice produces a target whose variance is nearly independent of $t$, improving both stability and sample quality, particularly designed for ODE-based samplers. Together with velocity prediction, the EDM formulation is our recommended choice for SBI problems (Section 6).

Each parameterization shifts the learning emphasis: $\boldsymbol{\epsilon}$-prediction is robust but weak at very low noise; $\mathbf{z}_0$-prediction excels at low noise but becomes unstable at high noise; velocity and EDM prediction recondition the problem to achieve uniform difficulty across time steps. While most SBI papers still use the score prediction, there is growing interest in EDM-style targets due to their improved stability and efficiency (Table 3). Crucially, with appropriate weighting of the loss, the different parameterizations are mathematically equivalent (Kingma & Gao, 2023):

$$\sigma_t^2\big\|\hat{s}(\mathbf{z}_t, \mathbf{x}, \lambda_t) - \nabla_{\mathbf{z}_t}\log p(\mathbf{z}_t \mid \mathbf{z}_0)\big\|_2^2 \qquad \text{(score prediction)} \tag{57}$$

$$= \|\hat{\boldsymbol{\epsilon}}_t - \boldsymbol{\epsilon}_t\|_2^2 \qquad \text{(noise prediction)} \tag{58}$$

$$= e^{\lambda_t}\|\hat{\mathbf{z}}_0 - \mathbf{z}_0\|_2^2 \qquad \text{(target prediction)} \tag{59}$$

$$= \frac{1}{\alpha_t^2(e^{-\lambda_t} + 1)^2}\|\hat{\mathbf{v}} - \mathbf{v}\|_2^2 \qquad \text{(velocity prediction)} \tag{60}$$

$$= \frac{1}{e^{-\lambda_t}/\sigma_{\text{target}}^2 + 1}\|\hat{\mathbf{F}} - \mathbf{F}\|_2^2 \qquad \text{(EDM prediction)} \tag{61}$$

If no correction is applied, switching the parameterization implicitly changes the weighting function. In fact, score prediction without weighting corresponds to using noise prediction with weights proportional to $1/\sigma_t^2$. A practical strategy is to predict any convenient target, map it back to the noise domain, and compute the loss there (Salimans & Ho, 2022). This ensures that the parameterization does not distort the effective weighting over noise levels $\lambda_t$. Moreover, it is also possible to predict a scalar potential $p(\mathbf{z}_t)$ to ensure the prediction is a valid score at the cost of repeatedly differentiating during inference (Guo et al., 2023). Different methods, such as continuous normalizing flows, flow matching, and consistency models, are all connected to diffusion models by the choice of parameterization of the neural estimator, as will be discussed in Section 5.6.

## 5.5 Design Choices During Inference

In order to sample from diffusion models, we need to solve the corresponding reverse process (Section 3). As we have seen before, the reverse diffusion process can either be formulated as a stochastic differential equation (SDE) or as a deterministic ordinary differential equation (ODE), where only the initial point is randomly sampled. This provides the foundation for applying or designing different samplers and numerical solvers, which can be swapped during inference.

**SDE Formulation** For a known forward process, the drift $f(t)$ and the diffusion coefficient $g(t)$ can be expressed through the log signal-to-noise ratio (SNR) $\lambda_t$ (Kingma & Gao, 2023). In the variance-preserving case, the drift and the diffusion coefficient are given by

$$f(t) = -\frac{1}{2}\left(\frac{\mathrm{d}}{\mathrm{d}t}\log(1 + e^{-\lambda_t})\right) \quad \text{and} \quad g(t)^2 = \frac{\mathrm{d}}{\mathrm{d}t}\log(1 + e^{-\lambda_t}), \tag{62}$$

respectively, with $\alpha_t^2 = \text{sigmoid}(\lambda_t)$, $\sigma_t^2 = \text{sigmoid}(-\lambda_t)$, and latent prior $p(\mathbf{z}_1) = \mathcal{N}(\mathbf{0}, \mathbf{I})$. For instance, under a linear schedule, the forward process reduces to

$$\mathrm{d}\mathbf{z}_t = -t\mathbf{z}\,\mathrm{d}t + \sqrt{2t}\,\mathrm{d}\mathbf{W}_t \tag{63}$$

Contrary to the name, the SDE does not preserve variance, as the variance of $\mathbf{z}_t$ implied by (63) is $\mathrm{Var}(\mathbf{z}_t) = 1 - e^{t^2}$. The term "variance-preserving" instead refers to the discrete-time parameterization where noise injection balances signal decay. For a known variance-exploding process, the drift and diffusion coefficients are given by

$$f(t) = 0 \quad \text{and} \quad g(t)^2 = \frac{\mathrm{d}}{\mathrm{d}t} \log(1 + e^{-\lambda_t}), \tag{64}$$

with $\alpha_t^2 = 1$, $\sigma_t^2 = e^{-\lambda_t}$, and base distribution $p(\mathbf{z}_1) = \mathcal{N}(\mathbf{0}, e^{-\lambda_1}\mathbf{I})$. Solving the reverse SDE (6) with the Euler-Maruyama scheme and step size $\Delta t$ gives

$$\mathbf{z}_{t-\Delta t} = \mathbf{z}_t - [f(t)\,\mathbf{z}_t - g(t)^2 \nabla_{\mathbf{z}_t} \log p_t(\mathbf{z}_t \mid \mathbf{x})]\Delta t + \sqrt{\Delta t}\,g(t)\boldsymbol{\epsilon}_t \tag{65}$$

with $\boldsymbol{\epsilon}_t \sim \mathcal{N}(\mathbf{0}, \mathbf{I})$. This continuous-time scheme recovers the discrete sampler of Ho et al. (2020).

This stochastic formulation permits the use of a broad range of SDE solvers. Because the dynamics include an additive noise term, one can employ solvers that achieve higher accuracy than the standard Euler-Maruyama scheme. Examples include SEA, a one-step method with strong order 1 for additive-noise SDEs (Foster et al., 2024), and ShARK, a two-step method attaining strong order 1.5 (Foster et al., 2024). In our experiments (Section 6.1), the adaptive two-step solver introduced by Jolicoeur-Martineau et al. (2021) provides the most favorable balance between speed and accuracy.

**ODE Formulation** We have seen that every diffusion SDE has an associated deterministic process called the probability flow ODE (Song et al., 2021c). The key observation is that the Fokker-Planck equation governing the marginal densities of the SDE admits both a stochastic solution (the reverse-time SDE) and a deterministic solution (the ODE). Both processes share the same marginal distributions $p(\mathbf{z}_t)$ for all $t \in [0, 1]$, meaning that solving either produces identically distributed samples in theory. The probability flow ODE can be expressed as

$$\mathrm{d}\mathbf{z}_t = v(\mathbf{z}_t, \mathbf{x}, t)\,\mathrm{d}t = \left[ f(t)\,\mathbf{z}_t - \frac{1}{2}g(t)^2 \nabla_{\mathbf{z}_t} \log p(\mathbf{z}_t \mid \mathbf{x}) \right] \mathrm{d}t. \tag{66}$$

The ODE provides a direct flow that transports the distribution smoothly from randomly sampled noise at $t = 1$ to the target at $t = 0$. In terms of $\alpha_t$ and $\sigma_t$, Karras et al. (2022) express the ODE as

$$\mathrm{d}\mathbf{z}_t = \frac{\alpha_t'}{\alpha_t}\mathbf{z}_t - \alpha_t^2 \sigma_t' \sigma_t \nabla_{\mathbf{z}_t} \log p\left( \frac{\mathbf{z}_t}{\alpha_t} \mid \mathbf{x} \right) \mathrm{d}t. \tag{67}$$

For the variance-exploding (VE) case, the above simplifies to $\mathrm{d}\mathbf{z}_t = -\sigma_t' \sigma_t \nabla_{\mathbf{z}_t} \log p(\mathbf{z}_t \mid \mathbf{x})\mathrm{d}t$. The resulting ODE can be approximated with any standard numerical solver, making high-order methods with adaptive step sizes particularly attractive, as they reduce approximation error and avoid manual discretization schedules (Section 6.1).

However, ODE solvers can potentially suffer from collapse errors, since deterministic flows may fail to explore the typical set of $p(\mathbf{z} \mid \mathbf{x})$. In contrast, the stochasticity in SDE solvers naturally spreads probability mass more broadly. Recent work shows that network architectures with skip connections (e.g., EDM parameterization) help mitigate mode collapse (Zhang & Zou, 2024). Our experiments suggest that SDE solvers outperform ODE solvers slightly on low-dimensional problems, but not on high-dimensional ones (Section 6). In SBI, collapse errors can be diagnosed by checking the calibration of the approximate posteriors (Modrák et al., 2025).

**Other sampling methods** Besides solving either the ODE or SDE, one can also employ the learned score directly in *annealed Langevin dynamics* (Song & Ermon, 2019). Langevin sampling evolves target samples through a discretized stochastic gradient ascent on the log density,

$$\mathbf{z}_{t-\Delta t} = \mathbf{z}_t + \frac{\eta_t}{2}\hat{s}(\mathbf{z}_t, \mathbf{x}, t) + \sqrt{\eta_t}\boldsymbol{\epsilon}_t \quad \text{with} \quad \boldsymbol{\epsilon}_t \sim \mathcal{N}(\mathbf{0}, \mathbf{I}), \tag{68}$$

where $\eta_t$ is a step size or temperature schedule that is gradually reduced ("annealed") as noise is removed. The temperature schedule is usually set in accordance with the noise schedule, for example $\eta_t \propto \sigma_t^2$.

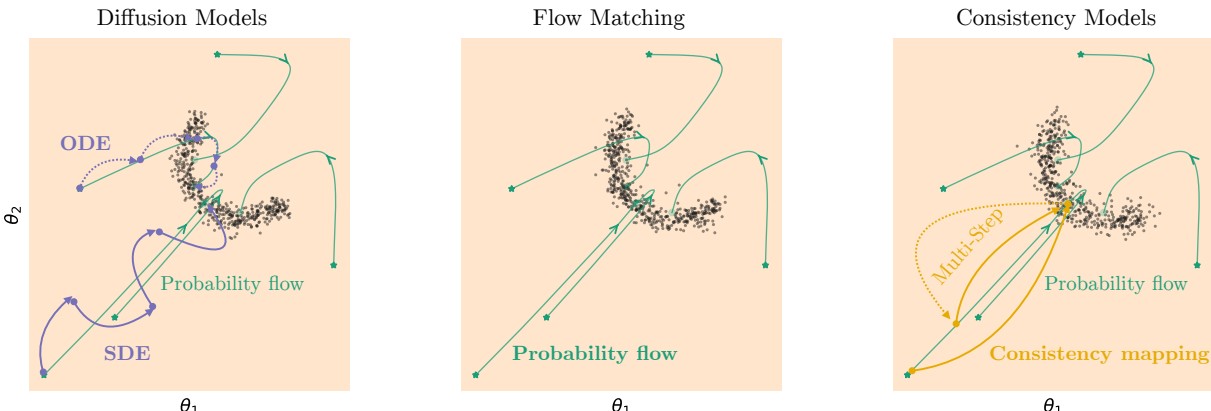

Figure 10: *Diffusion models, flow matching, and consistency models for posterior inference.* Diffusion models solve an ODE or an SDE following the probability flow; flow matching learns the vector field for any point along the probability flow to the posterior; consistency models learn to map any point along the probability flow to the posterior.

In contrast to solving the reverse SDE or ODE, the Langevin formulation explicitly discretizes time and applies stochastic updates at each step. This can be useful if the score is modified *post hoc* after training, since the corresponding marginal distribution $p(\mathbf{z}_t)$ may then belong to a different reverse process (Geffner et al., 2023; Linhart et al., 2026; Arruda et al., 2026). However, in practice, the reverse SDE is often still employed using a modified score with good results (Section 4). Song et al. (2021c) introduce corrector-predictor sampling, which combines an Euler-Maruyama step with an additional annealed Langevin step to improve image quality. However, our results *do not show an increased statistical accuracy for SBI tasks* (Section A.1.1).

So far, most SBI papers have employed either continuous or discretized versions of the reverse ODE or SDE for inference (Table 3). In the SBI setting, posterior calibration can be systematically checked using simulation-based calibration (SBC; Talts et al., 2018; Lemos et al., 2023; Modrák et al., 2025), making it straightforward to compare samplers in terms of statistical accuracy and efficiency. Since the choice of sampler is made at inference time, it does not affect training. This flexibility allows users to switch between different solvers depending on whether the goal is fast point estimation, fully Bayesian inference, or likelihood evaluation.

### 5.6 Design Choices Beyond the SDE Paradigm

Diffusion models are most commonly expressed through the score and the corresponding SDE, where noisy trajectories are simulated forward and reversed during inference (Section 3). However, recent work has introduced alternative training objectives that bypass explicit SDE simulation. These methods aim to retain the flexibility of diffusion models, but simplify parameterization and accelerate inference. Two prominent model families have emerged as particularly relevant for SBI: *conditional flow matching* (Wildberger et al., 2023) and *conditional consistency models* (Schmitt et al., 2025). Both can be interpreted as competitive extensions or simplifications of score matching that do not require solving an SDE for sampling.

#### 5.6.1 Conditional Flow Matching

A useful parameterization of score matching expresses the (deterministic) velocity field $v(\mathbf{z}_t, \mathbf{x}, t)$ of the probability flow (8) as a linear combination of clean samples $\mathbf{z}_0$ and noise $\boldsymbol{\epsilon}_t$. Given a noise schedule $(\alpha_t, \sigma_t)$ and a time-varying state $\mathbf{z}_t = \alpha_t \mathbf{z}_0 + \sigma_t \boldsymbol{\epsilon}_t$ with $\boldsymbol{\epsilon}_t \sim \mathcal{N}(\mathbf{0}, \mathbf{I})$, the velocity field can be expressed as a linear combination of the derivatives of $\alpha_t$ and $\sigma_t$:

$$v(\mathbf{z}_t, \mathbf{x}, t) := \frac{\mathrm{d}\mathbf{z}_t}{\mathrm{d}t} = \frac{\mathrm{d}}{\mathrm{d}t}\left(\alpha_t \mathbf{z}_0 + \sigma_t \boldsymbol{\epsilon}_t\right) = \frac{\mathrm{d}\alpha_t}{\mathrm{d}t}\mathbf{z}_0 + \frac{\mathrm{d}\sigma_t}{\mathrm{d}t}\boldsymbol{\epsilon}_t. \tag{69}$$

This formulation corresponds to the conditional version of *flow matching* (Lipman et al., 2023, only with reversed time $t$), also introduced as *rectified flow* (Liu et al., 2022) and discussed by Lu & Song (2025).

**Training flow matching** Conditional flow matching trains a neural network $\hat{u}(\mathbf{z}_t, \mathbf{x}, t) \approx v(\mathbf{z}_t, \mathbf{x}, t)$ to approximate the vector field in the probability flow (8). In combination with the flow-matching noise schedule $\alpha_t := 1 - t$ and $\sigma_t := t$ (with end time $t = 1$), this leads to the *flow-matching loss*:

$$\hat{u} = \underset{u}{\arg\min} \ \mathbb{E}_{\mathbf{z}_0 \sim p(\mathbf{z}_0), \, t \sim \mathcal{U}(0,1), \, \boldsymbol{\epsilon}_t \sim \mathcal{N}(\mathbf{0}, \mathbf{I})} \left[ \omega_t \left\| u\big((1-t)\, \mathbf{z}_0 + t\, \boldsymbol{\epsilon}_t, t\big) - (\boldsymbol{\epsilon}_t - \mathbf{z}_0) \right\|_2^2 \right], \tag{70}$$

where $\omega_t$ is again a weighting function. The prediction $\hat{\mathbf{u}} = \hat{u}(\mathbf{z}_t, \mathbf{x}, t)$ is therefore the average difference vector between pure noise $\boldsymbol{\epsilon}_t$ and noise-free target $\mathbf{z}_0$ of the (infinitely many) SDE trajectories that cross the location $\mathbf{z}_t$ at time $t$ (Figure 10).

Kingma & Gao (2023) show that the loss defined on the vector field $v(\mathbf{z}_t, \mathbf{x}, t)$ is equivalent to the standard noise-prediction loss, up to a time-dependent weighting:

$$\|\boldsymbol{\epsilon}_t - \hat{\boldsymbol{\epsilon}}_t\|_2^2 = \frac{1}{1 + e^{-\lambda_t} + 2e^{-\lambda_t/2}} \|v(\mathbf{z}_t, \mathbf{x}, t) - \hat{\mathbf{u}}\|_2^2. \tag{71}$$

In practice, flow matching is usually trained with the unweighted objective on $v$ and with the simple schedule $\alpha_t = 1 - t$, $\sigma_t = t$, where $t \sim \mathcal{U}[0,1]$. Flow matching tends to produce straighter trajectories compared to standard score-based parameterizations, and hence allows faster simulation of the corresponding ODE (Liu et al., 2022).

In the context of SBI, conditional flow matching was applied by Wildberger et al. (2023), who also proposed a power law distribution for $t$, with $p(t) \propto (1-t)^{1/(1+\rho)}$ and $\rho > -1$. Using this distribution corresponds to changing the weighting function, that is, assigning greater importance to the vector field closer to $\mathbf{z}_0$ for $\rho < 0$. Wildberger et al. (2023) and Orsini et al. (2025a) empirically found that $\rho < 0$ improves learning for distributions with sharp bounds. We observed that a value of $\rho = -0.6$ gives a weighting distribution of $\lambda_t$ similar to the EDM formulation (Figure 8D) and that it *improves learning only on low-dimensional SBI tasks* (Section 6). Similarly, Gebhard et al. (2025) put more emphasis on higher signal-to-noise ratio by sampling $\sqrt[4]{t} \sim \mathcal{U}[0,1]$.

**Optimal transport** An important extension is *optimal transport conditional flow matching (OT-CFM)*, proposed by Tong et al. (2024) and Pooladian et al. (2023). This approach generalizes standard flow matching by replacing the independent coupling $(\mathbf{z}_0, \mathbf{z}_1)$ with the 2-Wasserstein optimal transport plan $\pi(\mathbf{z}_0, \mathbf{z}_1)$ (Benamou & Brenier, 2000). Instead of sampling $\mathbf{z}_0$ and $\mathbf{z}_1$ independently, one samples $(\mathbf{z}_0, \mathbf{z}_1) \sim \pi$, which encourages interpolation paths to respect the optimal transport geometry of the target distribution. This modification has a key advantage: learned flows are straighter (Figure 8E), since trajectories follow optimal transport geodesics. Hence, inference is accelerated, as the network learns paths that are naturally shorter and closer to the target manifold.

The main drawback of OT-CFM is computational: training typically takes about $2.5\times$ longer due to the cost of computing and sampling from optimal transport couplings. The latter requires large batch sizes to find better pairs (Zhang et al., 2025), while learned couplings might be optimal on the batch level but partially wrong in comparison with the optimal transportation plan (Nguyen et al., 2022). Recent work tries to overcome the computational bottleneck with a semi-discrete formulation (Mousavi-Hosseini et al., 2025). Moreover, Fluri & Hofmann (2024) and Cheng & Schwing (2025) adapted this idea to the conditional setting. Since standard mini-batch reordering ignores the condition $\mathbf{x}$, the authors propose a modified transport cost with an additional penalty term $d(\mathbf{x}^{(i)}, \mathbf{x}^{(j)})$ leading to *conditional optimal transport (COT)*, where $d$ is a weighted distance measure on the raw or embedded observations $\mathbf{x}$ (Section A.1.1).

### 5.6.2 Conditional Consistency Models

Consistency models in discrete and continuous time (Song et al., 2023) can be seen as a distillation scheme for diffusion models, or as a standalone generative model family. In the context of SBI, discrete-time conditional consistency models were adapted by Schmitt et al. (2025). These models learn a *consistency function $c$* that

maps a noisy state $\mathbf{z}_t$ directly to the clean target $\mathbf{z}_0$. The defining property is that the mapping should be consistent across all points $(\mathbf{z}_t, t)$ along the same probability flow ODE trajectory: $c(\mathbf{z}_t, \mathbf{x}, t) = c(\mathbf{z}_{t'}, \mathbf{x}, t')$, with boundary condition $c(\mathbf{z}_0, \mathbf{x}, 0) = \mathbf{z}_0$. In practice, the consistency function is parameterized using skip connections:

$$c(\mathbf{z}_t, \mathbf{x}, t) = c_{\text{skip}}(t)\mathbf{z}_t + c_{\text{out}}(t)f(\mathbf{z}_t, \mathbf{x}, t), \tag{72}$$

where $c_{\text{skip}}(0) = 1$, $c_{\text{out}}(0) = 0$, and $f$ denotes the learnable consistency model. Inference is straightforward: we sample from the base distribution $\mathbf{z}_1$ and obtain the denoised output in a single step via $\mathbf{z}_0 = c(\mathbf{z}_1, \mathbf{x}, 1)$. Thus, consistency functions offer the appealing possibility of *single-step sampling*, thereby bypassing the need for ODE or SDE solvers.

**Training consistency models**   There are two principal approaches to training consistency models. The first distills a pre-trained diffusion model by sampling points along its probability flow ODE trajectory (8) and using them as supervision. This method is computationally expensive, as it requires repeated sampling from the diffusion model, and has not been applied to SBI. The second approach, adopted by Schmitt et al. (2025), trains a consistency model directly as a standalone estimator without relying on a pre-trained diffusion model. The training objective for discrete conditional consistency models is

$$\hat{c} = \underset{c}{\arg\min}\, \mathbb{E}_{\mathbf{z}_0, \mathbf{x} \sim p(\mathbf{z}_0, \mathbf{x}),\, t \sim p(t),\, \boldsymbol{\epsilon} \sim \mathcal{N}(\mathbf{0}, \mathbf{I})} \left[ \omega_t\, d\big(c(\mathbf{z}_t, \mathbf{x}, t), \bar{c}(\mathbf{z}_{t-\Delta t}, \mathbf{x}, t - \Delta t)\big) \right], \tag{73}$$

where $\Delta t > 0$ is the time step size, $\omega_t$ is a weighting function, $d$ is a distance metric, and $\bar{c}(\cdot)$ denotes a teacher model that is an exponential moving average (EMA) version of the student network $c(\cdot)$. Song & Dhariwal (2024) suggested to use $\omega_t = 1/\Delta t$, a Pseudo-Huber distance, a curriculum for the discretization steps, and to update the student model directly with the parameters of the teacher when trained in isolation. These settings were adopted by Schmitt et al. (2025) for SBI, together with a variance-exploding EDM noise schedule.

To construct training pairs $(\mathbf{z}_{t-\Delta t}, \mathbf{z}_t)$, one needs access to the score to integrate the probability flow ODE. In the absence of a pre-trained diffusion model, the score can be approximated by the unbiased estimator,

$$\nabla_{\mathbf{z}_t} \log p(\mathbf{z}_t) = -\mathbb{E}_{\mathbf{z}_0 \sim p(\mathbf{z}_0 \mid \mathbf{z}_t)} \left[ \frac{\mathbf{z}_t - \alpha_t \mathbf{z}_0}{\sigma_t^2} \right], \tag{74}$$

which is sufficient when using Euler integration as the ODE solver (Song et al., 2023). This substitution allows consistency models to be trained without explicit reliance on diffusion models, making them a flexible standalone sampler for SBI. However, using an ODE solver introduces additional discretization error. To avoid this, Song et al. (2023) shows that, in the limit $\Delta t \to 0$ with a squared error distance $d(\mathbf{z}_0, \hat{\mathbf{z}}_0) = \|\mathbf{z}_0 - \hat{\mathbf{z}}_0\|_2^2$, the gradient of (73) can be computed directly, without explicitly solving the ODE. Building on this idea, Lu & Song (2025) propose continuous consistency models based on trigonometric noise schedules derived from the unit-variance principle (similar to the EDM schedule). Their consistency function uses $c_{\text{skip}}(t) = \cos(0.5\pi t)$ and $c_{\text{out}}(t) = -\sin(0.5\pi t)\sigma_{\text{target}}$ with target standard deviation $\sigma_{\text{target}}$, yielding an MSE objective whose gradient matches (73). To further stabilize training, Lu & Song (2025) introduce several improvements: tangent normalization, tangent warm-up, and adaptive weighting, for which we refer to the original publication.

**Sampling with consistency models**   To improve sample quality, few-step sampling can be performed over a discrete sequence of time points $\tau_k$ with $\tau_0 = 1 > \ldots > \tau_K = 0$. The idea is to iteratively refine estimates of the clean sample $\mathbf{z}_0$ from progressively less noisy states. Starting from the initial distribution $\mathbf{z}_1$, one first evaluates $\hat{\mathbf{z}}_0^{(\tau_0)} = \hat{c}(\mathbf{z}_1, \mathbf{x}, \tau_0)$. At each subsequent step $k$, noise is reintroduced and the target estimate is updated according to:

$$\mathbf{z}_{\tau_k} = \alpha_{\tau_k} \hat{\mathbf{z}}_0^{(\tau_{k-1})} + \sigma_{\tau_k} \boldsymbol{\epsilon}_k \quad \text{with} \quad \boldsymbol{\epsilon}_k \sim \mathcal{N}(\mathbf{0}, \mathbf{I}) \tag{75}$$

$$\hat{\mathbf{z}}_0^{(\tau_k)} = \hat{c}(\mathbf{z}_{\tau_k}, \mathbf{x}, \tau_k). \tag{76}$$

Repeating these steps for decreasing $\tau_k$ produces a refined trajectory of reconstructions, with the final output $\hat{\mathbf{z}}_0^{(\tau_K)}$ serving as a sample from $q(\mathbf{z} \mid \mathbf{x}) \approx p(\mathbf{z} \mid \mathbf{x})$. Schmitt et al. (2025) show that few-step sampling in the order of $\approx 10$ steps strikes a good balance between accuracy and sampling speed on SBI problems.

In the SBI setting, consistency models have important tradeoffs. A key limitation is that they do not allow density evaluation at arbitrary targets $\mathbf{z}$, since the inverse of the consistency mapping is not available in closed form. However, they offer few-step sampling and strong performance in low-data regimes (Schmitt et al., 2025). To the best of our knowledge, continuous consistency models (Lu & Song, 2025) had not been applied in SBI prior to this work. Discrete variants have appeared only in the original work and in a subsequent medical imaging study (Zhao et al., 2025), where they matched the accuracy of diffusion models and MCMC, but achieved superior sampling speed.

## 6 Empirical Evaluation

A variety of design choices have been proposed for diffusion models. To assess their practical impact in SBI, we compare specific configurations and derive practical recommendations. For clarity, we adopt a fixed training protocol (Section A.1) and vary only the model family, the parameterization, or the noise schedule, while training models using the noise-prediction loss in (10). All models were trained using `BayesFlow` (Kühmichel et al., 2026).

We begin with a low-dimensional benchmark to establish baseline behavior. We then move to a high-dimensional ODE problem with high parameter and data dimensionality. Next, we study high-dimensional Gaussian random fields, which allow us to vary both parameter and observation dimensionality and thus evaluate the influence of the simulation budget. Finally, we consider an application involving compositional scores, where the interplay between model structure and design choices of the diffusion model becomes evident.

### 6.1 Case Study 1: Low-Dimensional Benchmarks

**Setting** Our first study presents the most comprehensive evaluation to date of diffusion models as posterior estimators on popular toy benchmark problems (Lueckmann et al., 2021). This study aims to highlight differences between the numerous design choices discussed in the preceding sections for the easiest types of problems (low-dimensional parameters, low-dimensional observations, easily obtainable high simulation budgets, see Table 2).

We compare a broad collection of model families and algorithmic variants:

- *Diffusion models under different noise schedules*, using EDM noise schedules (variance-preserving (VP) or variance-exploding (VE)) with EDM weighting, and diffusion models using a cosine VP schedule with sigmoid or likelihood weighting;

- *Diffusion models under different parameterizations*, including $\mathbf{F}$-, $\boldsymbol{v}$-, and $\boldsymbol{\epsilon}$-prediction and exponential moving average (EMA) variants of diffusion models (Section A.1);

- *Diffusion models with different backbones*, such as MLPs and a diffusion transformer (Section A.1.1);

- *Flow matching models* using either the uniform schedule, various optimal transport settings (Section A.1.1) or a power-law noise schedule;

- *Consistency models*, including both discrete and continuous formulations;

- *Various samplers*, including multiple ODE and SDE solvers (where applicable) and annealed Langevin dynamics and predictor-corrector (PC) sampling (Section A.1.1 for more details).

This setting allows us to isolate the impact of architectural, parameterization, noise-schedule, and solver choices under conditions where simulation cost is negligible and model training is unconstrained.

Our analysis commences with the widely used set of ten benchmark models from Lueckmann et al. (2021). These models span a range of low parameter and observation dimensionalities; for detailed descriptions, we refer to the original study by Lueckmann et al. (2021). Because simulations are inexpensive, and all problems are low-dimensional, we train each model for 1,000 epochs, generating 30,000 new simulations per

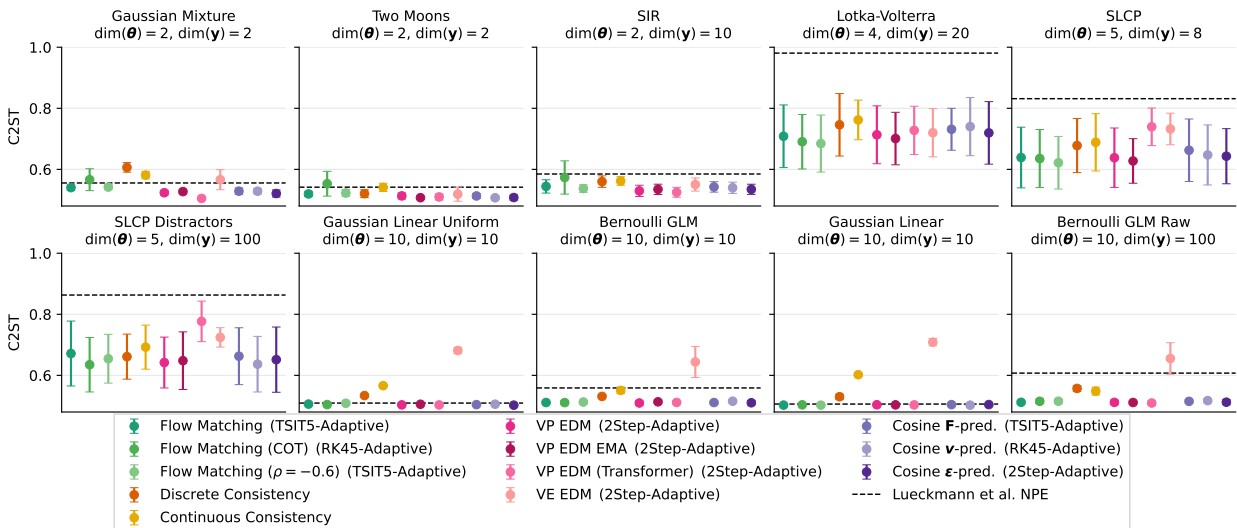

Figure 11: *Comparison of inference performance across design choices on the benchmark by Lueckmann et al. (2021) (Case Study 1).* We report mean and standard deviation of the C2ST, comparing samples from the diffusion model and the ground truth posterior samples for the 10 datasets from the benchmark (C2ST of 0.5 means that approximate and reference posterior are indistinguishable). As a baseline, we report the mean C2ST for the normalizing flow model in Lueckmann et al. (2021) trained on a budget of 100,000 simulations, showing how difficulty varies across problems.

epoch. All neural networks, if not otherwise stated, are MLPs with residual layers and FiLM conditioning for the time embedding without a separate summary network (Section A.1). To assess performance, we evaluate each trained model on the ten held-out simulated observations and compare the inferred posteriors with the benchmark "ground-truth" posteriors. We report the classifier two-sample test (C2ST) for a direct comparison with benchmark baselines (see Section A.1 for a definition).

**Noise schedules and weighting**  Diffusion models with a variance-preserving EDM noise schedule achieved the best overall performance, closely followed by cosine-based schedules (Figure 11). In contrast, the variance-exploding EDM schedule performed worse on several tasks, often producing higher C2ST scores. In terms of weighting, EDM seems to have a slight edge over sigmoid and likelihood weighting (Figure A.1).

**Parameterizations**  Across tasks, differences among the **F**-, $v$-, and $\epsilon$-parameterizations for the respective optimal sampler were generally small (Figure 11). However, the $\epsilon$-parameterization seems to be much more sensitive to the choice of the sampler (Figure A.1). EMA smoothing did not yield systematic improvements. In most cases, EMA and non-EMA variants performed similarly.

**Flow matching**  Flow matching models were competitive on all benchmarks but showed a slight overall performance drop on the smallest problems compared to diffusion models (Figure 11). Flow matching and the conditional optimal-transport (COT) variant performed comparably with no clear trend, whereas we find that ignoring the condition for computing the OT plan tends to harm accuracy (Figure A.2). Similar to previous studies (Wildberger et al., 2023; Orsini et al., 2025a), the power-law noise schedule with $\rho = -0.6$ improved performance in most tasks, especially for the more difficult Lotka-Volterra task.

**Consistency models**  Both discrete and continuous consistency models were competitive with diffusion models, but with a slightly lower accuracy while offering substantially faster inference, even though multi-step sampling was used (Figure A.1). There was no clear trend to suggest that discrete or continuous models perform better (Figure 11).

**Samplers**  Across diffusion and flow matching models, the adaptive two-step SDE solver was both the fastest and the most accurate (Figure 11). Performance differences among ODE and SDE solvers were small, except for the $\epsilon$-parameterization, which shows more variability (Figure A.1). However, annealed Langevin sampling did not reach the accuracy of ODE or SDE solvers, even with 4 times as many steps, and the predictor–corrector scheme also did not improve C2ST scores (Figure A.1). For ODEs, adaptive higher-order methods consistently outperformed Euler: they were faster and achieved lower error (but with only minor differences) by taking larger, fewer steps, with TSIT5 showing a slight advantage over RK45. Step rejection rates in the adaptive methods aligned with trajectory smoothness: flow matching trajectories did not trigger rejections; EDM-based diffusion models triggered fewer than ten; and cosine-schedule models typically caused ten to twenty.

**Diffusion backbone**  Under the same variance-preserving EDM configuration, the diffusion transformer backbone and residual MLP performed similarly on most benchmarks, with the transformer achieving the best overall result on the Gaussian mixture task. However, its gains over the MLP were generally small, and it was less accurate on the two SLCP tasks. It was also roughly $3.5\times$ slower in both training and inference.

**Recommendations for practice**  For low-dimensional problems, we recommend using diffusion models with a variance-preserving EDM noise schedule with $\mathbf{F}$ parameterization or flow matching, as both achieve comparable accuracy, while the latter is easier to implement and has fewer hyperparameters to tune. Across problems and samplers, these emerge as the most reliable choices. For some problems where a variance-preserving schedule does not achieve the desired contraction in the posterior, a variance-exploding noise schedule might be nevertheless beneficial. Moreover, we recommend the usage of adaptive solvers and, in particular, the two-step adaptive SDE solver in light of faster and more accurate inference. Only the consistency models are faster at the cost of potentially reduced accuracy.

Based on our results, we discourage the use of annealed Langevin sampling or predictor-corrector sampling, as they are harder to tune and can lead to poor results. Further, in these low-dimensional settings, attention via a diffusion transformer offered no clear benefit over a simple MLP. In higher-dimensional settings, however, transformer architectures may become more advantageous, particularly when masking is required to learn arbitrary conditionals, as in Section 4.3.3.

## 6.2 Case Study 2: Large-Scale ODE Benchmark

**Setting**  To evaluate the capabilities of diffusion models for SBI in actual practice, we consider a mechanistic model from the PEtab collection of ODE models (Hass et al., 2019). As a representative high-dimensional benchmark, we consider the Beer et al. (2014) model, which represents a quantitative description of the enzymatic reactions involved in indigoidine biosynthesis (Figure 12A). The model includes experimental condition specific parameters for bacterial growth ($\beta$, $\text{Bac}_{\text{max}}$ and time lag $\tau$), conversion of glutamine (Gln) to cyclized glutamine (cGln) ($k_{\text{syn}}$), dimerization of cGln to indigoidine (Ind) ($k_{\text{dim}}$) and degradation of indigoidine ($k_{\text{deg}}$). The model comprises 19 experimental conditions with more than $\dim(\mathbf{y}) = 27{,}000$ measurements (Figure 12B) and $\dim(\boldsymbol{\theta}) = 72$ parameters. It is one of the most data-rich ODE models in the collection, making it particularly suitable for testing the capacity of inference algorithms to handle large parameter spaces (see Section A.1.2 for more details).

**Performance evaluation**  Model performance was assessed on 1,000 datasets with 10 repeated training runs using expected calibration error, contraction, normalized root mean squared error (NRMSE), test accuracy with random points (TARP) and C2ST scores. For C2ST, the classifier is trained on posterior samples together with the log-likelihood as a data summary and compared to samples from the prior (Yao & Domke, 2023). For a single scalar ranking across methods, we ranked the sum of the empirically standardized metrics (see Section A.1 for a definition of the metrics).

For this experiment, diffusion models are paired with a summary network that embeds the high-dimensional observations before passing the representation into a concatenation of MLPs. Each inference method was evaluated with two different summary networks for time series; implementation details appear in Section A.1.2.

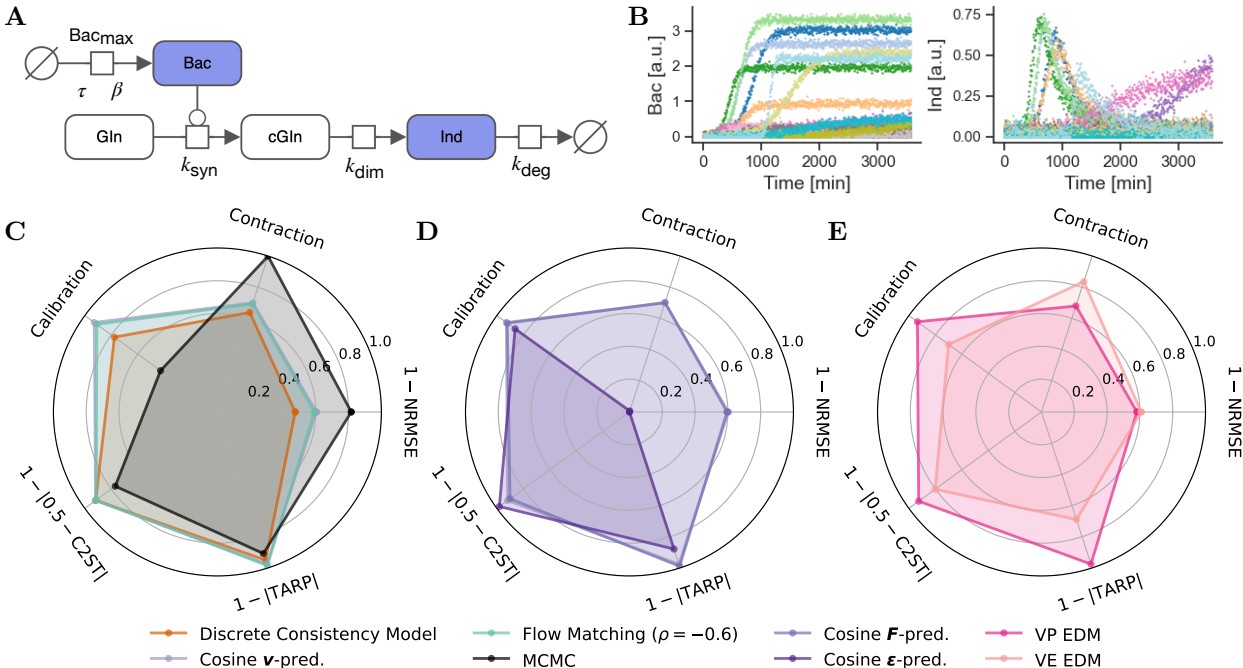

Figure 12: *Case Study 2: Comparison of inference performance across design choices.* **A** Visualization of the reactions described by the Beer et al. (2014) model. The parameters can vary per experimental conditions and some also for repeated experiments, leading to 72 parameters to be estimated. Blue indicates observed states. **B** Model simulation (with noise). **C** Best performing model from each family. We also show MCMC as a baseline. All metrics were scaled such that the optimal value would be 1 (only contraction has no optimal value) (Section A.1). **D** Diffusion models with cosine noise schedules and different parameterizations: **F**, **v**, and **ε** using the ODE sampler. **E** Diffusion models with variance-preserving and exploding EDM noise schedules using the ODE sampler.

**Results**  Across both summary networks, several design choices consistently led to improved accuracy and robustness (Figure 12). MCMC convergence is difficult to achieve for this problem because the parameter space is hard to explore. This is reflected in high contraction and very low NRMSE, but these gains come at the expense of calibration. Flow matching with a uniform noise schedule and its optimal transport variant achieved the lowest NRMSE, exhibiting low error and stable behavior. Diffusion models with cosine or EDM schedules in the **v**- and **F**-parameterizations, using either ODE or SDE samplers, performed similarly well, with marginally higher NRMSE but slightly better calibration according to expected calibration error and TARP. In contrast, the VE EDM schedule produced either excessive contraction with the ODE sampler or higher NRMSE with the SDE sampler, and suffered from increased calibration error in both settings (Table 4). This agrees with the behavior observed in the low-dimensional benchmarks.

C2ST was less informative in this high-dimensional experiment, even when using log-likelihood summaries, suggesting that it does not reliably discriminate between posterior approximations in this regime. In particular, when the update from prior to posterior was small, as for the **ε**-prediction models, C2ST alone was not a reliable measure of posterior accuracy. By contrast, settings with larger expected calibration error generally also have larger absolute TARP error. The relative ranking of inference methods was stable across repeated runs, indicating that the observed performance differences were driven primarily by the inference algorithms rather than random variation across training runs.

Training times were similar across most methods, only OT flow matching took 1.5 as long due to the added computation of optimal couplings; few-step inference with the consistency models was substantially faster, requiring only a handful of network evaluations compared to more than 100 for the diffusion-based approaches. Still, complete training and inference on 1,000 datasets required less wall-clock time than MCMC on a single

Table 4: *Case Study 2: Inference performance of diffusion models with their respective samplers on 1000 datasets.* Metrics: normalized RMSE (NRMSE), posterior calibration error, posterior contraction, classifier two-sample test (C2ST), and tests of accuracy with random points (TARP); see Section A.1 for details. The models are ranked by the sum of all empirically standardized metrics. Numerical values represent the mean (standard deviation) over 10 runs.

| Family | Design Choice | Sampler | NRMSE | Calibration Error | Contraction | C2ST | TARP | Rank |
|---|---|---|---|---|---|---|---|---|
| Diffusion | EDM, VP, **F** | ODE | 0.42 (0.005) | **0.03 (0.001)** | 0.68 (0.004) | 0.57 (0.008) | −0.02 (0.005) | 1 (2) |
| | Cosine, VP, **v** | SDE | 0.39 (0.005) | 0.04 (0.003) | 0.70 (0.003) | 0.59 (0.016) | −0.02 (0.004) | 3 (2) |
| | Cosine, VP, **v** | ODE | 0.40 (0.005) | 0.04 (0.003) | 0.70 (0.004) | 0.58 (0.025) | −0.02 (0.004) | 4 (4) |
| | EDM, VP, **F** | SDE | 0.40 (0.005) | 0.04 (0.002) | 0.70 (0.004) | 0.57 (0.015) | −0.03 (0.004) | 5 (3) |
| | Cosine, VP, **F** | ODE | 0.40 (0.003) | 0.04 (0.002) | 0.70 (0.002) | 0.60 (0.016) | **−0.01 (0.003)** | 6 (3) |
| | Cosine, VP, **F** | SDE | 0.39 (0.003) | 0.04 (0.001) | 0.71 (0.002) | 0.60 (0.017) | −0.02 (0.003) | 9 (3) |
| | Cosine, VP, $\epsilon$ | SDE | 0.52 (0.129) | 0.04 (0.001) | 0.62 (0.122) | 0.61 (0.020) | −0.05 (0.006) | 12 (2) |
| | EDM, VE, **F** | SDE | 0.67 (0.024) | 0.10 (0.003) | 0.51 (0.028) | 0.63 (0.013) | 0.15 (0.005) | 16 (1) |
| | EDM, VE, **F** | ODE | 0.39 (0.015) | 0.15 (0.002) | 0.84 (0.006) | 0.70 (0.010) | −0.31 (0.012) | 17 (2) |
| | Cosine, VP, $\epsilon$ | ODE | 1.38 (0.124) | 0.07 (0.005) | 0.01 (0.006) | **0.52 (0.005)** | 0.12 (0.006) | 18 (1) |
| Diffusion (EMA) | EDM, VP, **F** | ODE | 0.42 (0.006) | 0.03 (0.002) | 0.68 (0.005) | 0.57 (0.010) | −0.03 (0.006) | 2 (2) |
| | EDM, VP, **F** | SDE | 0.40 (0.005) | 0.04 (0.003) | 0.70 (0.004) | 0.59 (0.018) | −0.04 (0.003) | 8 (4) |
| Flow Matching | $\rho = -0.6$ | ODE | 0.41 (0.005) | 0.05 (0.004) | 0.69 (0.005) | 0.59 (0.016) | 0.01 (0.005) | 7 (2) |
| | Uniform | ODE | 0.39 (0.003) | 0.07 (0.002) | 0.77 (0.005) | 0.59 (0.016) | −0.06 (0.006) | 10 (2) |
| | OT | ODE | 0.39 (0.003) | 0.07 (0.002) | 0.76 (0.004) | 0.59 (0.010) | −0.05 (0.004) | 11 (1) |
| Consistency | Discrete | ODE | 0.53 (0.006) | 0.11 (0.067) | 0.64 (0.010) | 0.58 (0.072) | 0.05 (0.051) | 13 (2) |
| | Continuous | ODE | 0.70 (0.010) | 0.14 (0.004) | 0.49 (0.013) | 0.52 (0.010) | 0.15 (0.003) | 15 (1) |
| MCMC | - | - | **0.18 (0.004)** | 0.29 (0.002) | 1.00 (0.000) | 0.73 (0.026) | -0.09 (0.012) | 14 (0) |

dataset (under one day versus roughly three days), which fails to adequately explore the parameter space and hence attains poor calibration.

Both consistency variants showed higher NRMSE and calibration error, and the continuous consistency model in particular exhibited weaker contraction, suggesting that it produced posterior approximations that were less concentrated around the target than the competing methods. The discrete consistency model performed somewhat better, but its calibration error and TARP estimates were also more variable across runs. Using an exponential moving average (EMA) did not improve performance for this task (Section A.1).

In terms of parameterizations, the **v**- and **F**-parameterizations proved consistently stable, while $\epsilon$ prediction was not robust and even produced divergent trajectories for 5% of the samples. Regarding samplers, higher-order ODE methods generally yielded the best results, but in some instances SDE solvers improved accuracy, for example under $\epsilon$ prediction.

**Recommendations for practice**  In high-dimensional settings, the picture changes. Consistency models are no longer competitive with the other model families due to worse NRMSE and calibration. Flow matching and diffusion models were the most reliable choices in this experiment, but they emphasized slightly different aspects of performance. Flow matching achieved the lowest NRMSE, although the improvement over the best diffusion models was marginal. Diffusion models, in turn, achieved slightly better calibration. Among the diffusion variants, the VP noise schedule combined with either the **F**- or **v**-parameterization remained the most stable configuration. Direct noise prediction with $\epsilon$ was prone to instability and should be avoided.

### 6.3 Case Study 3: Gaussian Random Fields as a Scalable SBI Benchmark

**Setting**  Gaussian random fields (GRFs) can model spatial uncertainty when the quantity of interest is a latent field rather than a small parameter vector (Figure 13C). Under the assumptions of homogeneity and isotropy, a GRF is fully described by its mean and covariance, or equivalently, by a power spectrum, allowing complex spatial patterns to be summarized by a few interpretable parameters, such as overall variance and correlation structure. This makes GRFs a standard prior in applications where spatially correlated properties are only indirectly observed, including subsurface permeability fields in hydrology (Chen et al., 2025a), heterogeneous material properties in computational geosciences (Liu et al., 2019), or the primordial matter distribution in cosmology (Wandelt, 2012). In these domains, uncertainty quantification typically focuses on

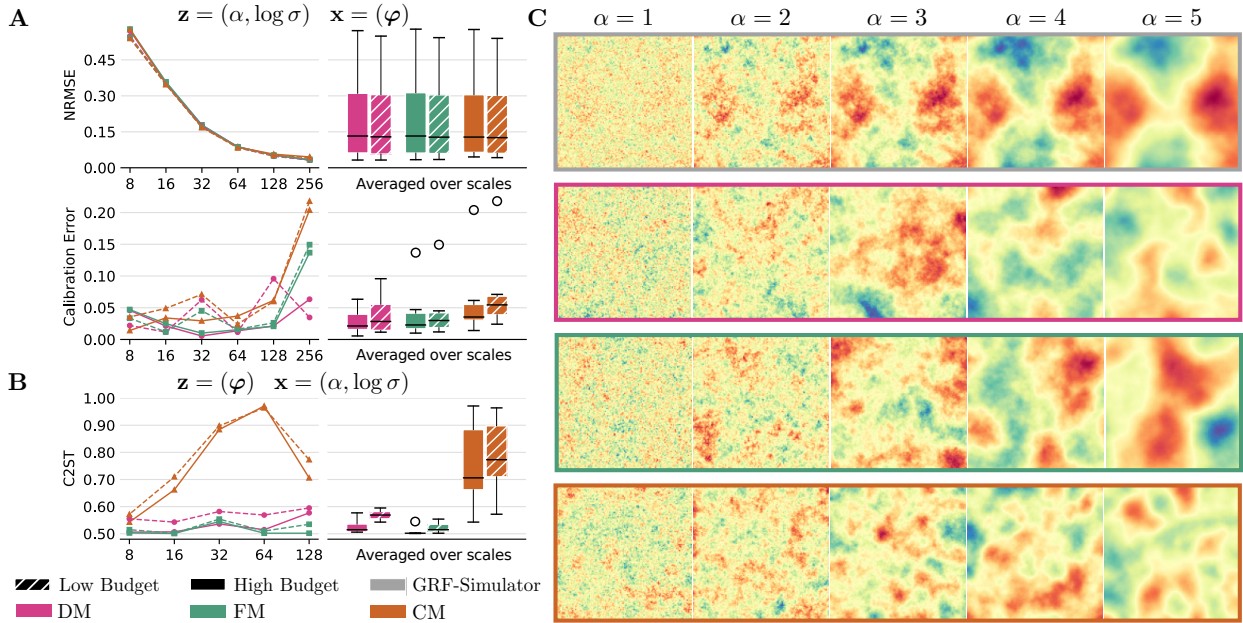

Figure 13: *Case Study 3: Gaussian random field benchmark across parameter dimensionalities, resolutions, and budgets.* We compare a variance-preserving EDM-based diffusion model (DM), flow matching (FM), and a consistency model (CM) on the GRF case study across resolutions and low-/high-budget training. Line plots show the respective metrics as a function of field size $r$, with box plots summarizing performance aggregated over all scales; in all plots, different colors denote DM, FM, and CM, while solid vs. hatched markers indicate high vs. low simulation budget. **(A)** Low-dimensional parameter case with $\mathbf{z} = (\alpha, \log \sigma)$ and $\mathbf{x} = (\boldsymbol{\varphi})$: NRMSE (top) and calibration error (bottom). **(B)** High-dimensional parameter case with $\mathbf{z} = (\boldsymbol{\varphi})$ and $\mathbf{x} = (\alpha, \log \sigma)$: C2ST accuracy (values near 0.5 indicate that generated and simulated samples are difficult to distinguish). **(C)** Qualitative GRF samples at $128 \times 128$ resolution for fixed $\log \sigma$ and varying $\alpha \in \{1, \dots, 5\}$ after high budget training: the top row shows the true GRF simulator, and the subsequent rows show DM, FM, and CM, respectively. DM and FM closely track the change in dominant structure size with $\alpha$, while CM exhibits visible mismatches near the edges of the prior over $\alpha$.

either low-dimensional parameters governing the spectrum of fluctuations or high-dimensional latent fields, often under low simulation budgets.

Our GRF benchmark is a controlled setting in which observation and target dimensionality can be scaled independently within a fixed simulator family and designed to answer three questions: (i) how different is performance between a variance-preserving EDM-based diffusion model (DM), flow matching (FM), and a consistency model (CM) in the low-dimensional target regime where the condition is a high-dimensional field; (ii) which methods remain reliable when the target is switched and the task becomes to generate a high-dimensional field, and (iii) how does performance change as a function of simulation budget.

We generate 2D GRFs with a power-law spectrum $P(k) \propto k^{-\alpha} \sigma$ using FyeldGenerator (Cadiou, 2022), and consider field resolutions $r \in \{8, 16, 32, 64, 128, 256\}$. The parameters are drawn from Gaussian priors

$$\log \sigma \sim \mathcal{N}(0, 0.3^2) \quad \text{and} \quad \alpha \sim \mathcal{N}(3, 0.5^2),$$

and collected in $\mathbf{z} = (\alpha, \log \sigma)$. We alternate between a low-dimensional case, where the task is to infer $\mathbf{z}$ from a single field, and a high-dimensional case, where the goal is to generate fields $\mathbf{z} = \boldsymbol{\varphi} \in \mathbb{R}^{r \times r}$ conditioned on $\mathbf{x} = (\alpha, \log \sigma)$. For each resolution, we train three model families: a variance-preserving EDM-based diffusion model, flow matching, and a consistency model; in two simulation regimes: an offline regime with 500 epochs that iterates over 5000 fixed simulations, and an online regime with 500 epochs and 5000 new simulations per epoch (totaling 2.5M instances). Together, these configurations span low- and high-dimensional parameters

and observations under both low and high simulation budgets, in line with the archetypes summarized in Table 2. Notably, the smallest high-dimensional parameter case (field of size $8 \times 8$) is roughly of the same size as the number of parameters (72) in the previous case study (Section 6.2).

In the low-dimensional case, a resolution-dependent residual convolutional summary network is employed to encode the field into an 8-dimensional summary, along with a fixed three-layer MLP diffusion backbone. Diffusion backbone and summary network are trained jointly end-to-end. In the high-dimensional case, the full field is treated as the target, and the scalar conditions are tiled to match the field resolution. In this case, a U-Net-style architecture is employed as the diffusion backbone, whose depth scales with the field resolution while the channel width remains fixed. In both cases, architectures are the same across methods at a given resolution; only the training objective differs.

In the high-dimensional setting, additional backbone comparisons are provided in Section A.1.3, where we report results for UViT and Residual UViT diffusion backbones (Hoogeboom et al., 2023; 2025). Performance in the low-dimensional case is evaluated using NRMSE and calibration error. In the high-dimensional case, we assess sample quality using a C2ST, estimated with a classifier whose architecture matches that of the summary network, but with a one-dimensional output. Further architectural, training, and simulator details are given in Section A.1.3.

**Results**  In the low-dimensional case, where $\mathbf{z} = (\alpha, \log \sigma)$ are inferred from a single field, all three families achieve similar point estimation accuracy across scales. The NRMSE curves in Figure 13A show only mild trends: CM is slightly better at coarse resolutions, FM is competitive in the middle range, and DM tends to perform best at the largest grids, but differences remain small, and we observe no systematic benefit of online over offline training for NRMSE. NRMSE is highest at the coarsest resolutions, which is consistent with the limited information content of small fields: an $8 \times 8$ or $16 \times 16$ realization contains far fewer Fourier modes, so empirical estimates of the spectrum and variance are noisier, making $(\alpha, \log \sigma)$ inherently harder to identify from a single draw.

As the resolution increases, each field provides more modes and better-averaged statistics, allowing the summary network to more reliably extract the relevant features, which in turn leads to a lower overall NRMSE. Calibration error reveals a clearer hierarchy: at low resolutions, online-trained CMs yield the best-calibrated posteriors, whereas at higher resolutions, DM is the most robust choice, with FM typically in second place and CMs exhibiting noticeable miscalibration. Online training enhances calibration for all methods once the resolution exceeds $16 \times 16$, with online DM yielding the most stable performance across scales.

In the high-dimensional case, where the whole field $\boldsymbol{\varphi}$ is treated as the target to be generated given $(\alpha, \log \sigma)$, we assess sample quality via C2ST accuracy (Figure 13B). Because C2ST is highly sensitive, performance is effectively "hit or miss": values near 50% indicate that generated and simulated samples are difficult to distinguish, while higher values reflect detectable discrepancies. CM behaves worst in this regime: it performs reasonably at low resolution, but its C2ST accuracy deteriorates rapidly with increasing grid size, and online training further degrades its performance. For the largest resolution C2ST improves again, as the classification problem for the classifier becomes much harder.

By contrast, FM and DM remain robust across scales and clearly benefit from online training, which reliably improves the C2ST score compared to the offline regime. Across scales (box plots), FM and DM both benefit from the higher simulation budget, with DM showing the largest gap between online and offline training. However, this gab vanishes for other backbones (Figure A.3). Qualitative samples in Figure 13C support these findings: for fixed $\log \sigma$ and varying $\alpha$, both DM and FM closely track the change in dominant structure size seen in the GRF simulator outputs, whereas CM tends to struggle near the edges of the prior over $\alpha$, producing fields that are too coarse at small $\alpha$ and too fine at large $\alpha$ relative to prior predictions.

**Recommendations for practice**  The new GRF benchmark with controllable dimensionality suggests that for low-dimensional inference, DM, FM, and CM achieve comparable NRMSE, but DM is the most reliable at higher resolutions, particularly with respect to calibration. For high-dimensional conditional generation, DMs and flow matching clearly outperform CMs and benefit from online training, translating additional

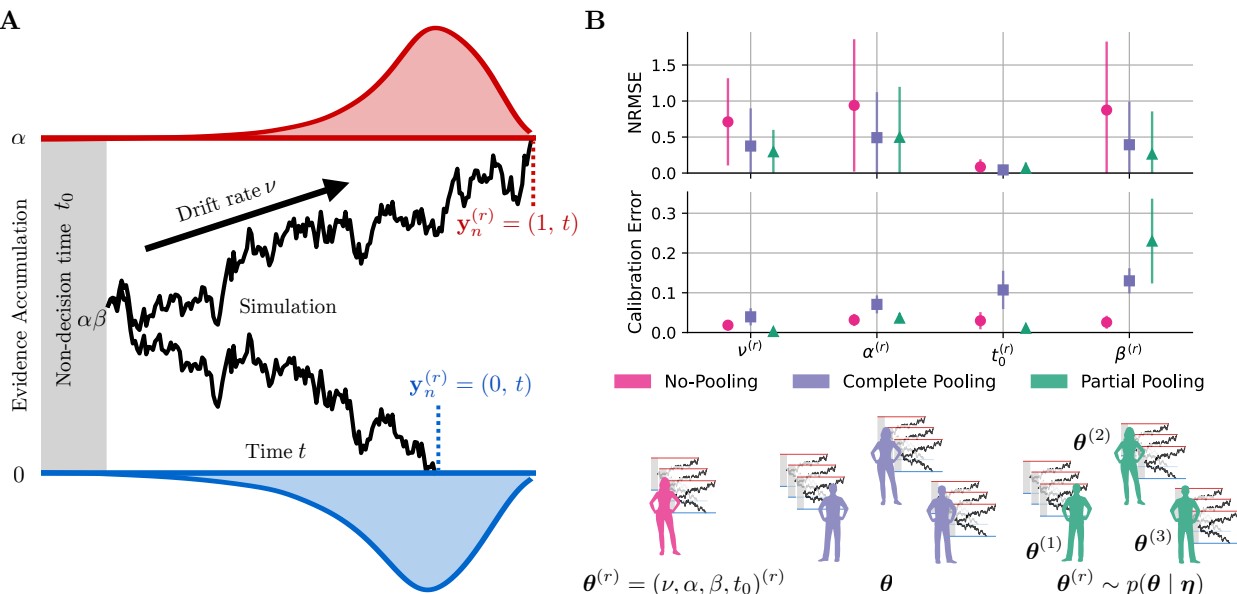

Figure 14: *Case Study 4: Pooling regimes for a cognitive model.* **A** Visualization of the simulator and the corresponding parameters. **B** NRMSE and calibration error (median and median absolute deviation) for the different pooling regimes.

simulation budget into higher sample fidelity. CMs are attractive when inference-time constraints dominate. Their sampling speed is orders of magnitude faster in our experiments, making them a viable option for coarse-resolution or latency-critical applications, albeit at the cost of reduced accuracy at larger scales.

### 6.4 Case Study 4: Pooling Regimes in Cognitive Modeling

**Setting**   In our final case study, we illustrate the utility of compositional inference for different structured Bayesian models (Section 4.2). Our simulator is an evidence accumulation model (EAM) (Ratcliff et al., 2016), widely used to study decision-making and recently used to benchmark amortized hierarchical inference (Habermann et al., 2025). The EAM describes binary decision-making as the accumulation of noisy information until one of two boundaries is reached (Figure 14A). Each trial is defined by four parameters: drift rate $\nu$, decision threshold $\alpha$, non-decision time $t_0$, and starting bias $\beta$. We simulate latent trajectories using the Euler-Maruyama scheme with absorbing boundaries at 0 and $\alpha$. The simulation returns both a binary choice and a reaction time (RT), with trials censored at $t = \max_t$ if no boundary is hit (Section A.1.4).

For this case study, we have two different priors: a flat and a hierarchical one. For each, we train a diffusion model with EDM noise schedule. For training, we simulate training instances with $R = 1$ subject and $N = 30$ trials per subject. Hence, as a summary network, we employ a `SetTransformer` (Lee et al., 2019) to account for the exchangeability of the trials. For inference, we simulate 100 test sets of $R = 100$ subjects, each with $N = 30$ trials. Thus, for the same subject, we have repeated observations of trials generated by the same parameter and, depending on the prior, parameters between subjects are assumed to be the same (*complete pooling*) or linked by a hierarchical structure (*partial pooling*). Details of the simulator and priors are given in Section A.1.4.

**No-pooling**   We first estimate subject-level parameters $p(\boldsymbol{\theta}^{(r)} = (\nu^{(r)}, \log \alpha^{(r)}, \log t_0^{(r)}, \beta^{(r)}) \mid \mathbf{y}^{(r)})$ independently. With limited information, posteriors remain broad but calibrated, exhibiting a high NRMSE (Figure 14B). These findings reflect the intrinsic difficulty of inferring subject-specific parameters from low trial numbers without any pooling.

**Complete pooling**   Complete pooling assumes that all subjects share a single parameter vector. Compositional score matching reuses the same model trained on single subjects and aggregates information across $R$

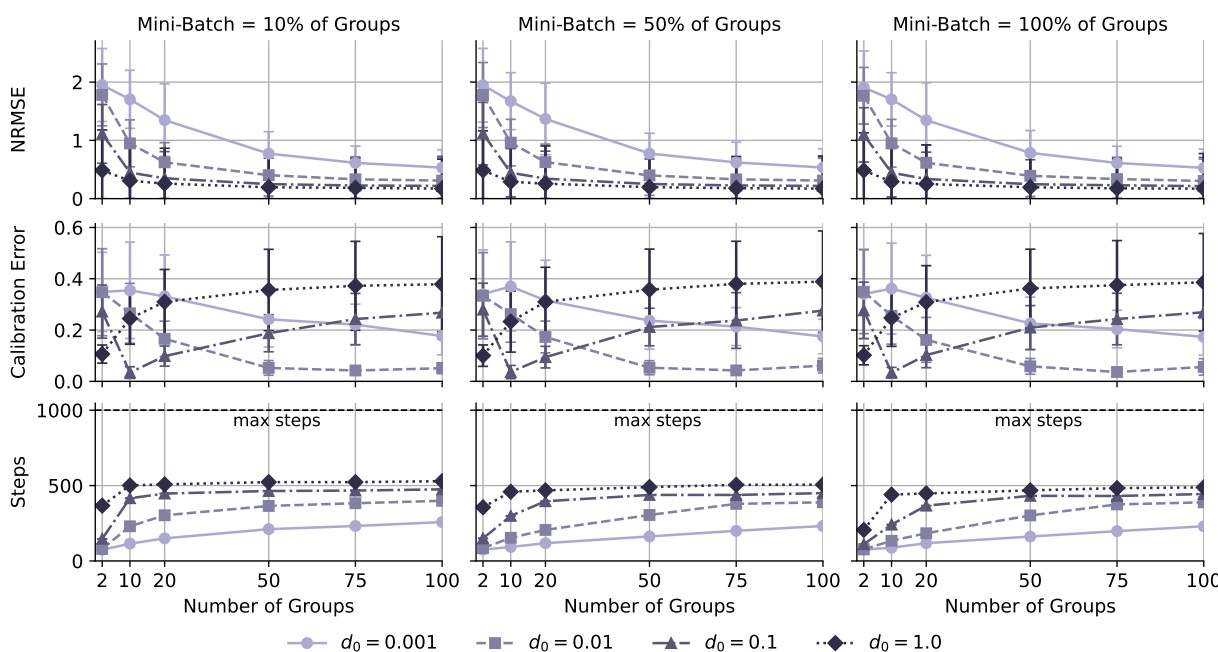

Figure 15: *Case Study 4: Scaling behavior of compositional sampling.* NRMSE, calibration error (mean over parameters), and integration adaptive steps for complete pooling via compositional sampling versus the number of groups, across mini-batch sizes and damping factors $d_0$.

subjects:

$$\nabla_{\boldsymbol{\theta}_t} \log p(\boldsymbol{\theta}_t \mid \{\mathbf{y}_{\text{obs}}^{(r)}\}_{r=1}^R) \approx d(t) \cdot \left((1-R)(1-t)\nabla_{\boldsymbol{\theta}_t} \log p(\boldsymbol{\theta}_t) + \frac{R}{M} \sum_{m=1}^M \hat{s}(\boldsymbol{\theta}_t, \mathbf{y}_{\text{obs}}^{(m)}, t)\right). \tag{77}$$

We start by sampling $\boldsymbol{\theta}_1 \sim \mathcal{N}(\mathbf{0}, \mathbf{I})$, and then solving the reverse SDE, where we sample a mini-batch of $M$ datasets per reverse step to reduce the computational costs and set $d(t)$ to have $d(0) < 1$ in order to improve calibration following (Arruda et al., 2026). This model yields a sharper global posterior by *training only on single-subject simulations.* As shown by the NRMSE in Figure 14B, additional evidence across subjects substantially improves estimation. In contrast, direct training without composition would require $R$ times as many simulations per training instance and would not scale to large data sets.

**Partial pooling**  Hierarchical models interpolate between no pooling and complete pooling by introducing both global and local parameters. Two models are trained on single-subject simulations: a global model predicts group-level means $\boldsymbol{\mu}$, variances $\boldsymbol{\sigma}$, and a shared $\beta$, while a local model predicts subject-specific parameters $\boldsymbol{\theta}^{(r)} = (\nu^{(r)}, \log \alpha^{(r)}, \log t_0^{(r)})$ conditional on those global quantities $\boldsymbol{\eta} = (\boldsymbol{\mu}, \boldsymbol{\sigma}, \beta)$. During training, the local model conditions on the true global parameters and for inference, we perform ancestral sampling: (1) draw samples from the global parameters via compositional diffusion and then (2) sample individual-level parameters conditioned on the global samples. Correspondingly, the following compositional score is approximated:

$$\nabla_{\boldsymbol{\eta}_t} \log p(\boldsymbol{\eta}_t \mid \{\mathbf{y}_{\text{obs}}^{(r)}\}_{r=1}^R) \approx d(t) \cdot \left((1-R)(1-t)\nabla_{\boldsymbol{\eta}_t} \log p(\boldsymbol{\eta}_t) + \frac{R}{M} \sum_{m=1}^M \hat{s}_{\text{global}}(\boldsymbol{\eta}_t, \mathbf{y}_{\text{obs}}^{(m)}, t)\right). \tag{78}$$

The local model does not need to allow compositional inference, hence we can employ a continuous consistency model for fast sampling of $p(\boldsymbol{\theta}_t^{(r)} \mid \mathbf{y}_{\text{obs}}^{(r)}, \boldsymbol{\eta})$. As expected, hierarchical estimation produces shrinkage toward the group means and leads to an overall lower RMSE compared to no- or complete pooling (Figure 14B).

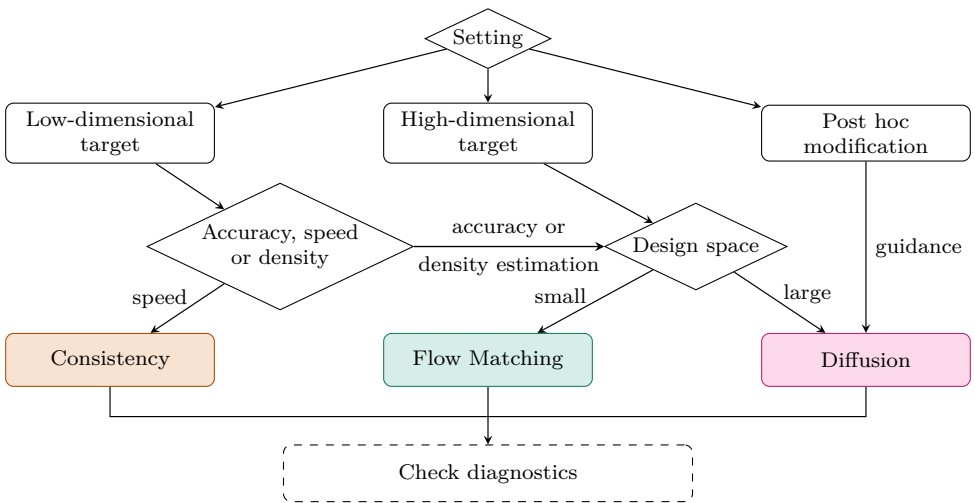

Figure 16: *Practical guide for using diffusion models in SBI.* The recommendations are based on empirical trends from our case studies and should serve as heuristics, not as rules. Low- and high-dimensional targets refer to the dimensionality of the inferred quantity. The final model should be checked with comprehensive Bayesian diagnostics.

**Scaling behavior of compositional sampling**  To analyze the behavior of compositional sampling, we varied the number of groups, the mini-batch size, and the damping factor $d_0$ (Figure 14C). Adaptive SDE integration is necessary because the number of solver steps increases with the number of composed groups. By contrast, the mini-batch size has almost no influence on NRMSE, calibration, or integration cost, indicating that subsampling groups at each reverse step is sufficient to capture the difference between individual scores. The damping factor $d_0$ has a stronger effect as smaller values can improve calibration at the cost of sharpness (NRMSE), although overly strong damping eventually degrades calibration as well. The optimal choice of $d_0$ therefore depends on the number of groups being composed, but importantly, it can be selected at inference time without retraining the model.

**Recommendation for practice**  Compositional inference provides novel means for leveraging the probabilistic structure of a Bayesian model to increase simulation and training efficiency. Yet, it requires tuning on a test data set as the compositional estimator has an accuracy-calibration trade-off, and more groups generally need more damping, but too much damping can also harm calibration. Since all diffusion models reuse the same simulation budget as in the no-pooling setting but scale to datasets involving hundreds of groups, compositional inference can become a flexible framework for large-scale hierarchical Bayesian analysis.

## 7   Discussion

This tutorial review synthesized recent developments in diffusion models for simulation-based inference (SBI), covering special use cases of the score and design choices for training and inference. Our analysis of noise schedules, parameterizations, and samplers reveals best practices and areas requiring further investigation. Next, we discuss key findings from our empirical evaluations (Figure 16), identify critical open questions, and outline promising future avenues.

**Empirical considerations**  Our empirical evaluations across the first three case studies reveal that design choices matter substantially, but their impact is dimension- and simulation-budget dependent. For the low-dimensional benchmark problems (*Case Study 1*, see Section 6.1), diffusion models with a variance-preserving EDM schedule or flow matching models were the best choices, while flow matching has fewer hyperparameters to tune and is therefore easier to implement. For high-dimensional inference (*Case Study 2*, see Section 6.2), diffusion models achieved the best ranking because of stronger calibration, while flow matching achieved the lowest NRMSE and remained competitive overall, while the variance-exploding EDM schedule exhibited

higher calibration errors in this setting. These results align with prior observations that variance-preserving schedules generally perform better than VE schedules for high-dimensional targets (Song et al., 2021c). However, not only dimensionality plays a role for choosing a specific design, but also the available simulation budget, making it hard to define a specific boundary where one model family performs better than the other one.

We observe analogous trends in the GRF experiment (*Case Study 3*, see Section 6.3). In the low-dimensional setting, variance-preserving EDM-based diffusion and flow-matching usually perform best, followed closely by consistency models. In the high-dimensional setting, flow matching seems to be less sensitive to the available simulation budget than the diffusion model. However, for more powerful transformer-based backbones this difference vanishes. By contrast, the consistency model performs poorly across simulation budgets, scales and backbones in this setting. This presumably relates to the additional burden of approximating the denoising trajectory in consistency models under a fixed backbone, which appears particularly challenging when the output is high-dimensional.

Critically, we found that direct noise prediction with simple Euler schemes was prone to instability and should be avoided in favor of $\mathbf{v}$- or $\mathbf{F}$-parameterizations. The choice of sampler also proved consequential in terms of speed and accuracy: adaptive higher-order methods generally yielded superior accuracy. Interestingly, using optimal transport during training of flow matching (i.e., conditional optimal transport) seems not to pay off for low- or high-dimensional problems. Consistency models offered dramatically faster inference (few network evaluations vs. 100 or more for other approaches) with comparable NRMSE on low-dimensional targets but slightly elevated calibration error, corroborating a practical speed-accuracy tradeoff. For increasing dimensionality of the target discrete and even more so continuous consistency models show decreasing accuracy, making them less suitable for higher-dimensional tasks in SBI.

Moreover, in low-dimensional settings, attention-based diffusion transformers offered no clear advantage over a simple MLP. In contrast, transformer architectures appeared beneficial in higher-dimensional problems and for inferring arbitrary conditionals via masking, where attention performed best. This also suggests that future benchmarking in SBI should place greater emphasis on carefully matched experimental settings and strong baselines when assessing new models on low-dimensional problems.

A limitation of these empirical comparisons is that our case studies do not exhaust the range of possible simulator outputs or target spaces encountered in SBI. The examples considered here focus on scientifically common settings in which the simulator output is either a time series or an unordered set of observations. Other modalities, such as graph-structured observations, were not studied empirically. We expect that many such extensions would either require adapting the diffusion model beyond Euclidean spaces or adapting the summary network rather than changing the diffusion-based posterior estimation framework itself. For instance, just as a `SetTransformer` can encode unordered observations into a fixed-dimensional representation, graph neural networks could be used to encode graph-valued simulator outputs (Zhou et al., 2020; Hoogeboom et al., 2022). A second limitation is that all case studies considered continuous real-valued target parameters, although discrete or mixed discrete-continuous parameters arise naturally in model choice, latent structure inference, and hierarchical Bayesian analysis (Schröder & Macke, 2024; Kucharský & Bürkner, 2026). Extending diffusion-based SBI to these settings is therefore an important direction for future work, particularly in light of recent work on SBI with discrete and mixed parameter spaces (Ghiglino et al., 2026; Gloeckler et al., 2026; Boelts et al., 2026).

In practice, we recommend using diffusion models with the variance-preserving EDM noise schedule and an adaptive SDE solver or flow matching with an adaptive higher-order ODE solver, as these appear to be the best-performing models across SBI problems. Consistency models can be useful when inference latency dominates and moderate accuracy is acceptable, especially in low-dimensional settings. However, they should not be the default choice for high-dimensional SBI unless validated carefully, since they showed weaker calibration and sample quality in our high-dimensional experiments. Promising new directions include mean flows (Geng et al., 2026) for fast density evaluation (Rehman et al., 2026), and energy matching (Balcerak et al., 2025) and equilibrium matching (Wang & Du, 2025) models, which learn a time-independent flow field, thereby simplifying the architecture of the backbone and outperforming flow-based models in image generation quality.

**Structured targets**  Our analysis of pooling regimes (*Case Study 4*, see Section 6.4) demonstrates the substantial efficiency gains elicited by compositional score matching. For the target cognitive model, compositional estimation reduced the simulation budget by multiple orders of magnitude compared to naive training. Furthermore, compositional estimation is applicable to hierarchical Bayesian models, reducing uncertainty in local parameters. However, we also observed that naive compositional estimation can suffer from error accumulation, necessitating some form of error-damping or stabilization (Arruda et al., 2026).

These findings underscore that hierarchical Bayesian models—long espoused as desirable defaults in Bayesian analysis but computationally prohibitive in traditional SBI—can become tractable with compositional diffusion approaches. The ability to train separate models for global and local parameters, then combine them through ancestral sampling, opens new avenues for analyzing large-scale hierarchical data sets across domains from cognitive science to systems biology.

**Guided simulation-based inference**  While guidance offers remarkable flexibility for inference-time adaptation, such as prior adjustment without retraining, it also introduces new error sources and theoretical complications. Any guidance potentially biases the reverse diffusion process, and changing the score implicitly assumes that the density $p(\mathbf{z}_t \mid \mathbf{z}_0)$ of the reverse diffusion process changes accordingly. Current practice often employs guided scores with SDE or ODE samplers designed for unmodified processes, which can lead to marginal distribution mismatch.

The fundamental question remains: under what conditions does guidance preserve statistical accuracy? Recent work by Vuong et al. (2025) suggests that samples may align with the marginals implied by the guidance, even if intermediate trajectories differ from true reverse diffusion. However, extensive characterization of the conditions under which this holds represents a critical open problem. This is particularly pressing for compositional estimation (Geffner et al., 2023; Linhart et al., 2026; Gloeckler et al., 2025; Arruda et al., 2026; Touron et al., 2026), where score composition across many observations amplifies guidance-induced errors. Practitioners may assess whether guidance hyperparameters produce calibrated samples using calibration diagnostic metrics, and, if needed, further refine these hyperparameters through approaches such as inference-time Bayesian optimization targeting these diagnostics (Arruda et al., 2026).

**Dimensionality and simulation budgets**  Our case studies show that dimensionality and simulation budget jointly affect both posterior calibration metrics and sample-quality diagnostics such as C2ST. In the low-dimensional GRF setting (*Case Study 3*, see Section 6.3), where a low-dimensional spectrum parameter is inferred from high-dimensional fields, increasing the budget from the offline to the online regime consistently reduces calibration error, with variance-preserving EDM-based diffusion models showing the most robust improvements at higher resolutions.

For the 72-dimensional ODE model (*Case Study 2*), C2ST is surprisingly insensitive: even when we use log-likelihood values as sufficient statistics and restrict the approximating family, its accuracy stays close to chance in regimes where other metrics and qualitative inspection suggest nontrivial differences between posterior approximations. In the high-dimensional GRF generation task (*Case Study 3*, see Section 6.3), C2ST becomes more informative and reflects budget sensitivity. However, this budget sensitivity also depends on the model backbone: transformer-based architectures appear to be more efficient and scalable than U-Nets, which is consistent with the motivation for their use in image generation tasks (Peebles & Xie, 2023).

Taken together, these findings suggest that C2ST can be useful but fragile, and it also comes with additional design choices: one must decide how many and which samples to use for training the classifier. This is an expensive step in high-dimensional SBI, where sampling from multi-step generative models is costly, and the classifier architecture itself has to be adapted to the data structure, adding hyperparameters and complicating comparisons across studies. Hermans et al. (2022) showed that posterior approximators can be systematically overconfident in regions of low prior mass while appearing well calibrated globally, which is especially true in high-dimensions, where the region of the target usually has low prior mass. Developing calibration diagnostics that remain informative in high dimensions and can distinguish approximation error from misspecification represents an urgent methodological need. Recent methods such as posterior SBC (Säilynoja et al., 2026) and alternative test quantities for SBC (Lemos et al., 2023; Modrák et al., 2025) offer promising directions, but their computational cost and sensitivity in extreme dimensions require further improvements.

Even though diffusion models enable amortized inference with fixed simulation budgets, many scientific domains face severe constraints where generating even thousands of high-fidelity simulations remains infeasible (Ohana et al., 2024). In fields such as fluid dynamics, producing a high-fidelity pair ($\boldsymbol{\theta}, \mathbf{y}_{\text{obs}}$) can take multiple days, rendering simulation-based training hardly feasible. Our review highlights a critical infrastructure gap: most SBI papers still evaluate exclusively on low-dimensional benchmarks, and applications to high-dimensional parameters remain rare. The field would benefit enormously from expanded benchmark suites with controllable dimensionality regimes and simulation budgets.

**Model misspecification** Diffusion models offer unique flexibility through score composition and guidance, which may be leveraged to mitigate the effects of model misspecification. This is highly pertinent in real-world applications, which invariably involve some form of misspecification, exposing the underlying "Sim2Real" problem (Schmitt et al., 2023). Recent work explores robust inference through generalized Bayesian approaches (Gao et al., 2023), unsupervised domain adaptation (Elsemüller et al., 2025), iterative refinement for misspecified priors (Barco et al., 2025), or leverages (scarce) ground-truth measurements to correct for misspecification in flow matching (Ruhlmann et al., 2025). Yet, diffusion models remain largely unexplored in the context of misspecified models (Kelly et al., 2025). Can we design objectives that explicitly interpolate between proper and generalized Bayesian inference (Bharti et al., 2026)? How should we combine scores from multiple competing models to achieve better predictive performance? These questions connect to broader themes in Bayesian model averaging and ensemble methods but require rethinking in the SBI context.

## 8 Conclusion

Diffusion models have rapidly emerged as powerful and flexible tools for SBI, offering comparative advantages in expressiveness, stability, and modularity. Their score-based formulation provides unique capabilities for guidance, composition, and adaptation that harmonize with the needs for the next generation of SBI methods. Our tutorial review demonstrates that careful attention to design choices (i.e., noise schedules, parameterizations, samplers, and compositional strategies) can substantially impact both statistical accuracy and speed.

However, realizing the full potential of diffusion models in SBI requires addressing fundamental questions about posterior validity under guidance, developing robust calibration diagnostics for high-dimensional settings, and creating standardized benchmarks that reflect the true complexity of scientific applications. As the field matures, we anticipate that diffusion models will not simply serve as drop-in replacements for existing architectures but will enable fundamentally new inference paradigms, from real-time adaptive experimental design to causal discovery in complex dynamic systems and foundation models that generalize across models, designs, and domains.

The rapid pace of innovation in both generative modeling and scientific computing suggests that many of the open questions identified here will see progress in the coming years. By providing a conceptual source for understanding design choices, evaluation metrics, and application domains, we hope this review can accelerate progress in SBI and scientific modeling in general.

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

# A  Appendix

## A.1  Empirical Evaluation: Additional Information

**Setting**  All models and samplers were implemented in `BayesFlow` 2.0.12 Kühmichel et al. (2026). We use $z$-score normalization as a target normalization strategy, as it was shown to give superior performance compared to min-max scaling (Orsini et al., 2025b). In our EDM settings, we adopt $\sigma_{\text{target}} = 1$, as we standardize our target during training, and we set $\sigma_{\min} = 0.0001$ and use the EDM weighting. For the cosine schedule, we use no shift $s = 0$ and a variance-preserving schedule with sigmoid weighting. We set $\lambda_{\min} = -15$ and $\lambda_{\max} = 15$ and rescale the time accordingly. For the consistency models, we employ multi-step sampling during inference with 10 (discrete case) or 15 steps (continuous case). All models were trained using Adam (for online training) (Kingma & Ba, 2015) or AdamW (for offline training) (Loshchilov & Hutter, 2019) with a cosine schedule for the learning rate (Loshchilov & Hutter, 2017) with an initial value of 5e-4.

A compact multilayer perceptron usually serves as the core backbone for the diffusion models. It consists of five fully connected layers with width 256, Mish activations (Misra, 2019), He normal initialization (He et al., 2015), and residual connections (He et al., 2016). Targets and conditions are concatenated together and mapped to a shared embedding, time/SNR is embedded using a Fourier embedding (Tancik et al., 2020) of dimension 32 similar to Wildberger et al. (2023), while the residual blocks use a linear FiLM conditioning (Perez et al., 2018) for the embedded time and layer-wise normalization (Ba et al., 2016) after the skip-connection. A small dropout rate (Srivastava et al., 2014) of 0.05 is used for all models.

Code to reproduce the experiments is available at anonymous.4open.science/r/diffusion-experiments-63C2.

**EMA**  Additionally, we evaluate whether applying an exponential moving average (EMA) to model parameters improves performance. EMA maintains a smoothed version of the network weights $\hat{\boldsymbol{\psi}}$ by updating them as a weighted average of past values $\hat{\boldsymbol{\psi}}_{i-1}$ and the new weights from the regular update of the diffusion model weights $\boldsymbol{\psi}_i$ at training step $i$,

$$\hat{\boldsymbol{\psi}}_i = \beta_i \cdot \hat{\boldsymbol{\psi}}_{i-1} + (1 - \beta_i) \cdot \boldsymbol{\psi}_i, \tag{79}$$

where $\beta_i = (1 - 1/i)^{\gamma+1}$ and we set $\gamma = 6.94$ as discussed in Karras et al. (2024). The averaged weights are then used during inference, which can lead to better generalization during inference with diffusion models (Karras et al., 2024).

**Classifier two-sample test (C2ST)**  Following Lopez-Paz & Oquab (2017), we quantify the discrepancy between an approximate posterior $q(\boldsymbol{\theta} \mid \mathbf{y})$ and a reference ("ground-truth") posterior $p(\boldsymbol{\theta} \mid \mathbf{y})$ via a classifier two-sample test. For a given observation $\mathbf{y}$, we draw i.i.d. samples

$$\{\boldsymbol{\theta}_{\text{approx}}^{(i)}\}_{i=1}^n \sim q(\boldsymbol{\theta} \mid \mathbf{y}), \qquad \{\boldsymbol{\theta}_{\text{ref}}^{(i)}\}_{i=1}^n \sim p(\boldsymbol{\theta} \mid \mathbf{y}),$$

label the approximate samples with $z = 0$ and the reference samples with $z = 1$, and train a binary classifier $f_\phi(\cdot)$ to distinguish them using a split into training and test sets. The C2ST score for a given $\mathbf{y}$ is the classification accuracy on the test set,

$$\text{C2ST}(\mathbf{y}) = \frac{1}{n_{\text{test}}} \sum_{i=1}^{n_{\text{test}}} \mathbb{I}\big(f_\phi(\boldsymbol{\theta}^{(i)}) = z^{(i)}\big),$$

where $\mathbb{I}(\cdot)$ denotes the indicator function. A value close to 0.5 indicates that the two posteriors are hard to distinguish, whereas values approaching 1 signal a large discrepancy.

Without access to reference posterior samples, when can leverage C2ST also using samples from the prior alongside with simulations and posterior samples for the corresponding simulation (Yao & Domke, 2023), which we used in our third case study. In our second case study, the classifier takes as input both parameters and the corresponding log-likelihood values as sufficient summary statistic of the data.

**Maximum mean discrepancy (MMD)**  Maximum mean discrepancy provides a kernel-based, nonparametric measure of discrepancy between two distributions (Gretton et al., 2012). Given posterior samples from

an approximate distribution $\hat{\boldsymbol{\theta}} \sim q(\boldsymbol{\theta} \mid \mathbf{y}_r)$ and reference samples from a reference ("ground-truth") posterior $\boldsymbol{\theta}^\star \sim p(\boldsymbol{\theta} \mid \mathbf{y}_r)$, the squared MMD is defined as

$$\mathrm{MMD}_r^2(q,p) = \frac{1}{S^2} \sum_{s,s'=1}^{S} k(\hat{\boldsymbol{\theta}}_{r,s}, \hat{\boldsymbol{\theta}}_{r,s'}) + \frac{1}{S^2} \sum_{s,s'=1}^{S} k(\boldsymbol{\theta}_{r,s}^\star, \boldsymbol{\theta}_{r,s'}^\star) - \frac{2}{S^2} \sum_{s,s'=1}^{S} k(\hat{\boldsymbol{\theta}}_{r,s}, \boldsymbol{\theta}_{r,s'}^\star),$$

where $k$ is a positive-definite kernel. We use a multiscale inverse-multiquadratic kernel, constructed as a sum of individual kernels, $k_\sigma(x,y) = \sigma/(|x-y|^2 + \sigma)$, evaluated at log-spaced scales $\sigma$ from $10^{-6}$ to $10^6$. MMD equals zero if and only if the two distributions match in the kernel mean embedding, making it sensitive to discrepancies in mean, covariance, and higher-order moments. Unlike C2ST, MMD does not require training a classifier and exhibits low variance in low-dimensional settings, though it can suffer in very high dimensions unless kernels are carefully tuned. We report the median MMD across datasets to obtain a scalar score, with smaller values indicating better agreement between the approximate and reference posteriors.

**Normalized root mean squared error (NRMSE)** Let $\boldsymbol{\theta}_r \in \mathbb{R}^K$ denote the ground-truth parameter vector for dataset $r \in \{1, \ldots, R\}$, and let $\hat{\boldsymbol{\theta}}_{r,s} \in \mathbb{R}^K$ denote the $s$-th posterior draw for dataset $r$ from an approximate posterior, with $s \in \{1, \ldots, S\}$. For parameter index $j \in \{1, \ldots, K\}$, we first define the posterior root mean squared error (RMSE) across posterior draws for a fixed dataset $r$:

$$\mathrm{RMSE}_{j,r}^{\mathrm{post}} = \sqrt{\frac{1}{S} \sum_{s=1}^{S} (\hat{\theta}_{r,s,j} - \theta_{r,j})^2}.$$

To normalize the posterior RMSE into an interpretable range, we divide by a prior predictive error scale estimated from prior samples. Let $\tilde{\theta}_{r,s,j}$ denote $S$ bootstrap samples drawn from the empirical prior distribution. We then define

$$\mathrm{RMSE}_{j,r}^{\mathrm{prior}} = \sqrt{\frac{1}{S} \sum_{s=1}^{S} (\tilde{\theta}_{r,s,j} - \theta_{r,j})^2}.$$

We normalize each posterior RMSE by the median prior RMSE across datasets:

$$\mathrm{NRMSE}_{j,r} = \frac{\mathrm{RMSE}_{j,r}^{\mathrm{post}}}{\mathrm{median}_{r'=1,\ldots,R}(\mathrm{RMSE}_{j,r'}^{\mathrm{prior}})}.$$

We then aggregate the NRMSE over datasets to obtain the NRMSE per parameter:

$$\mathrm{NRMSE}_j = \mathrm{median}_{r=1,\ldots,R}(\mathrm{NRMSE}_{j,r}),$$

and report the mean over parameters. The values of the normalization should be interpreted as 0 indicating the most informative (posterior is a point mass at ground truth) and 1 indicating non-informative (posterior equals prior) results.

**Expected calibration error (ECE)** Calibration measures the agreement between nominal credibility levels and empirical coverage. For each parameter $j$ and dataset $r$, we consider posterior samples $\{\hat{\boldsymbol{\theta}}_{r,s,j}\}_{s=1}^{S}$. For a grid of credibility levels $\alpha \in [\alpha_{\min}, \alpha_{\max}]$, we form central $(1-\alpha)$ credible intervals:

$$\ell_{r,j}(\alpha) = q_{r,j}\left(\frac{\alpha}{2}\right), \qquad u_{r,j}(\alpha) = q_{r,j}\left(1 - \frac{\alpha}{2}\right),$$

where $q_{r,j}(\cdot)$ is the empirical quantile. Coverage at level $\alpha$ is

$$\hat{c}_j(\alpha) = \frac{1}{R} \sum_{r=1}^{R} \mathbb{1}(\boldsymbol{\theta}_{r,j} \in [\ell_{r,j}(\alpha), u_{r,j}(\alpha)]),$$

and the absolute calibration error is

$$e_j(\alpha) = |\hat{c}_j(\alpha) - (1 - \alpha)|.$$

Aggregating across a grid $\{\alpha_m\}_{m=1}^M$ gives

$$\mathrm{ECE}_j = \mathrm{median}_{m=1,\ldots,M}\, e_j(\alpha_m).$$

As with the NRMSE, we report the mean over all parameters.

**Posterior contraction**   Posterior contraction quantifies the reduction in uncertainty from prior to posterior. The prior variance of parameter $j$ is estimated as

$$\mathrm{Var}_{\mathrm{prior}}(\boldsymbol{\theta}_j) = \mathrm{Var}_{r=1,\ldots,R}(\boldsymbol{\theta}_{r,j}),$$

and the posterior variance for dataset $r$ as

$$\mathrm{Var}_{\mathrm{post}}(\boldsymbol{\theta}_j \mid \mathbf{y}_r) = \mathrm{Var}_{s=1,\ldots,S}(\hat{\boldsymbol{\theta}}_{r,s,j}).$$

Define the per-dataset contraction:

$$\mathrm{PC}_{r,j} = \min\left(\max\left(1 - \frac{\mathrm{Var}_{\mathrm{post}}(\boldsymbol{\theta}_j \mid \mathbf{y}_r)}{\mathrm{Var}_{\mathrm{prior}}(\boldsymbol{\theta}_j)}, 0\right), 1\right).$$

Aggregating across datasets yields

$$\mathrm{PC}_j = \mathrm{median}_{r=1,\ldots,R}\mathrm{PC}_{r,j}.$$

We report the mean over parameters. Values close to 1 indicate strong contraction (substantial uncertainty reduction), whereas values near 0 indicate little contraction. However, higher contraction does not necessarily mean a better approximation of the "true" posterior. Contraction is thus only informative in the context of calibration.

**Test Accuracy with Random Points (TARP)**   To assess global posterior calibration, we also employ the TARP diagnostic introduced by Lemos et al. (2023). Let $\boldsymbol{\theta}_r \in \mathbb{R}^K$ denote the ground-truth parameter vector for dataset $r \in \{1,\ldots,R\}$, and let $\hat{\boldsymbol{\theta}}_{r,s} \in \mathbb{R}^K$ denote the $s$-th posterior draw for dataset $r$ from an approximate posterior, with $s \in \{1,\ldots,S\}$. For each dataset $r$, a random reference point $\tilde{\boldsymbol{\theta}}_r$ is sampled from the prior. Before computing distances, parameters are standardized using the empirical mean and standard deviation of the ground-truth parameters. Using a distance metric $d(\cdot,\cdot)$ (here the Euclidean distance), we compute

$$d_r^{(s)} = d(\tilde{\boldsymbol{\theta}}_r, \boldsymbol{\theta}_r^{(s)}), \qquad d_r^\star = d(\tilde{\boldsymbol{\theta}}_r, \boldsymbol{\theta}_r).$$

The TARP coverage statistic for dataset $r$ is then defined as the fraction of posterior samples that are closer to the reference point than the true parameter,

$$c_r = \frac{1}{S}\sum_{s=1}^S \mathbb{1}\left[d_r^{(s)} < d_r^\star\right].$$

If the posterior approximation is calibrated, the random variables $c_i$ are uniformly distributed on $[0,1]$. The empirical cumulative distribution function (ECDF) of the $c_i$ values therefore should follow the diagonal line

$$\mathrm{ECDF}(\alpha) = \alpha.$$

We summarize deviations from calibration using the area-to-curve (ATC),

$$\mathrm{ATC} = \int_{0.5}^1 (\mathrm{ECDF}(\alpha) - \alpha)\, d\alpha.$$

Values near zero indicate calibrated posteriors, positive values indicate overdispersed posteriors, and negative values indicate underdispersed posteriors.

### A.1.1 Case Study 1: Low-Dimensional Benchmarks

**Setup** We ran all benchmark problems using online training with 1,000 epochs, a batch size of 128 and 32,000 simulations. We then sampled as many posterior samples as in the reference posterior for each observation in the benchmark collection and compared the model with the corresponding sampler using C2ST configured as provided by Lueckmann et al. (2021). For each model, we tested different ODE and SDE (if possible) and timed the inference time. For each sampler, we used 500 steps or used an adaptive method with minimal 50 steps and maximal 1,000 steps (which was never reached). As classifier for the C2ST, we use the setup provided by Lueckmann et al. (2021).

**Problems** For a detailed description of the prior and simulators, we refer to the original benchmark (Lueckmann et al., 2021). To improve numerical stability and enforce parameter constraints, we apply simple parameter transformations depending on the problem. For the Lotka-Volterra and SIR models, a log transformation is applied to the parameters to enforce positivity. For the Gaussian mixture ($[-10, 10]$), SLCP with and without distractors ($[-3, 3]$), Gaussian linear with uniform prior ($[-1, 1]$), and two-moons problems ($[-1, 1]$), parameters are constrained to the corresponding fixed bounded interval via a sigmoid function. All parameters are transformed for training and back-transformed after posterior sampling. Transformed parameters and observables are standardized during training.

**ODE solvers** The ODE samplers included Euler, RK45 (with the coefficients used in the Dormand–Prince variant (Dormand, 1996), which is the standard ODE solver in many libraries), and TSIT5 (which has newer coefficients for Runge–Kutta that seem to be more efficient (Tsitouras, 2011)), in both fixed-step and adaptive variants. For the adaptive variants, we used the embedded 4th-order solution to form a scalar error estimate by taking the maximum $\ell_2$-norm over all state components, and updated the step size via a standard controller $h_{\text{new}} = h \cdot 0.9\big(\text{err}/(\text{atol} + \text{rtol} \max_i |\mathbf{z}_i|) + 10^{-12}\big)^{-1/5}$, clipped to stay within a factor of $[0.2, 5]$ and further constrained by minimum and maximum step-size bounds (default $\text{atol} = 10^{-6}$, $\text{rtol} = 10^{-4}$).

**Stochastic solvers** The SDE samplers covered Euler–Maruyama (EulerM) (with fixed, scheduled, adaptive, and predictor–corrector variants), SEA (an improved one-step method with strong order 1 for SDEs with additive noise) (Foster et al., 2024), ShARK (Foster et al., 2024) (two-step method with strong order 1.5 for SDEs with additive noise), an adaptive two-step method (Jolicoeur-Martineau et al., 2021), and annealed Langevin dynamics (Song & Ermon, 2019; 2020). The Euler–Maruyama adaptive variant followed Fang & Giles (2020), setting the step size proportional to $\max(1, \|\mathbf{x}\|^2)/\max(1, \|\mathbf{v}(\mathbf{x})\|^2)$ with clipping based on minimal/maximal steps. The scheduled variant, uses a fixed step size in the log SNR. The two-step adaptive scheme of Jolicoeur-Martineau et al. (2021) was implemented as an Euler–Heun predictor–corrector: we estimated the local error from the discrepancy between the Euler and Heun updates, compared the normalized error against combined relative/absolute tolerances ($\epsilon_{\text{abs}} = 0.02576$, 1% of 99% CI of standard normal as targets are standardized, and $\epsilon_{\text{rel}} = 0.1$), and used an accept–reject step with settings from Jolicoeur-Martineau et al. (2021) for step-size adaptation. Step size was also clipped based on minimal/maximal allowed steps. The Langevin-based sampler used four times the nominal step count, and predictor–corrector (PC) versions applied one (Euler–Maruyama) or five (Langevin) corrector updates per step. The implementation of the Langevin sampler was based on recommendations in Song & Ermon (2020) and for PC based on Song et al. (2021c).

**Architectures** A multilayer perceptron usually serves as the core backbone for the diffusion models as described above (Section A.1. As an alternative backbone, we employ a transformer (Vaswani et al., 2017) with a DiT-style design (Peebles & Xie, 2023). Each target and each condition dimension is tokenized individually and projected to a shared embedding of width 128, augmented with learnable per-dimension identifier embeddings and a learnable state embedding that marks each token as latent, observed, or missing similar to (Gloeckler et al., 2024). Target tokens form the residual stream processed by five transformer blocks, while condition tokens are embedded once and enter every block only in a cross-attention fashion. Each block combines multi-head self-attention with 4 heads (Vaswani et al., 2017) and a SwiGLU feed-forward network (Shazeer, 2020) of expansion factor 3, using RMS normalization of the queries and keys (Henry et al., 2020). We used no bias on the projections, Glorot-uniform initialization (Glorot & Bengio, 2010), and

the same time/SNR of dimension 32 as in the MLP, but injected through adaptive layer normalization with zero-initialized modulation (AdaLN-Zero) (Perez et al., 2018; Peebles & Xie, 2023).

We ran all analyses on a computing cluster. The computing cluster used an Intel Xeon Sapphire Rapids CPU with a core clock speed of up to 2.1 GHz and 60 GB of RAM. The neural network training was performed on a cluster node with an Nvidia A40 graphics card with 48 GB of VRAM. Timing is reported using `TensorFlow` as backend.

**OT-Conditional Flow Matching**  Optimal transport flow matching replaces the independent pairing of samples from the base and target distributions with a coupling induced by an (entropically regularized) optimal transport plan (Tong et al., 2024; Pooladian et al., 2023). Given a mini-batch $\{\mathbf{z}_0^{(i)}, \mathbf{z}_1^{(i)}\}_{i=1}^n$, a cost matrix $\mathbf{C}_{ij} = \|\mathbf{z}_0^{(i)} - \mathbf{z}_1^{(j)}\|_2^2$ is constructed and converted into a kernel

$$\mathbf{K}_{ij} = \exp(-\mathbf{C}_{ij}/\epsilon),$$

where $\epsilon > 0$ is an entropic regularization parameter. The Sinkhorn–Knopp algorithm then alternates row and column normalizations of $\mathbf{K}$ until the prescribed marginals are matched, yielding a transport plan $\pi \in \mathbb{R}^{n \times m}$ (Sinkhorn & Knopp, 1967; Cuturi, 2013). We set $\epsilon = 0.1$, a maximal number of 100 optimization steps and use a log-stabilized version of the Sinkhorn–Knopp algorithm.

While balanced uniform marginals are used for standard OT, *partial* optimal transport augments the problem with a dummy mass to transport only a fraction $s \in (0, 1)$ of the probability mass, which can improve robustness to misspecified pairings in mini-batch settings (Nguyen et al., 2022). For conditional OT, the transport cost can be modified by incorporating a condition-dependent penalty, effectively restricting transport to pairs with similar conditions and reducing spurious matches (Fluri & Hofmann, 2024; Cheng & Schwing, 2025). We compute a pairwise cosine distance for all conditions and add this to the cost matrix $\mathbf{C}$ with a weight $w$ determined by the ratio $r(w)$, which measures the proportion of samples that are considered potential optimal transport candidates as described by Cheng & Schwing (2025).

We benchmarked standard optimal transport (OT), partial optimal transport (POT) with $s = 0.8$, conditional optimal transport (COT) with a $r(w) = 0.01$ and both strategies together (CPOT) with $r(w) = 0.01$ and $s = 0.9$. We generally observe that not enough contraction is achieved with standard or partial OT. COT, and CPOT with $r(w) = 0.01$ improved upon standard OT, reaching a similar level of accuracy as without any form of OT (see Figure A.2). Higher values of $r$ lead to straighter path, which can be beneficial if fewer steps in the sampler must be used, but straighter paths do not improve C2ST in general (see Figure A.2), e.g., for the SIR problem, straighter paths help while sampling with only 10 steps but not for more steps.

### A.1.2 Case Study 2: Large-Scale ODE Benchmark

**Data generation**  Synthetic datasets are simulated under the original experimental conditions with parameters sampled from normal priors on the parameter scale, centered at the published parameter estimates. Noise is added according to the Gaussian measurement model specified. This allows to compute the likelihood and its gradients and controlled comparison between Markov chain Monte Carlo (MCMC) and a range of diffusion based approaches. We use `AMICI` (Fröhlich et al., 2021) for simulation with forward sensitives to compute gradients and the No U-Turn Sampler (NUTS) (Homan & Gelman, 2014) implemented in `pyPESTO` (Schälte et al., 2023). Due to the high dimensions, exploring the parameter space is difficult even for a gradient based algorithm. We fixed the simulation budget to 32,768 simulations.

**Architecture**  To obtain a scalar comparison across methods, we rank them by the sum of ECE and NRMSE. For this setting, we combine the diffusion models with a summary network, which embeds the data before passing it through the same backbone as before. All models were tested with two distinct summary networks implemented in `BayesFlow` (Kühmichel et al., 2026):

- A `FusionTransformer` that applies stacked self-attention layers to encode the data, followed by cross-attention with a learnable template vector that is summarized via a LSTM network with 128 units.

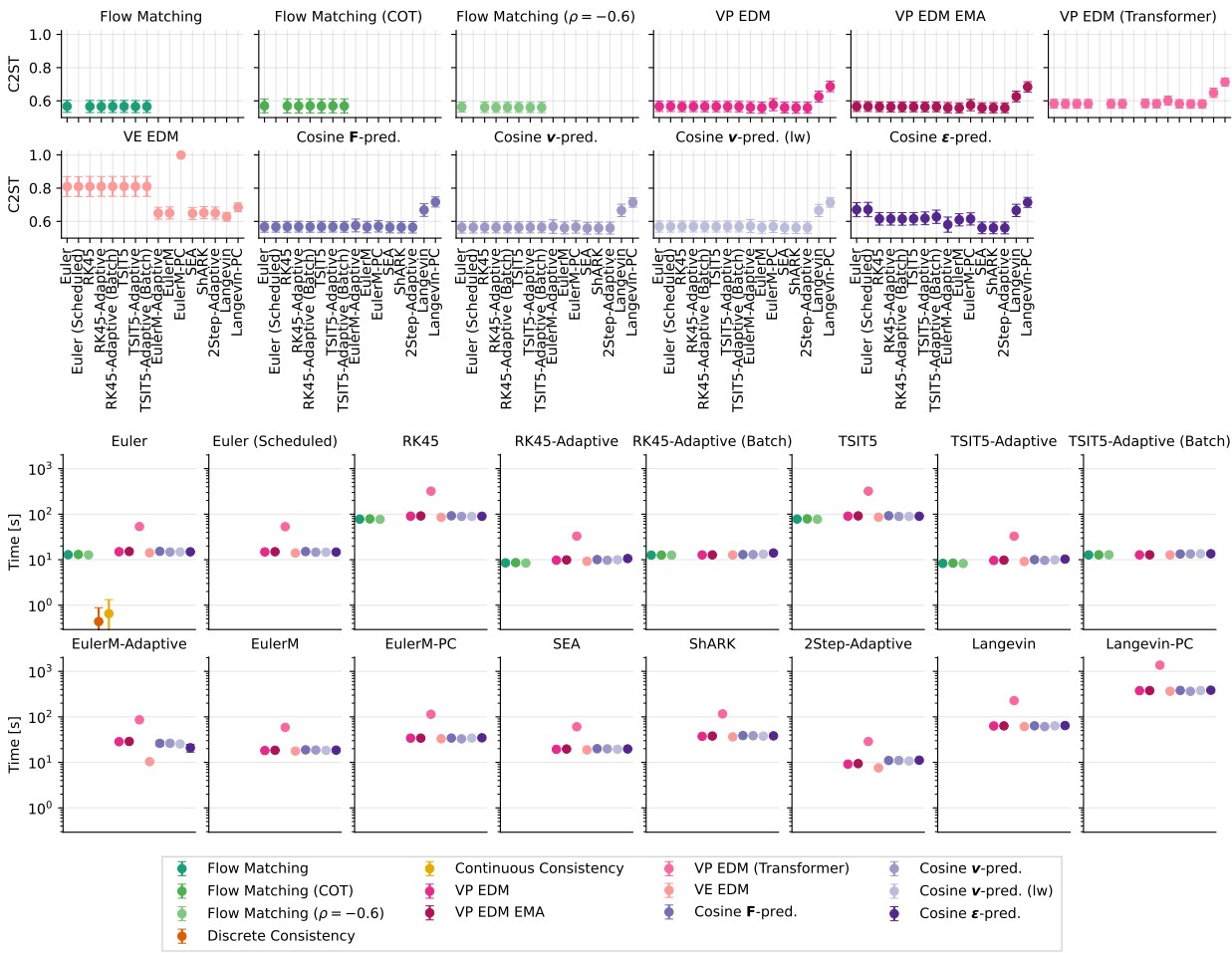

Figure A.1: *Extended Case Study.* Mean (and std) of C2ST and timing for all samplers across all benchmark problems.

- And a `TimeSeriesNetwork` that follows the LSTNet architecture (Zhang & Mikelsons, 2023a). It combines convolutional layers to extract local temporal features with a LSTM with 256 units that captures long-range dependencies.

Both network were used in their standard settings, with the summary dimension set to twice the number of parameters. As backbone for all models we used 5 layers of residual layers with each 256 units. To enforce parameter bounds given by the problem, we apply a sigmoid transformation to the parameters for training and back-transformed after posterior sampling. Transformed parameters and log-transformed observables are standardized during training.

For sampling, we use adaptive ODE and SDE solvers. We used TSIT5 as ODE solver and the two-step adaptive solver by (Jolicoeur-Martineau et al., 2021) with settings as described before in Section A.1.1. As a classifier, we use a sequence of two MLPs with width of 128 and 5-fold cross-validation.

### A.1.3 Case Study 3: Gaussian Random Fields as a Scalable SBI Benchmark

The GRF benchmark consists of two different tasks: (i) a low-dimensional target regime where the condition is a high-dimensional field; (ii) target and condition are switched and the task becomes to generate a high-dimensional field.

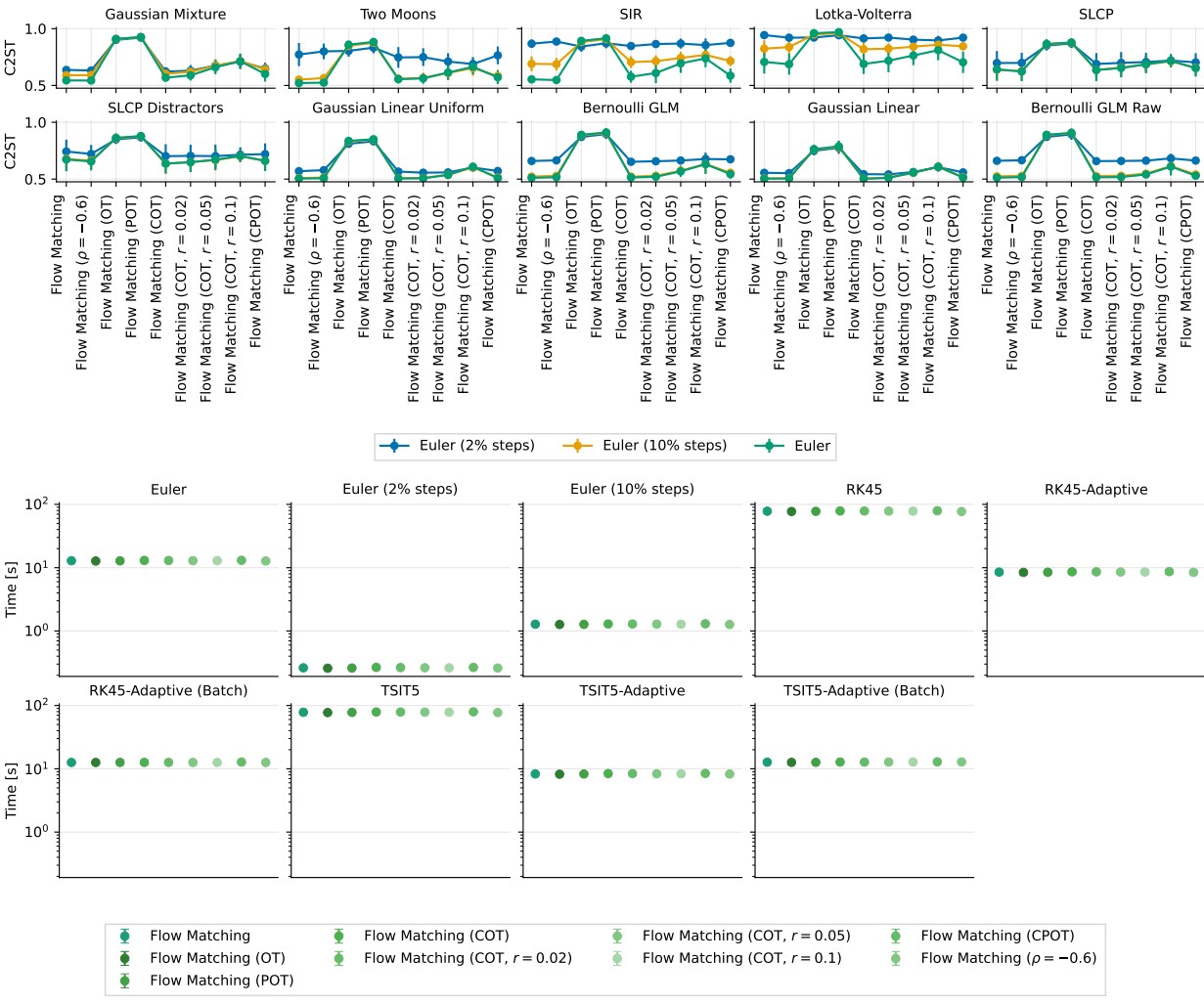

Figure A.2: *Optimal Transport Flow Matching.* Mean (and std) of C2ST and timing for all variants of optimal transport flow matching across all benchmark problems.

**GRF simulator.** We implement the GRF prior using the `FyeldGenerator` (Cadiou, 2022). This takes as input (i) a sampling rule for a random field in Fourier space, (ii) a target power spectrum, (iii) the desired output shape, and (iv) a physical pixel size, and returns a real-valued random field on a regular grid.

For a given spatial resolution $r \in \{8, 16, 32, 64, 128, 256^\star\}$ (where $r = 256$ is only used for the case $\mathbf{z} = (\alpha, \log \sigma)$), we consider square grids of shape $(r, r)$ and draw hyperparameters once per simulation from the Gaussian priors

$$\log \sigma \sim \mathcal{N}(0, 0.3^2) \quad \text{and} \quad \alpha \sim \mathcal{N}(3, 0.5^2), \tag{80}$$

as described in the main text. These are collected in $\mathbf{z} = (\alpha, \log \sigma)$.

Conditioned on $(\alpha, \log \sigma)$, we define a power-law spectrum

$$P(k) \propto k^{-\alpha} \sigma^2, \qquad \sigma = \exp(\log \sigma), \tag{81}$$

where $k$ denotes the norm of the spatial frequency vector in Fourier space. The routine `generate_field` then proceeds in three conceptual steps.

First, a grid of Fourier frequencies is constructed by calling a discrete-frequency routine (analogous to `fftfreq`) for each spatial dimension, accounting for a user-specified pixel size (the *unit length*). These

Figure A.3: *Backbone comparison for the high-dimensional GRF field-generation task.* The target is the field $\mathbf{z} = (\boldsymbol{\varphi})$, conditioned on $\mathbf{x} = (\alpha, \log\sigma)$. We compare UNet, UViT, and Residual UViT backbones for diffusion models (DM), flow matching (FM), and consistency models (CM) under low- and high-budget training. Lines show C2ST scores across field resolutions, and boxplots summarize performance averaged over resolutions. Values close to $(0.5)$ indicate that generated fields are difficult to distinguish from simulator samples. Across backbones, DM and FM remain close to the simulator baseline, whereas CM deteriorates at larger resolutions. UViT and Residual UViT yield small improvements over the UNet baseline, but the qualitative ranking of methods is stable across backbone choices.

frequencies are combined into a multidimensional grid, and their Euclidean norm defines $k$ at each Fourier mode. Second, a random complex-valued field in Fourier space is drawn from a user-provided "statistic" function. In our case, this statistic corresponds to independent complex Gaussian noise at each mode, so that the resulting field is zero-mean and has random phases. This field is then rescaled mode-wise by the square root of the target power spectrum, i.e. by $\sqrt{P(k)}$ evaluated on the frequency grid. This enforces the desired second-order structure and yields a sample whose covariance is consistent with the specified spectrum. Finally, an inverse multidimensional FFT is applied to this scaled Fourier field and the real part of the result is taken as a spatial field $\varphi$ of shape $(r, r)$.

The *unit length* argument controls the physical spacing per pixel, which we set proportional to $1/|\alpha|$ in our experiments. This choice tames otherwise extreme variations in the overall variance across the support of the prior and yields fields whose amplitude remains within a numerically convenient range (approximately from $10^{-3}$ to $10^{1}$ instead of $10^{-4}$ to $10^{2}$ with constant unit length of one), while preserving the qualitative effect of $\alpha$ on the characteristic structure size.

We wrap this prior–likelihood pair into a simulator: each simulator call first samples $(\alpha, \log\sigma)$ from the Gaussian priors, then uses `generate_field` with the corresponding simulator spectrum and unit length to produce a single-channel field. The same simulator (including priors, spectrum, and field-generation mechanism) is used for both the low-dimensional parameter experiments, where $(\alpha, \log\sigma)$ are the inference target, and the high-dimensional parameter experiments, where the field $\varphi$ is treated as the parameter to be generated.

**Architectures.** At a given resolution, for a fixed parameter setting (low- vs. high-dimensional) and simulation budget, all three model families share exactly the same backbone architecture.

In the low-dimensional parameter case, the field $\boldsymbol{\varphi} \in \mathbb{R}^{r \times r}$ is first encoded by a small residual convolutional network into an 8-dimensional summary, which is then processed by a fully connected backbone. The summary network is a resolution-dependent ResCNN built from stages, where each stage consists of a double-convolution residual block followed by a max-pooling operation. Concretely, each residual block uses $3 \times 3$ convolutions with 16 channels, with a non-linearity after each convolution and a skip connection across the pair of convolutions. The number of stages grows with resolution: for $r \in \{8, 16\}$ we use a single stage, for $r \in \{32, 64\}$ two stages, and for $r \in \{128, 256\}$ four stages. After the final stage, the spatial feature map is projected to an 8-dimensional vector; this summary is the input to a three-layer MLP with 64 hidden units per layer, no batch normalization, and no dropout.

The classifier used for the C2ST evaluation in the high-dimensional experiments reuses exactly the same resolution-dependent ResCNN as this summary network. The only architectural difference is the final output layer: instead of an 8-dimensional summary, the classifier produces a single scalar logit. We feed the classifier

triplets $(\boldsymbol{\varphi}, \alpha, \log \sigma)$ and map the aggregated features to a scalar for binary discrimination between simulated and generated triplets.

In the high-dimensional parameter case, the field $\boldsymbol{\varphi} \in \mathbb{R}^{r \times r}$ itself is treated as the parameter to be generated, while the scalars $(\alpha, \log \sigma)$ serve as conditioning inputs and are tiled to the corresponding field resolution. We use the `BayesFlow` U-Net backbone as the main architecture (Kühmichel et al., 2026), with a fixed channel width of 32 and a resolution-dependent number of stages: two stages for $r = 8$, three for $r \in \{16, 32\}$, four for $r = 64$, and five for $r = 128$. To assess sensitivity to the field backbone, we additionally report results for UViT and Residual UViT variants (Hoogeboom et al., 2023; 2025) in Figure A.3. For these variants, we keep the same stage widths and use two transformer blocks with transformer width 32 and dropout 0.1. Across backbones, the method rankings are consistent, with only small improvements for the UViT-style variants over the U-Net baseline. We therefore retain the U-Net in the main experiments as a simple and conservative baseline, and use the UViT and Residual UViT results as an appendix-level robustness check.

**Training and evaluation.** In the low-dimensional case, DM, FM, and CM are trained with a batch size of 32 and the AdamW optimizer with a learning rate of $1e{-}4$. For each resolution, we simulate 5000 training fields and 500 validation fields, and for each validation field, we drew 1000 posterior samples to compute NRMSE and calibration metrics. We also evaluated C2ST in this low-dimensional case using the same classifier as for the high-dimensional case, again with the triplet as input, and found that all models reached approximately 50% across all settings.

In the high-dimensional case, DM, FM, and CM are again trained with a batch size 32. For C2ST, we re-simulate 5000 training and 500 validation instances; for each condition, we generate one field. The classifier uses a batch size of 16, AdamW with a learning rate of $10^{-3}$ and weight decay of $10^{-2}$, and is trained with binary cross-entropy and early stopping on validation accuracy; the best validation accuracy is reported as the C2ST score.

Sampling for DM and FM is performed using the TSIT5 ODE solver with 500 steps, whereas the consistency model uses only 10 steps. On a single NVIDIA GeForce RTX 4090 GPU, this difference in step count translates into a substantial wall-clock gap in the high-dimensional field-generation setting: generating the C2ST training set takes on the order of one hour for DM and FM, but about one minute for CM.

### A.1.4 Case Study 4: Pooling Regimes in Cognitive Modeling

For this case study, we have two different priors: a flat and a hierarchical one. For each we train a diffusion model with EDM noise schedule with the standard backbone as in the first case studies for 500 epochs. For training, we simulate 256 batches of size 128, with $R = 1$ subject per training instance and $N = 30$ trials per subject. Hence, as a summary network, we employ a `SetTransformer` (Lee et al., 2019) with a summary dimension of 16 and dropout of 0.1 to account for the exchangeability of the trials. For inference, we simulate 100 test sets of 100 subjects, each with 30 trials. To solve the reverse SDE, we employ the two-step adaptive method by (Jolicoeur-Martineau et al., 2021) with a maximum of 1,000 steps. We use $d(t) = d_0(\frac{d_1}{d_0})^t$ with $d_0 = 0.1$ (partial pooling) and $d_0 = 0.01$ (complete pooling) and $d_1 = 1$ to improve calibration of the compositional score.

**Simulator** We simulate binary choices and reaction times from an evidence accumulation process with absorbing boundaries at 0 and $\alpha$. A single trial is defined by the drift rate $\nu$, the boundary separation $\alpha$, the non-decision time $t_0$, and the relative starting point $\beta \in [0, 1]$.

Given $(\nu, \alpha, t_0, \beta)$, we simulate a single trial by discretizing the stochastic differential equation

$$\mathrm{d}Y_t = \nu \, \Delta t + \mathrm{d}W_t, \quad Y_0 = \beta \alpha,$$

with absorbing boundaries at 0 and $\alpha$. We use an Euler–Maruyama scheme with time step $\Delta t = 10^{-3}$, and a maximal decision time of $t_{\max} = 10\,\mathrm{s}$. The process is initialized as $Y_0 = \beta \alpha$ at $t_0$ and iterated while $0 \le Y_t \le \alpha$ and $t \le 10$. The observed binary choice $c$ is 1 when the upper boundary is hit or 0 else.

**No-pooling and complete pooling**  As prior for the parameters we set for each subject $r \in \{1, \ldots, R\}$ and simulated trial $n \in \{1, \ldots, N\}$

$$\nu^{(r)} \sim \mathcal{N}(0.5, \exp(-1)), \quad \log \alpha^{(r)} \sim \mathcal{N}(0, \exp(-3)), \quad \log t_0^{(r)} \sim \mathcal{N}(-1, \exp(-1)),$$

$$\beta^{(r)} \sim \mathrm{Beta}(a = 50, b = 50), \quad \mathbf{y}_n^{(r)} \sim \mathrm{EAM}(\nu^{(r)}, \alpha^{(r)}, t_0^{(r)}, \beta^{(r)}).$$

For inference in the no-pooling case, we select one subject per dataset, and we assume independence across subjects. In the case of complete pooling, we assume that all subjects in one dataset are described by the same parameters (Bardenet et al., 2017) and we use the compositional score over all subjects following (Arruda et al., 2026). We use a mini-batch size of 3 for the compositional score.

To facilitate training, we represent the starting bias via a Gaussian variable $z_\beta$ and transform it into a Beta-distributed random variable. Concretely,

$$\beta = \mathrm{Beta}^{-1}(a, b)(u), \quad u = \Phi(z_\beta), \quad z_\beta \sim \mathcal{N}(0, 1),$$

where $\Phi$ is the standard normal CDF and $\mathrm{Beta}^{-1}(a, b)$ the inverse CDF of the $\mathrm{Beta}(a, b)$ distribution. This yields $\beta \sim \mathrm{Beta}(50, 50)$ while keeping a simple Gaussian parameterization for inference.

**Partial pooling**  Each subject specific parameter $\boldsymbol{\theta}^{(r)}$ is drawn from a group level distribution governed by a global mean $\boldsymbol{\mu}$ and a global standard deviation $\boldsymbol{\sigma}$. This allows subjects to differ, while introducing a hierarchical prior that ties them together (Habermann et al., 2025):

$$
\begin{aligned}
&\mu_\nu \sim \mathcal{N}(0.5, 0.3), & &\log \sigma_\nu \sim \mathcal{N}(-1, 1), & &\nu^{(r)} \sim \mathcal{N}(\mu_\nu, \sigma_\nu) \\
&\mu_\alpha \sim \mathcal{N}(0, 0.05), & &\log \sigma_\alpha \sim \mathcal{N}(-3, 1), & &\log \alpha^{(r)} \sim \mathcal{N}(\mu_\alpha, \sigma_\alpha) \\
&\mu_{t_0} \sim \mathcal{N}(-1, 0.3), & &\log \sigma_{t_0} \sim \mathcal{N}(-1, 0.3), & &\log t_0^{(r)} \sim \mathcal{N}(\mu_{t_0}, \sigma_{t_0}) \\
&\beta \sim \mathrm{Beta}(a = 50, b = 50), & &\mathbf{y}_n^{(r)} \sim \mathrm{EAM}(\nu^{(r)}, \alpha^{(r)}, t_0^{(r)}, \beta).
\end{aligned}
$$

In this setting, we can train a global model to predict the global parameters $(\boldsymbol{\mu}, \boldsymbol{\sigma}, \beta)$ on individual subjects with $N$ trials. Similar to the complete pooling case, and with the same setting, we get samples of the global means and standard deviations with compositional diffusion, following Arruda et al. (2026). A local model, here a continuous consistency model, trained on single individuals with $N$ trials, but with an additional condition, which are the global parameters, predicts us the per subject parameters $(\nu^{(r)}, \alpha^{(r)}, t_0^{(r)})$ conditional on the global parameters. The local model conditions on observations of a single subject and, during training, the true global parameters. For inference, we perform ancestral sampling: draw a global parameter from the group level posterior and then condition the local prediction on this global sample.

