# OpenReview forum: "Diffusion Models in Simulation-Based Inference: A Tutorial Review"
_TMLR — Under review for TMLR_

### Review · Reviewer_wsPp · 2026-05-09

**Summary Of Contributions:**

This submission presents a tutorial review of diffusion models applied to simulation-based inference, covering: foundational exposition of SBI and score-based diffusion models, a scoping review of recent applications, and an empirical comparison of design choices including noise schedules, model parameterizations, weighting functions, and ODE/SDE solvers. The empirical evaluation spans four case studies: a low-dimensional benchmark suite adapted from Lueckmann et al. [1], a 72-parameter ODE model, a Gaussian random field benchmark introduced by the authors, and a cognitive evidence accumulation model used to illustrate compositional score matching under pooling hierarchies. The paper additionally covers classifier-free guidance, compositional score aggregation, optimal transport flow matching, and consistency models in the SBI context. Claimed contributions beyond synthesis include the first application of continuous consistency models to SBI and the GRF benchmark.

**Additional Comments:**

References:
1. Lueckmann JM, Boelts J, Greenberg D, Goncalves P, Macke J. Benchmarking simulation-based inference. InInternational conference on artificial intelligence and statistics 2021 Mar 18 (pp. 343-351). PMLR.
2. Hermans J, Delaunoy A, Rozet F, Wehenkel A, Begy V, Louppe G. A crisis in simulation-based inference? Beware, your posterior approximations can be unfaithful. Transactions on Machine Learning Research. 2022 Sep.
3. Lemos P, Coogan A, Hezaveh Y, Perreault-Levasseur L. Sampling-based accuracy testing of posterior estimators for general inference. InInternational Conference on Machine Learning 2023 Jul 3 (pp. 19256-19273). PMLR.
4. Arruda J, Pandey V, Sherry C, Barroso M, Intes X, Hasenauer J, Radev ST. Compositional amortized inference for large-scale hierarchical Bayesian models. arXiv preprint arXiv:2505.14429. 2025 May 20.
5. Radev ST, Schmitt M, Schumacher L, Elsemüller L, Pratz V, Schälte Y, Köthe U, Bürkner PC. BayesFlow: Amortized Bayesian workflows with neural networks. arXiv preprint arXiv:2306.16015. 2023 Jun 28.
6. Song Y, Sohl-Dickstein J, Kingma DP, Kumar A, Ermon S, Poole B. Score-based generative modeling through stochastic differential equations. arXiv preprint arXiv:2011.13456. 2020 Nov 26.
7. Karras T, Aittala M, Aila T, Laine S. Elucidating the design space of diffusion-based generative models. Advances in neural information processing systems. 2022 Dec 6;35:26565-77.
8. Holderrieth P, Erives E. An introduction to flow matching and diffusion models. arXiv preprint arXiv:2506.02070. 2025 Jun 2.
9. Geffner T, Papamakarios G, Mnih A. Compositional score modeling for simulation-based inference. InInternational Conference on Machine Learning 2023 Jul 3 (pp. 11098-11116). PMLR.
10. Linhart J, Cardoso GV, Gramfort A, Corff SL, Rodrigues PL. Diffusion posterior sampling for simulation-based inference in tall data settings. arXiv preprint arXiv:2404.07593. 2024 Apr 11.
11. Vuong AB, McCann MT, Santos JE, Lin YT. Are We Really Learning the Score Function? Reinterpreting Diffusion Models Through Wasserstein Gradient Flow Matching. arXiv preprint arXiv:2509.00336. 2025 Aug 30.
12. Touron C, Cardoso GV, Arbel J, Rodrigues PL. Error analysis of a compositional score-based algorithm for simulation-based inference. arXiv preprint arXiv:2510.15817. 2025 Oct 17.

**Audience:**

Yes

**Audience Explanation:**

The theoretical exposition in Sections 2 through 4 is competent but synthesizes material already unified in prior tutorial treatments, including the BayesFlow framework [5], the original score-based SDE paper [6], the EDM formulation [7], and the flow matching tutorial [8]. The additive structure of scores under Bayesian factorizations is a known property exploited in Geffner et al. [9] and Linhart et al. [10]; the present paper reviews this material without extending it analytically. The empirical findings confirm existing intuitions. VP schedules are robust, VE can be unstable in high dimensions, F- and v-parameterizations are more stable than direct noise prediction, and consistency models trade accuracy for speed. These are findings that practitioners in the field would consider expected rather than surprising, and the paper does not provide a formal account of why these patterns hold.

**Broader Impact Concerns:**

No significant broader impact concerns

**Claims And Evidence:**

No

**Claims Explanation:**

The presentation of continuous consistency models as a novel SBI contribution is supported only by a passing result in Table 4, where the continuous consistency model underperforms relative to the discrete variant and to flow matching on the large-scale ODE case study. The empirical evidence therefore argues against this as a positive contribution. The GRF benchmark is described as a new contribution, but the benchmark itself is a straightforward two-parameter power-law spectral model with no simulator mismatch, no sequential structure, and no likelihood intractability; it is significantly simpler than the ODE models already in common use [1] and contributes less as a standalone benchmark than as an illustration of dimension scaling. The statement that "for most problems, the VP schedule is a suitable choice" is repeated across sections but depends heavily on the problem class, and the paper's own results show flow matching outperforming VP EDM on the high-dimensional case (Table 4, rank 1), which undermines the universality of this recommendation.

**Requested Changes:**

The paper assembles known results without offering a unifying perspective that would allow a reader to predict, rather than merely observe post hoc, which design choices will succeed on a given problem.

The GRF benchmark requires substantive extension to serve as a lasting reference. A two-parameter spectral model is not representative of the high-dimensional intractable-likelihood settings that motivate SBI; the authors should consider whether an alternative benchmark with genuine likelihood intractability and at least moderate parameter dimensionality would better illustrate the design-choice trade-offs they intend to characterize. As it stands, the GRF results in Figure 12 largely confirm that diffusion models and flow matching scale to image generation, which is already well established outside SBI.

The submission discusses SBC and C2ST as the primary diagnostic tools but omits two important lines of work that are now standard in the SBI community. Hermans et al. [2] showed that posterior approximators can be systematically overconfident in regions of low prior mass while appearing well calibrated globally, and introduced local expected coverage tests that detect this failure mode; this result directly motivates the observation in Section 7 that C2ST is fragile in high dimensions, but the connection is never drawn. Lemos et al. [3] proposed sampling-based highest-posterior-density coverage tests that provide marginal frequentist guarantees without requiring a reference posterior, which is directly relevant to the high-dimensional ODE setting in Case Study 2 where reference samples are expensive. Neither paper is cited, despite being among the most widely adopted diagnostic tools in the field. A tutorial review that recommends C2ST and SBC without acknowledging these alternatives and their relative strengths cannot serve as an authoritative reference for practitioners.

The discussion of guidance and score composition in Section 4 would benefit from an explicit treatment of the bias introduced by guidance, which the paper acknowledges but does not quantify. The statement that samples "will align with the marginals implied by the guidance, even if intermediate trajectories differ" (attributed to Vuong et al. [11]) is presented without conditions or caveats, yet the guidance bias problem is precisely what limits the reliability of compositional estimation when error accumulation becomes severe, as Touron et al. [12] show analytically. A tutorial review on this topic should give readers the tools to diagnose when guidance is safe rather than simply referencing the phenomenon.

The paper recommends "VP EDM with F-parameterization and adaptive SDE solver" for low-dimensional problems and "flow matching with adaptive higher-order ODE solver" for high-dimensional problems, but the boundary between these is never defined. Since the case studies do not include a systematic sweep across parameter dimensionality within a fixed problem family, the dimension at which the recommendation changes cannot be inferred from the results.

---

> ### Author Response · Authors · 2026-06-01
>
> We thank the reviewer for the detailed comments. We agree that several statements in the original version could be read as stronger than intended, especially regarding the novelty of individual model classes and the generality of our empirical recommendations. In the revision we therefore substantially address these statements.
>
> Our intended contribution is to provide a **controlled empirical comparison of these design choices** under matched simulation, architecture, training-budget, and diagnostic conditions. Much of the current literature evaluates individual parameterizations, samplers, or generative objectives in separate papers, on different simulators, with different architectures, different posterior diagnostics, and different computational budgets. This fragmentation makes it difficult for practitioners to determine whether performance differences are due to the modeling choice itself or to uncontrolled variation in implementation, benchmark choice, or evaluation. Prior design-space studies the reviewer references (Karras et al., Song et al., Holderrieth and Erives) optimize for sample fidelity rather than posterior accuracy: a different optimization target, under which different design choices might be optimal.
>
> Crucially, our exhaustive **Table 3 clearly shows that most current SBI papers still use suboptimal designs** like noise-prediction instead of v-prediction and there has not yet been a controlled SBI-specific comparison isolating these effects, which motivates the head-to-head comparison under SBI-relevant metrics that the paper provides.
>
> Beyond the empirical comparison itself, the manuscript also **synthesizes several SBI-specific topics that are currently dispersed across the literature**: classifier-free guidance specific to SBI (guidance using constraints, prior-time adaptation, simulator feedback) and inference under structured targets (multiscale, causal, and joint estimation, as well as score learning beyond Euclidean spaces). These topics are dispersed across the SBI literature and are not synthesized or brought under a common denominator elsewhere.

---

> ### Author Response · Authors · 2026-06-01
> **Continuous consistency models**
>
> > The presentation of continuous consistency models as a novel SBI contribution is supported only by a passing result in Table 4, where the continuous consistency model underperforms relative to the discrete variant and to flow matching on the large-scale ODE case study. The empirical evidence therefore argues against this as a positive contribution.
>
> The Table 4 result the reviewer cites reflects only Case Study 2, the 72-parameter ODE. In Figure 10, we apply the continuous consistency model to the Lueckmann benchmark problems and show that it performs on par with the discrete variant and is competitive with diffusion models. Our results show that consistency models (both discrete and continuous) become harder to train as dimensionality grows; for the discrete variant this is also visible in the GRF benchmark. The contribution is therefore a regime-dependent finding: parity with the discrete variant at low dimension, degradation at higher dimension. We have revised the contribution language and the discussion accordingly:
>
> Revised text in bold: *Consistency models offered dramatically faster inference (few network evaluations vs. 100 or more for other approaches) with comparable NRMSE on low-dimensional targets but slightly elevated calibration error, suggesting a practical speed-accuracy tradeoff. **For increasing dimensionality of the target discrete and even more so continuous consistency models show decreasing accuracy, making them less suitable for higher-dimensional tasks in SBI**.*

---

> ### Author Response · Authors · 2026-06-01
> **GRF benchmark**
>
> > The GRF benchmark is described as a new contribution, but the benchmark itself is a straightforward two-parameter power-law spectral model with no simulator mismatch, no sequential structure, and no likelihood intractability; it is significantly simpler than the ODE models already in common use [1] and contributes less as a standalone benchmark than as an illustration of dimension scaling.
>
> > The GRF benchmark requires substantive extension to serve as a lasting reference. A two-parameter spectral model is not representative of the high-dimensional intractable-likelihood settings that motivate SBI; the authors should consider whether an alternative benchmark with genuine likelihood intractability and at least moderate parameter dimensionality would better illustrate the design-choice trade-offs they intend to characterize. As it stands, the GRF results in Figure 12 largely confirm that diffusion models and flow matching scale to image generation, which is already well established outside SBI.
>
> We may have communicated the structure of Case Study 3 less clearly than we intended, and we are happy to fix this. The main purpose of the GRF benchmark is not to introduce a maximally realistic SBI task, but to provide a controlled setting in which observation and target dimensionality can be scaled independently within a fixed simulator family. The GRF benchmark consists of two distinct tasks within a single controlled simulator family: (a) inferring two-dimensional spectral parameters from increasingly high-dimensional observations (Figure 12A), and (b) inferring increasingly high-dimensional random fields as the target (Figure 12B). The design allows the dimensionality of both target and observation space to be varied within a fixed simulator.
>
> The two-parameter power-law spectrum the reviewer describes is the lowest parameter configuration in (a), not the entirety of the benchmark. The benchmark is what allows us to observe how consistency models begin to fail at higher dimensions and how the role of the simulation budget changes with dimension. This is then evaluated by classic SBI metric, namely C2ST, and not the standard FID on images used outside the SBI literature. We also stated that “The field would benefit enormously from expanded benchmark suites with controllable dimensionality regimes and simulation budgets”. Furthermore, upon request of reviewer SAEw, we will add an ablation study to quantify the impact of the backbone (UNet vs UVit) with respect to dimension, simulation budget and diffusion model type.
>
> Revised text in bold:
> *Our GRF benchmark is **a controlled setting in which observation and target dimensionality can be scaled independently within a fixed simulator family and** designed to answer three questions: (i) how different is performance between a variance-preserving EDM-based diffusion model (DM), flow matching (FM), and a consistency model (CM) in the low-dimensional target regime where the condition is a high-dimensional field; (ii) which methods remain reliable when the target is switched and the task becomes to generate a high-dimensional field, and (iii) how does performance change as a function of simulation budget.*

---

> ### Author Response · Authors · 2026-06-01
> **Practical recommendations**
>
> > The statement that "for most problems, the VP schedule is a suitable choice" is repeated across sections but depends heavily on the problem class, and the paper's own results show flow matching outperforming VP EDM on the high-dimensional case (Table 4, rank 1), which undermines the universality of this recommendation.
>
> > The empirical findings confirm existing intuitions [...].
>
> > Since the case studies do not include a systematic sweep across parameter dimensionality within a fixed problem family, the dimension at which the recommendation changes cannot be inferred from the results.
>
> > The paper assembles known results without offering a unifying perspective that would allow a reader to predict, rather than merely observe post hoc, which design choices will succeed on a given problem.
>
>
> Thank you for these comments. We agree that the original recommendation language was too broad. The VP schedule is not a universal recommendation. We have revised the recommendations accordingly to match the empirical evidence.
>
> As can be seen from our table 3, in the SBI community this is not yet adopted “existing intuitions”. Across our case studies, VP EDM with F-parameterization and an adaptive SDE solver along with flow matching gives the most reliable performance in the low-dimensional regime exemplified by Case Study 1. Flow matching with an adaptive higher-order ODE solver marginally outperforms VP EDM in the high-dimensional regimes of Case Studies 3, but shows slightly worse calibration in case study 2. Continuous and discrete consistency models are competitive when sampling speed is a central constraint and are well-suited to low-dimensional problems, but they degrade at higher dimension and should not be used as a default choice for high-dimensional posterior estimation.
> Case study 3 does a systematic sweep across parameter dimensionality, and shows that not only low vs high dimensionality but also simulation budget plays a role. A simple boundary holding for any kind of problem can therefore not be defined. We revised the corresponding section to reflect this.
> We also updated case study 2 to include TARP as a metric and now compute the rank as a sum of the empirically standardized metrics (before only NRME and calibration error). This makes the rank now less sensitive to scales and changes the overall ordering a bit.
> As suggested by reviewer SAEw, we will add a flow chart to the empirical results to suggest models for practitioners allowing them to select the appropriate model family before running the task itself.
>
> We removed the sentence: *In our case studies, we find that for most problems, the VP schedule is a suitable choice.*
>
> Revised text in bold:
> Case Study 1:
> *For low-dimensional problems, we recommend to use diffusion models with a variance-preserving EDM noise schedule with F parameterization **or flow matching as both achieve comparable accuracy, while the latter is easier to implement and has less hyperparameters to tune**. Across problems and samplers, **these emerge** as the most reliable choices.*
>
> Case Study 2:
>
> ***In high dimensional settings, the picture changes. Consistency models are no longer competitive with the other model families due to worse NRMSE and calibration.
> Flow matching and diffusion models were the most reliable choices in this experiment, but they emphasized slightly different aspects of performance. Flow matching achieved the lowest NRMSE, although the improvement over the best diffusion models was marginal.
> Diffusion models, in turn, showed slightly better calibration. Among the diffusion variants, the VP noise schedule combined with either the $\mathbf{F}$- or $\mathbf{v}$-parameterization remained the most stable configuration.
> Direct noise prediction with $\boldsymbol{\epsilon}$ was prone to instability and should be avoided in this setting.***
>
>
> Discussion:
>
> *Our empirical evaluations across the first three case studies reveal that design choices matter substantially, but their impact is dimension- **and simulation-budget** dependent. For the low-dimensional benchmark problems (Case Study 1, see subsection 6.1), diffusion models with a variance-preserving EDM schedule **or flow matching models** were the best choices, **while flow matching has fewer hyperparameters to tune and is therefore easier to implement**. For high-dimensional inference (Case Study 2,  see subsection 6.2), **diffusion models achieved the best ranking because of stronger calibration, while flow matching achieved the lowest NRMSE and remained competitive overall, while the variance-exploding EDM schedule exhibited higher calibration errors in this setting. However, not only dimensionality plays a role for choosing a specific design, but also the available simulation budget, making it hard to define a specific boundary where one model family performs better than the other one.***

---

> ### Author Response · Authors · 2026-06-01
> **Novelty**
>
> > The theoretical exposition in Sections 2 through 4 is competent but synthesizes material already unified in prior tutorial treatments, including the BayesFlow framework [5], the original score-based SDE paper [6], the EDM formulation [7], and the flow matching tutorial [8].
>
> Thank you. We agree that the manuscript does not introduce a fundamentally new generative modeling framework. Its contribution is instead to synthesize, evaluate, and contextualize existing diffusion and flow-based approaches under SBI-specific conditions. Indeed we build on the mentioned papers, however besides [5] which does not cover diffusion models, all of these papers concentrate on unconditional models or use classifier-free guidance concepts, hence we present their results in a more SBI appropriate manner for conditional models. As can be seen from our table 3, in the SBI community design choices and their consequences are not yet entirely known to the community as this is a rapidly growing field.
>
> The manuscript also synthesizes several SBI-specific topics that are currently dispersed across the literature: classifier-free guidance specific to SBI (guidance using constraints, prior-time adaptation, simulator feedback) and inference under structured targets (multiscale, causal, and joint estimation, as well as score learning beyond Euclidean spaces). These topics are dispersed across the SBI literature and are not synthesized or brought under a common denominator elsewhere.
>
> Introduction:
> *As diffusion models have been rapidly adopted in SBI and now underpin numerous recent developments (Table 3), this paper addresses three needs (Figure 2): First, it introduces the foundations of SBI (section 2) and diffusion models **for SBI** (section 3) in a tutorial style, establishing a common conceptual and methodological baseline. Second, it provides a scoping review of diffusion model applications and adaptations in SBI (section 4). Third, it elucidates, via conceptual exposition (section 5) and tutorial-style empirical demonstration (section 6), the specific design considerations needed to turn diffusion models into general-purpose SBI engines.*

---

> ### Author Response · Authors · 2026-06-01
> **Missing diagnostics**
>
> > The submission discusses SBC and C2ST as the primary diagnostic tools but omits two important lines of work that are now standard in the SBI community. [...]
>
> We agree that the original diagnostic discussion was incomplete. The revised manuscript adds the relevant papers of Hermans et al. and Lemos et al. Moreover, we added TARP to the case study 2, which shows a similar ranking as the empirical calibration error reported in the original manuscript. We also note that random-point coverage diagnostics can be understood within the SBC framework as a particular choice of test quantity (Modrak et al., 2025). We now compute the rank as a sum of the empirically standardized metrics to include TARP as well. This makes the rank now less sensitive to scales.
>
> Excerpt of the new table for case study 2:
>
> *Case Study 2: Inference performance of diffusion models with their respective samplers on 1000 datasets.} Metrics: normalized RMSE (NRMSE), posterior calibration error, posterior contraction, classifier two-sample test (C2ST), and tests of accuracy with random points (TARP) (subsection A.1 for details). The models are ranked by the sum of all empirically standardized metrics. Numerical values represent the mean (standard deviation) over 10 repeated runs.*
>
> | Family | Design choice | Sampler | NRMSE | Calibration error | Contraction | C2ST | TARP | Rank |
> |---|---|---|---:|---:|---:|---:|---:|---:|
> | Diffusion | EDM, VP, F | ODE | 0.42 (0.005) | **0.03 (0.001)** | 0.68 (0.004) | 0.57 (0.008) | -0.02 (0.005) | 1 (2) |
> | Diffusion | Cosine, VP, v | SDE | 0.39 (0.005) | 0.04 (0.003) | 0.70 (0.003) | 0.59 (0.016) | -0.02 (0.004) | 3 (2) |
> | Flow Matching | ρ = -0.6 | ODE | 0.41 (0.005) | 0.05 (0.004) | 0.69 (0.005) | 0.59 (0.016) | 0.01 (0.005) | 7 (2) |
> | Flow Matching | Uniform | ODE | **0.39 (0.003)** | 0.07 (0.002) | 0.77 (0.005) | 0.59 (0.016) | -0.06 (0.006) | 10 (2) |
> | Flow Matching | OT | ODE | **0.39 (0.003)** | 0.07 (0.002) | 0.76 (0.004) | 0.59 (0.010) | -0.05 (0.004) | 11 (1) |
> | Consistency | Continuous | ODE | 0.70 (0.010) | 0.14 (0.004) | 0.49 (0.013) | 0.52 (0.010) | 0.15 (0.003) | 14 (1) |
>
> Revised text in bold:
> *Thus, proxy metrics that bypass evaluation, such as simulation-based calibration (SBC;
> Talts et al., 2018; Yao & Domke, 2023; **Lemos et al., 2023**, Modrák et al., 2025), are typically used and actively researched (e.g., Säilynoja et al., 2025; Bansal et al., 2025).*
>
>
> Discussion:
> *Taken together, these findings suggest that C2ST can be useful but fragile, and it also comes with additional design choices: one must decide how many and which samples to use for training the classifier. This is an expensive step in high-dimensional SBI, where sampling from multi-step generative models is costly, and the classifier architecture itself has to be adapted to the data structure, adding hyperparameters and complicating comparisons across studies. **Hermans et al. (2022), showed that posterior approximators can be systematically overconfident in regions of low prior mass while appearing well calibrated globally, which is especially true in high-dimensions, where the region of the target usually has low prior mass.** Developing calibration diagnostics that remain informative in high dimensions and can distinguish approximation error from misspecification represents an urgent methodological need. Recent methods such as posterior SBC (Säilynoja et al., 2025) and alternative test quantities for SBC (**Lemos et al., 2023**; Modrák et al., 2025) offer promising directions, but their computational cost and sensitivity in extreme dimensions require further improvements.*

---

> ### Author Response · Authors · 2026-06-01
> **Guidance and score-composition bias**
>
> > The additive structure of scores under Bayesian factorizations is a known property exploited in Geffner et al. [9] and Linhart et al. [10]; the present paper reviews this material without extending it analytically.
>
> > The discussion of guidance and score composition in Section 4 would benefit from an explicit treatment of the bias introduced by guidance, which the paper acknowledges but does not quantify. The statement that samples "will align with the marginals implied by the guidance, even if intermediate trajectories differ" (attributed to Vuong et al. [11]) is presented without conditions or caveats, yet the guidance bias problem is precisely what limits the reliability of compositional estimation when error accumulation becomes severe, as Touron et al. [12] show analytically. A tutorial review on this topic should give readers the tools to diagnose when guidance is safe rather than simply referencing the phenomenon.
>
>
> Thank you for pointing this out. Touron et al. (2025) was already cited in the corresponding section but the treatment was too brief. We have expanded the presentation of their findings in Section 4 and added them to the discussion. However, we think that a full treatment of the analytical derivation would be beyond the scope of this review. We also improved the link to Vuong et al. as we did not want to claim that guidance is not inducing any bias unconditionally.
>
> Revised text in bold:
> *Ensuring $\Lambda(\theta_t)$ to be positive definite makes the approximation more stable (Gloeckler et al., 2025), but Touron et al. (2025) show that error accumulation under the assumptions of GAUSS is directly related to the sampling quality of each individual posterior **and provide a bound on the mean squared error between the compositional score and its estimate as a function of the individual score errors and the error in the estimation of the precision matrix $\Sigma^{-1}_{t,n}$.***
>
> *However, the flexibility that makes guidance attractive also carries a cost: any guidance biases the reverse diffusion process (Chidambaram et al., 2024). Changing the score means that one implicitly assumes that the marginal density $p(z_t | z_0)$ of the reverse diffusion process changes accordingly. Depending on the type of guidance, this can make the new score unstable and require Langevin, MCMC-based **or weighted samplers** to correct for the error (Geffner et al., 2023; Sjöberg et al., 2023; **Skreta et al., 2025**). However, the need for correction depends on the application and the guidance strength. **Moreover, Vuong et al. (2025) argue that under a Wasserstein-gradient-flow interpretation, diffusion sampling can remain effective even when the neural vector field is not an exact score, since correct marginal transport does not require exact reverse-time path equivalence. This, however, does not remove guidance bias; a guidance term still changes the effective vector field, and the resulting marginal flow is reliable only insofar as this altered field remains compatible with the intended density evolution. Hence, applying guidance in the context of SBI necessitates careful checking of posterior calibration to quantify potential bias induced by guidance**.*
>
>
> Discussion:
> *The fundamental question remains: under what conditions does guidance preserve statistical accuracy? Recent work by Vuong et al. (2025) suggests that samples **may** align with the marginals implied by the guidance, even if intermediate trajectories differ from true reverse diffusion. However, extensive characterization of the conditions under which this holds represents a critical open problem. This is particularly pressing for compositional estimation (Geffner et al., 2023; Linhart et al., 2024; Gloeckler et al., 2025; Arruda et al., 2025, **Touron et al., 2025**), where score composition across many observations amplifies guidance-induced errors. **Practitioners may assess whether guidance hyperparameters produce calibrated samples using calibration diagnostic metrics, and, if needed, further refine these hyperparameters through approaches such as Bayesian optimization targeting these diagnostics**.*

---

### Review · Reviewer_SAEw · 2026-05-19

**Summary Of Contributions:**

## Summary:
This survey review explores the use of diffusion models for simulation-based inference (SBI), which involves inferring latent parameters of a simulator from observed data when likelihoods are intractable. The work covers the problem setting of SBI, the foundations of diffusion models (DMs), as well as more advanced DM topics including posterior and likelihood estimation, joint estimation, and conditional distribution modelling. The work explores more topics on special use cases of DMs' score functions in SBI, including guidance during inference and score function composition. The survey then comprehensively covers design choices of DMs specifically for SMI, including noise schedules, weighting functions, model parameterizations, etc. It uses experiments to compare different design choices and makes recommendations based on the empirical study results. The work concludes by discussing open challenges around guidance-induced bias, calibration diagnostics in high dimensions, and robustness to model misspecification.

## Strengths:
1. The significance of the surveying topic cannot be overstated. The topic is at the intersection of many important and rapidly advancing areas including diffusion models, AI4Sciences, Bayesian machine learning, etc., with a broad audience.
2. The subtopics surveyed by the work are very thorough from generic DM training and inference to special topics related to SBI like guidance during inference, score function composition, and model selection. The works are also very timely with work up to 2025. It is a strong survey on what DM can and cannot do in SBI and how to use DM for SBI.
3. The work also provides empirical study results in Section 6. This is unusual for a surveying work and actually does more than what a survey work could do.
4. The work is very well-organized and presented. For example, the flow chart in Figure 1 provides useful guidance for paper reading and makes it easy to follow. Figure 2 based on toy examples also provides additional clarifications on different terminologies.

## Weakness:
I don't see any major weakness in this work.
I have a few minor concerns. Please see requested changes.

**Audience:**

Yes

**Audience Explanation:**

As I said in my summary, the topic of this survey work, DM based SMI, is of interests to many important and rapidly advancing areas including diffusion models, AI4Sciences, Bayesian machine learning, etc in the machine learning and AI research community.

**Broader Impact Concerns:**

As a survey and tutorial paper, there's no broader impact that could directly come from this work itself.

**Claims And Evidence:**

Yes

**Claims Explanation:**

The new claims made by the work come from Section 6. Most of them are well supported, including:
1. The claims on impacts and competitiveness of different design choices in Section 6.1, covering model family, noise schedule, parameterizations, and sampler,
2. the characterizations and comparisons of DM, FM, and CM under different settings for GRM modelling in Section 6.3,
3. the claims on compositional score matching with inference in different pooling regimes made in Section 6.4.

However, some claims might require further clarifications from the authors:
1. The authors made recommendations for flow matching in both Section 6.1 (low-dimensional case) and Section 6.2 (high-dimensional case) based on the empirical results. But the experiment results also show the performance gap between FM and DM is not significant. For example, the calibration error gap between FM and DM in Table 2 can be negligible, but this metric alone has a much larger MAD in each model than the gap between DM and FM. Similar problems can be found in Figure 10.

**Requested Changes:**

I would like to suggest the following changes:
1. To justify the recommendations of FM over the other diffusion models, I would like the author to have experiment results that shows more significant gaps. The authors could consider other metrics, or multi-seed experiment results. In addition, the authors could also make the case for FM based on other criteria like the easiness of implementation or hyper-parameter choice.
2. For each setting in each study case, the author only consider one type of model architecture like MLP, U-Net or SetTransformer. I would advise the authors consider a case study that can compare different DM model architectures that also include DiT in the architecture comparison.
3. I would also recommend the model to summarize the empirical study results in decision trees or flow chart in the style of "if setting A, use design choice B". In addition, each case study is based on a specific benchmark dataset. I would like to see discussions on how conclusions from these case studies to data or tasks not covered the case studies.
4. Some prior works might be missing for the references of the work:
- Sohl-Dickstein et al. (2015) — "Deep Unsupervised Learning using Nonequilibrium Thermodynamics". This is the original diffusion model paper.
- Peebles et al. (2023) — "Scalable Diffusion Models with Transformers". DiT is an important incremental work in the study of diffusion model architectures.

---

> ### Author Response · Authors · 2026-06-01
> **Clarifications on model recommendations, backbone ablations, empirical limitations, and related work**
>
> Thank you for the overall positive and constructive feedback. Regarding the suggested changes:
>
> 1. The recommendation of FM over the other diffusion models would indeed benefit from a **more careful justification. We repeated the case study 2 as a multi-seed experiment and added TARP as another metric**. We now compute the rank of the methods based on the performance across all metrics (NRMSE, ECE, contraction, TARP, C2ST) and check the rank across runs, which changes the rank and our recommendation. We will update the experimental and discussion part to make more careful recommendations (see our answers to reviewer wsPp) and visualize this in a flow chart. In particular, we will recommend:
>     - Consistency models perform well in low-dimensional settings and are faster during inference at the cost of accuracy, which might be acceptable for an initial analysis of a scientific model. However, for high accuracy or density evaluation they are usually not suitable.
>     - In general diffusion models match more or less flow matching models. However, flow matching models have fewer hyperparameters, making them a suitable and robust choice for settings where high accuracy is required. We noted that for flow matching adaptive solvers often can take larger steps than for diffusion models showing that the learned flow matching velocity might be smoother. For tasks where post hoc adaptation such as guidance or compositional approaches can be employed, we recommend diffusion models.
>
> 2. In case study 3, we will **add a study on different backbones** comparing UNet vs UVit and analyze the impact of the backbone depending on whether we use diffusion, flow matching or consistency using different simulation budgets. We expect UVit to perform at least as good if not better than UNet for sufficient training.
>
> 3. We agree that the current case studies do not cover all possible data modalities or target types, and we will make this limitation more explicit in the discussion. Our case studies **focus on scientifically common settings in which the simulator output is either a time series or an unordered set of observations**. Other data modalities, such as graph-structured observations, were not considered empirically but mentioned in Section 4.3.4 (Learning Scores Beyond Euclidean Spaces). We agree that our experiments only consider continuous real-valued target parameters and not discrete or mixed discrete-continuous parameters. This is an important limitation, especially in light of recent work. We will **add a discussion of this limitation** and clarify that extending the proposed approach to discrete or mixed target spaces is an interesting direction for future work. We note that in posterior inference one can view most extensions primarily as a question of choosing an appropriate data encoder as a summary network. For example, as a Set Transformer can encode unordered sets into a fixed-dimensional representation, graph neural networks could be used to encode graph-valued simulator outputs before applying the inference network. Thus, while the encoder would need to be adapted, the diffusion-based posterior estimation framework itself is not restricted to the specific data modalities used in our case studies.
>
>
>    Added part in the discussion:
>
>     *A limitation of these empirical comparisons is that our case studies do not exhaust the range of possible simulator outputs or target spaces encountered in SBI. The examples considered here focus on scientifically common settings in which the simulator output is either a time series or an unordered set of observations. Other modalities, such as graph-structured observations, were not studied empirically. We expect that many such extensions would either require adapting the diffusion model beyond Euclidean spaces or adapting the summary network rather than changing the diffusion-based posterior estimation framework itself. For instance, just as a SetTransformer can encode unordered observations into a fixed-dimensional representation, graph neural networks could be used to encode graph-valued simulator outputs (Zhou et al., 2020; Hoogeboom et al., 2022). A second limitation is that all case studies considered continuous real-valued target parameters, although discrete or mixed discrete-continuous parameters arise naturally in model choice, latent structure inference, and hierarchical Bayesian analysis (Schröder & Macke, 2024; Kucharský & Bürkner, 2026). Extending diffusion-based SBI to these settings is therefore an important direction for future work, particularly in light of recent work on SBI with discrete and mixed parameter spaces (Ghiglino et al., 2026; Gloeckler et al., 2026; Boelts et al., 2026).*
>
> 4. Thank you for pointing out these **important prior works**, which we are happy to add.

---

> > ### Author Response · Authors · 2026-06-16
> > **Added study on different backbones**
> >
> > In the revised manuscript, we have substantially expanded Case Study 3 to investigate the role of the backbone. We now compare an improved U-Net backbone against two transformer-based alternatives: U-ViT and Residual U-ViT, motivated by recent developments in image diffusion models, including Peebles and Xie (2023) and Hoogeboom et al. (2023, 2025).
> >
> > The revised results show that the choice of backbone has a substantial effect on simulation efficiency. In particular, transformer-based architectures are less sensitive to reductions in the simulation budget than U-Net-style architectures. This suggests that the architectural improvements introduced in modern image diffusion models can also translate into improved performance in simulation-based inference, especially in regimes where simulations are expensive and the available training budget is limited.

---

### Review · Reviewer_7cUS · 2026-07-16

**Summary Of Contributions:**

The paper provides a broad tutorial review of diffusion models for simulation-based inference (SBI). It unifies neural posterior, likelihood, and joint estimation under a common notation and reviews the main mathematical foundations, including forward diffusion, reverse-time SDEs, probability-flow ODEs, score matching, flow matching, conditional generation, and density estimation.

**Audience:**

Yes

**Audience Explanation:**

The paper addresses a timely topic at the intersection of simulation-based
inference, diffusion models, Bayesian computation, and scientific machine
learning. Its unified treatment of posterior, likelihood, and joint
estimation, together with discussions of score composition, guidance, flow
matching, consistency models, solver efficiency, and hierarchical inference,
will be useful to both researchers and practitioners.

The empirical finding that suitable diffusion-model design choices depend on
the target dimension, observation dimension, simulation budget, and inference
cost is also relevant to the TMLR audience. Therefore, the paper has clear
audience interest, although some mathematical and empirical claims require
stronger validation.

**Broader Impact Concerns:**

No major ethical concern is apparent, as the work uses synthetic or mechanistic simulators and does not directly involve human subjects. However, it should be noted that inaccurate or misspecified SBI posteriors could lead to overconfident decisions in high-stakes areas such as biomedicine, epidemiology, climate science, and engineering.

**Claims And Evidence:**

Yes

**Claims Explanation:**

The paper provides evidence that diffusion models are flexible SBI approximators and that their performance depends strongly on the noise schedule, parameterization, sampler, target dimensionality, and simulation budget. Its experiments also support the practical trade-off that flow-matching variants can perform well in higher-dimensional settings, while consistency models substantially reduce sampling time. However, the broader theoretical and best-practice claims are only partially supported because of mathematical inconsistencies and insufficiently controlled comparisons. For example, Eqs. (7)–(8) use the transition score $\nabla_{z_t}\log p(z_t\mid z_0),$ whereas the reverse-time SDE and probability-flow ODE require the marginal conditional score $\nabla_{z_t}\log p_t(z_t\mid x).$ In addition, Eq. (33) appears to contain an incorrect posterior factorization.

**Requested Changes:**

- Correct the interpretation of reverse diffusion. It generates a new sample from the target distribution rather than recovering the original training sample.

- Clarify that the probability-flow ODE shares marginal distributions with the SDE; it is not the average of individual SDE trajectories.

- Present Eq. 29 as an approximation unless the authors provide assumptions under which score composition remains exact after Gaussian noising.

- Analyze compositional-score error with respect to the number of factors, diffusion time, mini-batch size, damping function, posterior overlap, and non-Gaussianity.

---

> ### Author Response · Authors · 2026-07-21
> **Corrections of minor inconsistencies in the mathematical notation**
>
> > The paper provides evidence that diffusion models are flexible SBI approximators and that their performance depends strongly on the noise schedule, parameterization, sampler, target dimensionality, and simulation budget. Its experiments also support the practical trade-off that flow-matching variants can perform well in higher-dimensional settings, while consistency models substantially reduce sampling time. However, the broader theoretical and best-practice claims are only partially supported because of mathematical inconsistencies and insufficiently controlled comparisons. For example, Eqs. (7)–(8) use the transition score $\nabla_{z_t} \log p(z_t \mid z_0)$ whereas the reverse-time SDE and probability-flow ODE require the marginal conditional score $\nabla_{z_t} \log p(z_t \mid x)$. In addition, Eq. (33) appears to contain an incorrect posterior factorization.”
>
> Thank you for these valuable and thorough comments. Indeed, the notation mixed regression targets with the marginal score. We corrected the notation, and equations (7)-(8) now use the score $\nabla_{z_t} \log p(z_t)$, which is unknown. The following equations (9)-(10) then explicitly show that the conditional denoising loss is equivalent to the unconditional denoising loss (see changes below). In the reverse process, the learned score can then be used for sampling. In Eq. (33), the later term should be indeed a posterior, not a likelihood, which we corrected.
>
>
> The new paragraph reads as follows:
>
> As (7) and (8) show, simulating
> the reverse process requires the score $\nabla_{z_t}\log p(z_t)$ of the marginal
> distribution. This poses a problem for training: $p(z_t)$ depends on the unknown
> data distribution and is usually not available in closed form.
> The \emph{conditional} score
> $\nabla_{z_t}\log p(z_t \mid z_0)$, by contrast, is known analytically: for the forward process in (5), the conditional score is given by $\nabla_{z_t}\log p(z_t \mid z_0) = -\epsilon_t/\sigma_t$.
>
> \emph{Denoising score matching}
> (Vincent, 2011; Song & Ermon, 2019) exploits the property that regressing onto the conditional score $\nabla_{z_t}\log p(z_t\mid z_0)$ actually recovers the marginal score $\nabla_{z_t}\log p(z_t)$.
> Hence, we can train an approximator (usually a
> deep neural network) $\hat{s}(z_t, t)$ via regression onto conditional scores:
> $$\hat{s} = \arg\min_s \mathbb{E}_{z_0\sim p(z_0), t\sim U(0,1), z_t\sim p(z_t\mid z_0)} \left[\omega_t \Vert s(z_t, t) - \nabla \log p(z_t) \Vert \right]$$
>
> $$= \arg\min_s \mathbb{E}_{z_0\sim p(z_0), t\sim U(0,1),\epsilon_t\sim N(0,I)} \left[\omega_t \left\Vert s(\alpha_t z_0+\sigma_t\epsilon_t, t) + \epsilon_t / \sigma_t \right\Vert\right].$$
>
>
>
> > Correct the interpretation of reverse diffusion. It generates a new sample from the target distribution rather than recovering the original training sample.
>
> Thank you for this comment. We changed the wording throughout the manuscript to make sure that new samples are from the target distribution and not an exact recovery of the original sample.
>
>
> > Clarify that the probability-flow ODE shares marginal distributions with the SDE; it is not the average of individual SDE trajectories.
>
>
> We appreciate this helpful request and updated the manuscript to clarify this point as follows:
>
> Alternatively to a reverse SDE, we can also define a deterministic denoising process that follows the *same marginal distribution $p(z_t)$*.
> This process is called the probability flow, [...]
>
> > Present Eq. 29 as an approximation unless the authors provide assumptions under which score composition remains exact after Gaussian noising.
>
> We clarified that the density in Eq. 29 might not be the marginal density of the reverse SDE, which is generally unknown, and which is the source of the bias already discussed in Section 4.1.1 in the context of guidance. We changed the equation to an approximation and added:
>
> However, compositional estimation *does not necessarily follow the true score of the marginal distribution $p(\theta_t \mid y_{1:N})$ induced by composition and is exact only for $t=0$*.

---

> ### Author Response · Authors · 2026-07-21
> **Scaling analysis of compositional sampling**
>
> > Analyze compositional-score error with respect to the number of factors, diffusion time, mini-batch size, damping function, posterior overlap, and non-Gaussianity.
>
> Thank you for this valuable comment. In case study 4, we now added a figure showing the effect of the number of factors, mini-batching, and damping on nrmse, posterior calibration, and the adaptive integration steps, which is relevant for practitioners to know. The posteriors in this example are non-Gaussian and each observation stems from an independent stochastic simulation.
>
> We added a new Figure 15 and added the following paragraph in the case study:
>
> \paragraph{Scaling behavior of compositional sampling}
> To analyze the behavior of compositional sampling, we varied the number of groups, the mini-batch size, and the damping factor $d_0$ (Figure 15). Adaptive SDE integration is necessary because the number of solver steps increases with the number of composed groups. By contrast, the mini-batch size has almost no influence on NRMSE, calibration, or integration cost, indicating that subsampling groups at each reverse step is sufficient to capture the difference between individual scores. The damping factor $d_0$ has a stronger effect as smaller values can improve calibration at the cost of sharpness (NRMSE), although overly strong damping eventually degrades calibration as well. The optimal choice of $d_0$ therefore depends on the number of groups being composed, but importantly, it can be selected at inference time without retraining the model.

---

### Author Response · Authors · 2026-07-21
**Improved manuscript with new experiments**

We thank the reviewers for their constructive feedback. We updated the manuscript by adding:
- Additional transformer-backbone experiments, including MLP versus transformer, UNet versus UViT, and an example using masked targets and observations showing the power of a transformer backbone.
- TARP as an additional diagnostic.
- A multi-seed evaluation of Case Study 2 and a revised model ranking across multiple metrics.
- A scaling analysis of compositional sampling, including the effects of mini-batch size, damping, and adaptive solver steps.
- An expanded discussion of guidance and compositional bias.
- A clearer account of the limitations of our case studies regarding data modalities and parameter types.
- Improved practical recommendations and a flow chart to guide practitioners.
- Corrections of minor inconsistencies in the mathematical notation.

We believe that these revisions have substantially improved the manuscript and strengthened both its theoretical presentation and practical relevance.